

# *Psittacosaurus houi*, a longer snouted psittacosaurid from the Lower Cretaceous Lujiatun Unit of Yixian Formation, China, with the synonymy of the unresolved genus *Hongshanosaurus* revisited

Asato Ishikawa[1], Wenjie Zheng[2], Takuya Imai[3,4], Soki Hattori[3,4], Masateru Shibata[3,4], Soichiro Kawabe[3,4] and Xingsheng Jin[2]

[1] Department of Bioscience and Biotechnology, Fukui Prefectural University, Eiheiji, Fukui, Japan
[2] Zhejiang Museum of Natural History, Hangzhou, Zhejiang, China
[3] Faculty of Dinosaur Paleontology, Fukui Prefectural University, Eiheiji, Fukui, Japan
[4] Fukui Prefectural Dinosaur Museum, Katsuyama, Fukui, Japan

Corresponding author
Asato Ishikawa,
aam.1995jp@gmail.com

## ABSTRACT

The taxonomic validity of some genera and species within Psittacosauridae has been disputed, including that of *Hongshanosaurus houi*, which has been synonymized with *Psittacosaurus lujiatunensis* previously. To assess the validity of the former genus and species and elucidate the taxonomy and diversity in Psittacosauridae, we describe a nearly complete psittacosaurid skull (ZMNH M12414) with the aid of computed-tomography techniques. The specimen comes from the Lujiatun Unit of the Lower Cretaceous Yixian Formation, Liaoning, northeastern China, which has also produced *Psittacosaurus major*, *P. lujiatunensis*, and *H. houi*. ZMNH M12414 exhibits a series of unique features that are also present in the adult referred skull of *H. houi* (IVPP V12617), indicating that both specimens are attributable to the same species within Psittacosauridae. The proportionally large length of the snout used to diagnose *H. houi* cannot be used for taxonomic distinction of *Psittacosaurus* because this character is also found in *P. amitabha*, supporting that the genus *Hongshanosaurus* is a junior synonym of *Psittacosaurus*. On the other hand, ZMNH M12414 and IVPP V12617 exhibit a set of features that are not observed in any other species of *Psittacosaurus*, including *P. lujiatunensis*. Because of these features, the specimen in question better fits as its own species within *Psittacosaurus*: *P. houi*. A phylogenetic analysis supports the validity of *P. houi*, where the specimens form an independent species within *Psittacosaurus*. The computed-tomography techniques employed in the present study facilitated re-assessment of the taxonomy and morphological diversity of *Psittacosaurus*, and its application is encouraged for previously described dinosaur taxa whose validities are in question. By utilizing such techniques, the detailed evaluation of ontogenetic, intraspecific, and interspecific variations will be crucial to understand the true taxonomy and diversity of *Psittacosaurus* in future studies.

## INTRODUCTION

Psittacosauridae represents a family of basal ceratopsian dinosaurs with a bipedal posture and characteristic upper and lower jaws that form a beak-like rostrum similar to that of a parrot (*Osborn, 1923*). In the Lower Cretaceous, hundreds to thousands of psittacosaurid specimens have been reported from the Barremian to Albian of China, Mongolia and Russia (*Sereno, 2010*; *Sereno, Zhao & Tan, 2010*; *Napoli et al., 2019*; *Podlesnov et al., 2023*). The genus *Psittacosaurus* is one of the possible two genera that comprise the family Psittacosauridae, and the most species-rich genus within non-avian dinosaurs (*Sereno, 2010*).

In the past, multiple species of *Psittacosaurus* were recognized, and their validities have been continuously tested. Along with the description of the type species, *Psittacosaurus mongoliensis* (AMNH 6254), *Osborn (1923)* described a specimen from the Ondai Sayr locality (AMNH 6253) and assigned it to a new genus and species, *Protiguanodon mongoliense*. In a study that followed, these two genera were placed within a single new family Psittacosauridae based on the overall similarity between the two (*Osborn, 1924*). Additionally, two new species, *P. osborni* and *P. tingi*, were erected following Osborn's studies (*Young, 1931*). In *Young (1958)*, another new species, *P. sinensis*, was described, which was followed by still another species, *P. youngi* (*Zhao, 1962*). In the 1980s, three additional species were described, *P. guyangensis*, *P. xinjiangensis*, and *P. meileyingensis* (*Cheng, 1983*; *Sereno & Chao, 1988*; *Sereno et al., 1988*). Subsequently, eight new species were added to the genus by the end of 2010 (*Russell & Zhao, 1996*; *Averianov et al., 2006*; *Zhou et al., 2006*; *Sereno et al., 2007*; *Sereno, Zhao & Tan, 2010*), bringing the total number of potential species of *Psittacosaurus* to 16 (*Sereno, 2010*). On the other hand, a comparative study by *Sereno (2010)* recognized nine valid species: *P. mongoliensis*, *P. neimongoliensis*, *P. sinensis*, *P. sibiricus*, *P. lujiatunensis*, *P. major*, *P. meileyingensis*, *P. xinjiangensis* and *P. gobiensis*. More recently, *Napoli et al. (2019)* added *P. amitabha* as a new species, with the total number of *Psittacosaurus* species still being debated.

Another genus that constitutes the family Psittacosauridae is *Hongshanosaurus*, which contains a single species *Hongshanosaurus houi*. The species was described based on a juvenile skull (IVPP V12704, holotype) and an adult skull (IVPP V12617, referred specimen) (*You, Xu & Wang, 2003*; *You & Xu, 2005*). *Hongshanosaurus* was distinguished from *Psittacosaurus* by an oval-shaped external naris and orbit and a longer rostrum (*You & Xu, 2005*). However, a lack of definitive morphological differences between *Hongshanosaurus* and *Psittacosaurus* has led to multiple proposals that the former may be a junior synonym of the latter. For example, *Sereno (2010)* argued that the longer rostrum in *Hongshanosaurus* might come from the taphonomic deformation and synonymized *H. houi* with *P. lujiatunensis*. Additionally, *Hedrick & Dodson (2013)* supported the synonymy of *H. houi* with *P. lujiatunensis* based on a three-dimensional geometric morphometric analysis in which *H. houi* formed a morphological cluster with

*P. lujiatunensis* and *P. major*. The study further tested the validity of the diagnosis of *P. lujiatunensis*, *P. major*, and *H. houi*, respectively, using 25 specimens of these psittacosaurids. The taxonomic analysis in *Hedrick & Dodson (2013)* showed that most of the diagnostic characters were intraspecifically and taphonomically variable, thus invalidating the diagnosis and leading to the conclusion that *P. major* and *H. houi* are junior synonyms of *P. lujiatunensis*. However, *Napoli et al. (2019)* criticized *Hedrick & Dodson (2013)* for failing to account for ontogenetic and taphonomic variation, as well as intraspecific variation found in modern animals, in their geometric morphometric analysis and argued that *P. lujiatunensis* and *P. major* were separate species because they did not form a sister clade in the phylogenetic analysis. While *Napoli et al. (2019)* did not consider the results of the taxonomic analysis by *Hedrick & Dodson (2013)*, their phylogenetic analysis possibly suggests that *H. houi* is also distinct from *P. lujiatunensis* and *P. major* if it had been included as another operational taxonomic unit (OTU).

In this study, we describe a well-preserved adult skull (ZMNH M12414) of a psittacosaurid with a long snout resembling that of *Hongshanosaurus houi*. We test the purported validity of the genus *Hongshanosaurus* and address current *Psittacosaurus* taxonomy based on the morphology and the phylogenetic analysis. The application of computed tomography (CT) techniques has allowed the non-destructive rendering of individual skull elements, including those that are difficult to observe in articulated specimens (such as endocranial elements), and helped us to describe each of them. This leads to a revision of diagnostic characters based on the less deformed and articulated specimen and allows us to address the taxonomy and specific diversity within Psittacosauridae. While the synonymy of *Hongshanosaurus* with *Psittacosaurus* is supported, our study recognizes *Psittacosaurus* (formerly *Hongshanosaurus*) *houi* as a valid species that can be distinguished from all other *Psittacosaurus* species, including the coeval *Psittacosaurus lujiatunensis*.

## MATERIALS AND METHODS

### Specimen and locality

ZMNH M12414 is represented by a mostly complete, articulated skull. The posterior surface of the skull is obscured by sandstone matrix (Figs. S1A, S1E). The specimen was collected from the Lower Cretaceous Lujiatun Unit that crops out near the village of Lujiatun, Beipiao, western Liaoning, in northeastern China. Unfortunately, the specimen was delivered to ZMNH without detailed locality and horizon data. The specimen was assigned to *Psittacosaurus* when deposited at ZMNH, and *Sakagami et al. (2023)* assigned it to *P. lujiatunensis*. However, no formal description was given prior to the present study.

The Lujiatun Unit is overlain by the Lower Lava Unit and comprises the lowermost part of the Yixian Formation, Jehol Group in northeastern China. The Lujiatun Unit in Lujiatun village has been extensively studied in *Rogers et al. (2015)* through logging, which is summarized as follows. Sedimentologically, the Lujiatun Unit generally consists of tuffaceous siltstones, grey siltstones, and a variegated sequence of sandstones from the bottom to the top of the unit. The deposition of these sedimentary horizons is interpreted as a result of a series of sheet flood following the remobilization of pyroclastic material

onto the floodplain. The sheet flood resulted in quick burials and fine preservation of the remains of terrestrial organisms in the Lujiatun Unit.

Volcanic tuff from the Lujiatun Unit cropping out in the Jin-Yang Basin have yielded U-Pb chemical abrasion-isotope dilution-isotope ratio mass spectrometry of 125.719 ± 0.025 Ma (Barremian, *Zhong et al., 2021*). The psittacosaurid species previously reported from the unit include *Psittacosaurus lujiatunensis* (*Zhou et al., 2006*), *P. major* (*Sereno et al., 2007*; *You, Tanoue & Dodson, 2008*), and *Hongshanosaurus houi* (*You, Xu & Wang, 2003*; *You & Xu, 2005*).

## Reconstruction of the skull using computed tomography techniques

The skull of ZMNH M12414 was scanned with a high-resolution X-ray CT (NIKON XTH 320; Nikon, Tokyo, Japan) at the College of Civil Engineering and Architecture, Zhejiang University, Hangzhou, Zhejiang, China. Tomographic images were obtained using a voltage of 300 kV, a current of 280 µA, an interslice spacing of 0.08 mm and an image size of 1,920 × 1,507 pixels. These parameters resulted in a voxel size of 1.00 mm, along the z-axis and 0.11 mm in the x- and y-axes. The resulting tomographic images (DICOM files) were segmented, and the three-dimensional computer model of the specimen was rendered using Amira ver. 2019.4 (Thermo Fisher Scientific, Tokyo, Japan).

## Observations and measurements

Observations and measurements of ZMNH M12414 were performed on the rendered three-dimensional computer model (Figs. S1B, S1D and S1F) with Amira ver. 2019.4 (Thermo Fisher Scientific, Tokyo, Japan). For comparative purposes, measurements and other anatomical information on previously described psittacosaurid specimens were obtained from the texts and figures in previously published literature (Table S1). Following *Sereno (2010)*, the preorbital length was measured from the anterior margin of the rostral to the anterior margin of the orbit, and the skull length was measured from the anterior margin of the rostral to the posterior margin of the quadrate condyle. In addition, the basal skull length was measured from the anterior margin of the rostral to the posterior margin of the occipital condyle, following *Bullar et al. (2019)* (Fig. S2).

## Phylogenetic analysis

To assess the phylogenetic relationships among species of the genus *Psittacosaurus*, *Psittacosaurus houi*, *P. sinensis* and *P. sibiricus* were added to the character-taxon matrix of *Napoli et al. (2019)*, which is based on the original matrix in *Han et al. (2018)*. *Psittacosaurus houi* was scored based on the rendered three-dimensional computer model of ZMNH M12414 (Figs. 1–2, S3–S33) and the descriptions of IVPP V12617 in the previously published literature (*You & Xu, 2005*; *Tanoue, You & Dodson, 2009*; *Taylor et al., 2017*; *Bullar et al., 2019*; *Landi et al., 2021*; *Han et al., 2018*). *Psittacosaurus sinensis* was scored based on published literature for the holotype IVPP V738 (*Young, 1958*; *Sereno, 1990*; *Tanoue, You & Dodson, 2009*; *Sereno, 2010*) and a referred specimen BNHM BPV149 (*Chao, 1962*; *Sereno, 2010*). *Psittacosaurus sibiricus* was scored based on published literature for the holotype PMTGU 16/4–20 (*Averianov et al., 2006*) and referred

specimens KOKM 22985/2 (*Podlesnov et al., 2023*) and PM TGU 16/0–15, 20, 30–39, 16/1–11, 51, 136, 137, 166, 167, 175, 176, 179, 200, 201–203, 209, 216, 223, 228, 258, 271, 274, 276, 281, 283 and 284 (*Averianov et al., 2006*).

The character-taxon matrix in *Napoli et al. (2019)* was further modified in the following perspectives. We chose to set the OTUs of *P. mongoliensis* and *lujiatunensis* species based on the holotype specimens. This was because it was possible that "*Psittacosaurus mongoliensis* composite" and "*Psittacosaurus lujiatunensis* composite" in the previous studies included the specimens belonging to different species. For *P. major*, LH PV1 and CAGS-IG-VD-004 were integrated into a single OTU. While character 103, the distinctive indentation on the midline of the posterior edge of the parietals, was scored as absent (1) in *P. amitabha* and *P. major* in *Napoli et al. (2019)*, they were scored as present (0) in the present study as it is apparent in the dorsal views (*You, Tanoue & Dodson, 2008*; *Napoli et al., 2019*). For character 238, the presence of 14 dorsal vertebrae was included in the first derived (15 vertebrae) state defined by *Han et al. (2018)*, allowing *P. sibiricus* with 14 dorsal vertebrae (*Averianov et al., 2006*) to be scored.

Following the previous studies (*Han et al., 2018*; *Napoli et al., 2019*), the matrix was analyzed using T.N.T. version 1.5 (*Goloboff, Farris & Nixon, 2008*), treating all characters as equally weighted and the following characters as ordered (additive): 2, 23, 31, 40, 126, 164, 196, 203, 204, 222, 227, 238, 243, 247, 268, 292, 296, 302, 306, 320, 361. The analysis was performed with the default settings, except that the maximum number of trees was set to 99,999. The traditional search was chosen to find the most parsimonious trees (MPTs) with the tree bisection-reconnection (TBR) algorithm, where 1,000 replicates of random stepwise addition and 100 trees held at each step. Then, a second TBR search was performed using the MPTs obtained in the previous procedure, where branches were collapsed if the minimal branch length was zero. The strict consensus tree was generated based on the MPTs found after the second TBR search. Bootstrap and Bremer support values were calculated as support indices.

# RESULTS

## Systematic Paleontology

Dinosauria *Owen, 1842*
Ornithischia *Seeley, 1888*
Ceratopsia *Marsh, 1890*
Psittacosauridae *Osborn, 1923*

*Psittacosaurus Osborn, 1923*

*Psittacosaurus houi You, Xu & Wang, 2003*

*Hongshanosaurus houi You, Xu & Wang, 2003* (original description)

*Psittacosaurus lujiatunensis You, Xu & Wang, 2003*: *Sereno, 2010* (synonymized)

*Psittacosaurus houi You, Xu & Wang, 2003*: Ishikawa, Zheng, Imai, Hattori, Shibata, Kawabe, and Jin, 2024 (new combination)

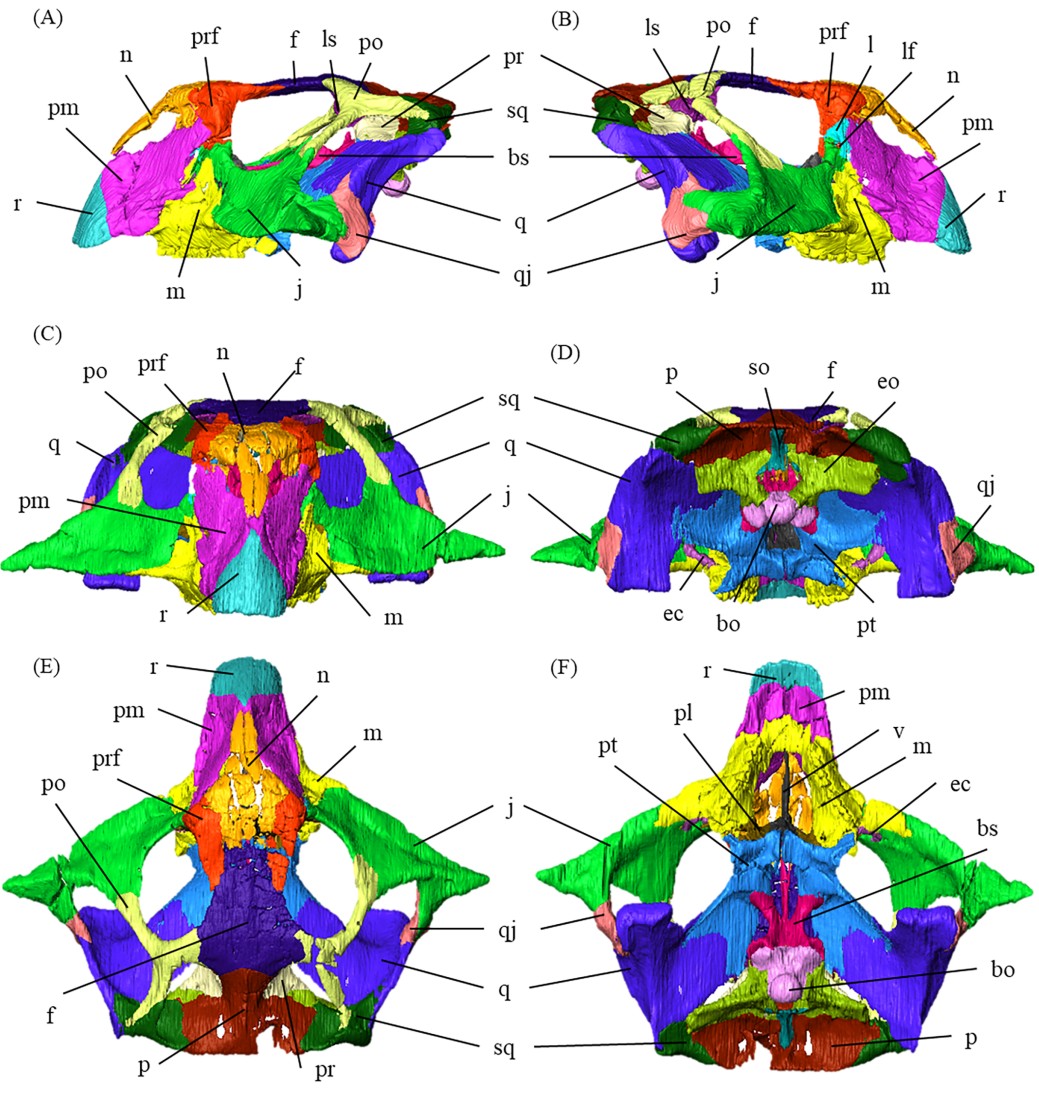

**Figure 1 Segmented cranium of *P. houi* (ZMNH M12414) in left lateral (A), right lateral (B), anterior (C), posterior (D), dorsal (E) and ventral (F) views.** Abbreviations: bo, basioccipital; bs, basisphenoid; ec, ectopterygoid; eo, exoccipital; f, frontal; j, jugal; l, lacrimal; lf, lacrimal foramen; ls, laterosphenoid; m, maxilla; n, nasal; p, parietal; pl, palatine; pm, premaxilla; po, postorbital; pr, prootic; prf, prefrontal; pt, pterygoid; q, quadrate; qj, quadratojugal; r, rostral; so, supraoccipital; sq, squamosal; v, vomer. Scale bar equals 50 mm.

Holotype—IVPP V12704, a nearly complete juvenile skull with lower jaws.

Referred specimens—IVPP V12617, a complete adult skull with lower jaws; ZMNH M12414, a complete adult skull with lower jaws.

Locality and horizon—Lujiatun, Liaoning, People's Republic of China; Lujiatun Unit, Yixian Formation, upper Barremian, Early Cretaceous (*Zhong et al., 2021*).

Diagnosis—*Psittacosaurus houi* is diagnosed by the following autapomorphies: (1) narrow prefrontal-premaxilla contact; (2) higher ventral margin of the premaxilla raised above the maxillary tooth row; (3) axes of maxillary tooth row and dorsal process of jugal oriented at

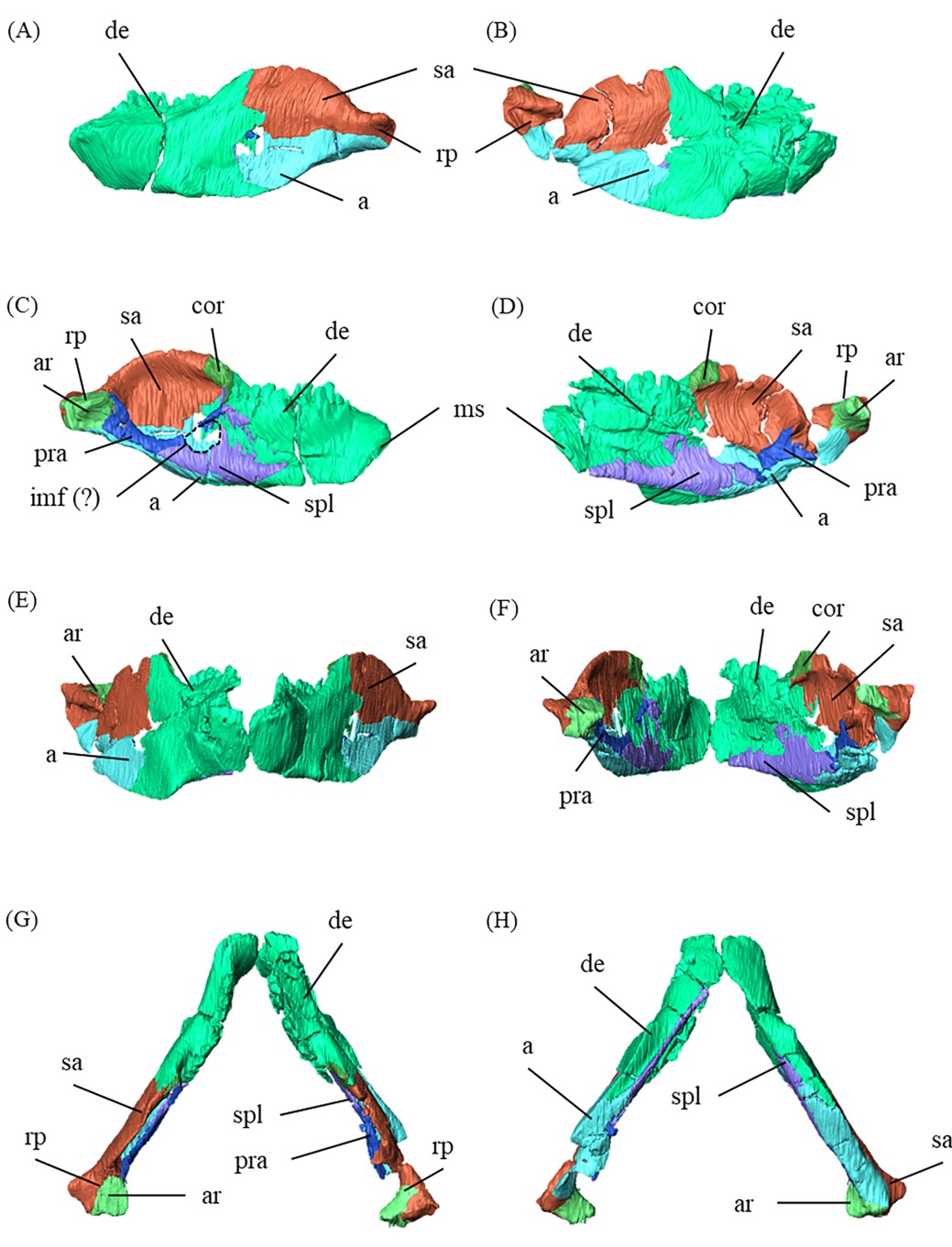

**Figure 2 Mandibles of *P. houi* (ZMNH M12414) with individual lower jaw elements colored in left lateral (A), right lateral (B), left medial (C), right medial (D), anterior (E), posterior (F), dorsal (G) and ventral (H) views.** Abbreviations: a, angular; ar, articular; cor, coronoid; de, dentary; imf, internal mandibular fenestra; pra, prearticular; sa, surangular; spl, splenial. Scale bar equals 50 mm.

an angle of about 135 degrees in lateral view; (4) subtriangular supraoccipital widest at its ventral margin; (5) long jugal bar of the postorbital, nearly twice the length of the temporal bar of the bone. Additionally, *P. houi* is distinguished from other *Psittacosaurus* species by the combination of the following characters: long preorbital region reaching about one half of the skull length (shared with *P. amitabha*); posterodorsally-elongated laterotemporal fenestra oriented at an angle of about 45 degrees in lateral view (shared with *P. amitabha*); nearly linear posterior margin of the parietal, perpendicular to the sagittal crest, with no midline indentation (shared with *P. meileyingensis*); and anterior margin of rostral and that is nasal gently sloped posterodorsally at an angle of 30 degrees from the vertical (shared with *P. amitabha*).

## DESCRIPTION AND COMPARISONS

ZMNH M12414 measures 123 mm in skull length, 196 mm in width from the lateral-most extents of the jugal horns, and 125 mm in the height from the ventral-most dentary flange to the dorsal-most skull roof (Table S2).

The skull is divided into 32 elements by suture lines observable in the CT data (Fig. S34). The skull exhibits fractures in some elements, including the premaxillae, nasals, the right prefrontal, postorbital, squamosal, parietal, angular, surangular, left maxilla/jugal contact, splenial, and dentary. It also shows slight distortion in the right supratemporal fenestra and mandibular condyle. However, these deformations do not appear to affect intra- and interspecific morphological comparisons (see Discussion).

Histological evidence suggests that IVPP V12617, the other referred specimen of *Psittacosaurus houi*, is fully mature and at least 10 years old at the time of death (*Zhao et al., 2013*). Because the basal skull length of ZMNH M12414 is 139.1 mm long and almost equivalent to that of IVPP V12617 (143.7 mm long; *Bullar et al., 2019*), we tentatively regard the former as mature. This interpretation about the full maturity of ZMNH M12414 is also supported by the cranial sutures that are nearly obliterated in some parts.

In dorsal view, the skull is wider than long (Fig. 1E), as in *Psittacosaurus sinensis* (IVPP V738; *Sereno, 2010*) and *P. lujiatunensis* (*Zhou et al., 2006*). Notably, the skull exhibits a relatively large proportion for the preorbital length from the anterior-most extent of the rostral bone to the anterior-most orbital margin, against the total skull length (the preorbital length proportion, PLP). The PLP against the total skull length is approximately 50%, being similar to 52% of IVPP V12617 and 46% of *P. amitabha* (*Napoli et al., 2019*), and unlike those of other known *Psittacosaurus* species which exhibit less than 40% (*Sereno, 2010*).

Rostral—In anterior view, the rostral is subtriangular with a broad ventral margin, and the nasal process extends dorsally (Fig. S3C). The rostral is also triangular in lateral view, having a nearly vertical sutural contact with the premaxilla (Figs. S3A, S3B). Because the anterior-most ends of the nasals are not fully preserved, the articulation between the rostral and nasal is obscured (Figs. 1A–1C). As in *Psittacosaurus mongoliensis* (*Osborn, 1923*), *P. sinensis* (*Sereno, 2010*; *Chao, 1962*), *P. amitabha* (*Napoli et al., 2019*), *P. major* (*Sereno et al., 2007*; *You, Tanoue & Dodson, 2008*) and IVPP V12617 (*You & Xu, 2005*), the anterior margin formed by the rostral and nasal slopes posterodorsally (Figs. 1A, 1B).

However, the margin slopes more gently at an angle of 30 degrees in ZMNH M12414, IVPP V12617 (*You & Xu, 2005*), and *P. amitabha* (*Napoli et al., 2019*). The anterior-most extent of the rostral is rounded in dorsal and ventral views (Figs. S3E, S3F), and neither pointed nor strongly bowed ventrally in lateral view (Figs. S3A, S3B).

Premaxilla—Both premaxillae are preserved and make up most of the lateral surfaces of the snout (Figs. 1A, 1B). The premaxilla contacts the rostral anteriorly, the nasal and prefrontal dorsally, and the lacrimal and maxilla posteriorly as in other species except for *Psittacosaurus sinensis* (*Sereno, 2010*), in which the premaxilla also meets the jugal posteriorly (Fig. S4). Unlike in other *Psittacosaurus* species (*Sereno et al., 1988*; *Russell & Zhao, 1996*; *Zhou et al., 2006*; *Sereno et al., 2007*; *You, Tanoue & Dodson, 2008*; *Sereno, 2010*; *Sereno, Zhao & Tan, 2010*), the sutural contact between prefrontal and premaxilla is relatively narrow in lateral view (Figs. 1A, 1B). A similar character, though even wider, is also found in *P. major* (LH PV1; *Sereno et al., 2007*), whereas the contact is even narrower in ZMNH M12414 and IVPP V12617 (*You & Xu, 2005*).

The anterodorsal portions of both premaxillae are fragmented and partially missing (Figs. S4A–S4D). The posterior expansion of the posterolateral process of the premaxilla excludes the maxilla from the external nares, which are bounded solely by the nasals and premaxillae, as in all other *Psittacosaurus* species (*Sereno et al., 1988*; *Russell & Zhao, 1996*; *Zhou et al., 2006*; *Sereno et al., 2007*; *You, Tanoue & Dodson, 2008*; *Sereno, 2010*; *Sereno, Zhao & Tan, 2010*; *Napoli et al., 2019*). The lateral surface of the premaxilla is smooth and slightly depressed below the external naris, but has multiple grooves and pits near its buccal margin, which is particularly evident on the right premaxilla (Fig. S4B) probably due to the erosion of the left premaxilla. A well-developed crest extends anterodorsally along the premaxillary-maxillary suture and a conspicuous groove crosses this suture horizontally just below the distal end of the premaxillary-maxillary ridge (Figs. S4A, S4B), as in *Psittacosaurus meileyingensis* (IVPP V7705; *Sereno et al., 1988*) and *P. lujiatunensis* (*Zhou et al., 2006*). In lateral view, the position of the ventral margin of the premaxilla is raised above maxillary tooth row (Figs. 1A, 1B), as in IVPP V12617 (*You & Xu, 2005*; *Bullar et al., 2019*). Similar features are present in *P. sinensis* (*Sereno, 2010*), *P. mongoliensis* (*Sereno, 2010*) and *P. major* (*Sereno et al., 2007*; *You, Tanoue & Dodson, 2008*), but the distance from the posterior end of the ventral margin of the premaxilla to the maxillary tooth row is the widest in ZMNH M12414 and IVPP V12617 (*You & Xu, 2005*; *Bullar et al., 2019*).

Maxilla—Both maxillae are preserved (Fig. S5). In lateral view, the maxilla is subtriangular and contacts the jugal posteriorly, the premaxilla anteriorly and the lacrimal dorsally as in other *Psittacosaurus* species (*Sereno et al., 1988*; *Russell & Zhao, 1996*; *Zhou et al., 2006*; *Sereno et al., 2007*; *You, Tanoue & Dodson, 2008*; *Sereno, 2010*; *Sereno, Zhao & Tan, 2010*; *Napoli et al., 2019*; *Podlesnov et al., 2023*). There are nine alveoli for each maxilla, with seven on the left and nine on the right occupied by the erupted teeth (Figs. S5C–S5F). The position of the external naris is located above the anterior part of the maxilla as in *P. major* (CAGS-IG-VD-004; *You, Tanoue & Dodson, 2008*), *P. meileyingensis* (*Sereno et al., 1988*), and *P. amitabha* (*Napoli et al., 2019*) (Figs. 1A, 1B). The dorsal part of the maxilla is tall, trapeziform and reaches to the level with the ventral

border of the orbit (Figs. 1A, 1B) as in *Psittacosaurus meileyingensis* (*Sereno et al., 1988*), *P. neimongoliensis* (*Russell & Zhao, 1996*), *P. lujiatunensis* (*Zhou et al., 2006*), *P. major* (*Sereno et al., 2007*; *You, Tanoue & Dodson, 2008*), *P. sibiricus* (KOKM 22985/2; *Podlesnov et al., 2023*) and IVPP V12617 (*You & Xu, 2005*), but differing from the lower, triangular maxilla in all others. As in *P. lujiatunensis* (*Zhou et al., 2006*), *P. mongoliensis* (*Sereno, 2010*), *P. amitabha* (*Napoli et al., 2019*) and *P. gobiensis* (*Sereno, Zhao & Tan, 2010*), a sub-triangular maxillary fossa (Figs. S5A, S5B) is present on the lateral surface with a horizontal eminence located along its ventral margin. The fossa is anteroposteriorly longer than dorsoventrally deep and positioned anterior to the orbit as in *P. amitabha* (*Napoli et al., 2019*) and *P. major* (*Sereno et al., 2007*). *Sereno (2010)* describes the neurovascular foramina that open within or on the rim of the maxillary fossa and the anterolateral maxillary foramen located near or along the suture with the premaxilla in *Psittacosaurus*. The presence of these features is not confirmed in ZMNH M12414 due to limited resolution of CT images. Like most *Psittacosaurus* species (*Sereno et al., 1988*; *Zhou et al., 2006*; *Sereno et al., 2007*; *Sereno, Zhao & Tan, 2010*; *Sereno, 2010*), a maxillary protuberance is found posterior to the maxillary fossa at the posterior margin near the maxilla-jugal suture (Figs. S5A, S5B). In ventral view, the distance between the anterior ends of the tooth row is wide, as the ratio of the width across the anterior ends to that across the posterior ends is greater than 45% (Fig. 1F), as in *P. amitabha* (*Napoli et al., 2019*) and *P. lujiatunensis* (*Zhou et al., 2006*).

Lacrimal—Both lacrimals are preserved, with the right one being better preserved (Fig. S6). As in other *Psittacosaurus* species (*Sereno et al., 1988*; *Russell & Zhao, 1996*; *Zhou et al., 2006*; *Sereno et al., 2007*; *You, Tanoue & Dodson, 2008*; *Sereno, 2010*; *Sereno, Zhao & Tan, 2010*; *Napoli et al., 2019*; *Podlesnov et al., 2023*), the lacrimal forms the anterior margin of the orbit, and it is bounded by the premaxilla anteriorly, the prefrontal dorsally and the jugal ventrally with a relatively smaller trapezoid shape than other *Psittacosaurus* (Figs. 1A, 1B). It contributes to a small part of the anteroventral border of the orbit. The lacrimal foramen is located at the boundary between the lacrimal and the jugal (Fig. 1B, S6B), whereas the foramen is located along the orbital margin in other *Psittacosaurus* species (*Zhao, 1962*; *Sereno et al., 1988*; *Sereno, 1990*; *Russell & Zhao, 1996*; *Zhou et al., 2006*; *Sereno et al., 2007*; *Sereno, Zhao & Tan, 2010*; *Sereno, 2010*). This may be due to the partial breakage of the lateral surface of the right lacrimal, causing the lacrimal foramen to appear in a more anteroventral position. A right lacrimal canal continuous with the foramen can be seen in the CT image (Fig. S35), supporting the identification of the lacrimal foramen.

Nasal—Despite multiple fractures, the nasals retain their overall shape (Fig. S7). In dorsal view, the nasal is a long bone that flanks its counterpart in the anterior half including the rostroventral process, where it contacts the rostral bone ventrally (Figs. S7E, S7F). The posterior part of the internasal suture is unclear (Figs. S7E, S7F). The posterior part of the nasal expands laterally, reaching its maximum width where it meets the prefrontal. At its narrowest part, the main body of the nasal is about as wide as the prefrontal and it terminates posteriorly in contact with the frontals above the orbit

(Fig. 1E) as in other *Psittacosaurus* species (*Sereno et al., 1988*; *Russell & Zhao, 1996*; *Zhou et al., 2006*; *Sereno et al., 2007*; *Sereno, 2010*; *Sereno, Zhao & Tan, 2010*).

Frontal—Both frontals are preserved (Fig. S8). The frontals are flat and form a broad central element of the skull roof, constituting the posterodorsal rim of the orbit (Fig. 1E). In dorsal view, both frontals are fused into a single unit along the midline as the interfrontal suture cannot be observed (Figs. S8E, S8F). The frontals contact the nasals anteriorly, the prefrontals laterally, the postorbitals posterolaterally, and the parietals posteriorly. The ventral surface of the frontals has distinct depressions for the olfactory bulbs anteriorly and for the cerebral hemispheres posteriorly (Fig. S8F).

Prefrontal—Both prefrontals are preserved, but the right element exhibits some fractures (Fig. S9). They are narrow and meet the premaxillae and lacrimals ventrally, forming the anterodorsal corner of the orbit in left lateral view (Figs. 1A, 1B). In dorsal view, the prefrontal contacts the nasal anteromedially and the frontal posteriorly (Fig. 1E). The maximum width of the prefrontal is nearly as wide as that of the nasal (Fig. 1E) as commonly seen in other *Psittacosaurus* species (*Sereno et al., 1988*; *Russell & Zhao, 1996*; *Sereno et al., 2007*; *You, Tanoue & Dodson, 2008*; *Sereno, 2010*; *Sereno, Zhao & Tan, 2010*).

Parietal—Both parietals are preserved (Fig. S10); however, a fracture on the right side near the parietal frill (Fig. S10E) resulted in the ventral distortion of the right supratemporal fenestra (Fig. 1D). The parietals form the posterior-most element of the central skull roof and fuse along the midline to form a low sagittal crest (Figs. S10A, S10B and S10E), as in other *Psittacosaurus* (*Sereno et al., 1988*; *Averianov et al., 2006*; *Zhou et al., 2006*; *Sereno et al., 2007*; *You, Tanoue & Dodson, 2008*; *Sereno, 2010*; *Sereno, Zhao & Tan, 2010*; *Napoli et al., 2019*; *Podlesnov et al., 2023*). The parietal contacts the frontal and the postorbital anteriorly, the laterosphenoid anteroventrally, the squamosal laterally and the supraoccipital ventrally. In dorsal view, the posterior margin of the parietal is almost linear and runs perpendicular to the sagittal crest (Fig. S10E) as in IVPP V12617 (*Bullar et al., 2019*) and in *Psittacosaurus meileyingensis* (*Sereno et al., 1988*), while all other species have a distinct incised margin at the middle of the posterior end of parietals. In *P. meileyingensis* (*Sereno et al., 1988*), the lateral process of the parietal shows a distinctive slope angled posterodorsally, whereas the parietal broadly contacts the squamosal in IVPP V12617 (*You & Xu, 2005*) and ZMNH M12414 (Fig. 1E).

Postorbital—Both postorbitals are preserved (Fig. S11). The postorbital is a three-pronged element composed of the bar on the skull roof, the temporal bar and the jugal bar that separates the orbit from the infratemporal fenestra (Figs. S11A–S11D). The left postorbital is present in its original position, whereas the right postorbital is fractured at the center of its triradiate form, and the temporal bar is disarticulated from the squamosal (Figs. S11B, S11D). This results in a slight ventral distortion of the right supratemporal fenestra along with a fracture in the parietals, as seen in posterior view (Fig. 1D). The jugal bar is thick and relatively long, compared with that of all other *Psittacosaurus* species, overlapping the jugal and forming most of the posterior border of the orbit (Figs. 1A, 1B). The jugal bar is the longest prong in the postorbital and is nearly twice as long as the temporal bar (Figs. S11A–S11D). This condition appears to be an

autapomorphy of *P. houi*, shared with the holotype (IVPP V12704; *You, Xu & Wang, 2003*) and another referred specimen (IVPP V12617; *You & Xu, 2005*). In contrast, the length of the jugal bar is shorter than or nearly equal to that of the temporal bar in other *Psittacosaurus* species (*Russell & Zhao, 1996*; *Averianov et al., 2006*; *Zhou et al., 2006*; *Sereno et al., 2007*; *You, Tanoue & Dodson, 2008*; *Sereno, 2010*; *Sereno, Zhao & Tan, 2010*; *Napoli et al., 2019*). The distal-most extent of the left jugal bar is missing (Figs. S11A, S11B, S11I and S11K). The jugal bar is expanded anteroventrally (Figs. S11A–S11D) as in *Psittacosaurus lujiatunensis* (*Zhou et al., 2006*) and *P. meileyingensis* (*Sereno et al., 1988*), differing from the narrow tip in all others. The temporal bar of the postorbital laterally overlaps the postorbital bar of the squamosal and forms the entire dorsal border of the infratemporal fenestra in lateral view (Fig. 1A). The bar on the skull roof is shorter than the temporal and jugal bars (Figs. S11A–S11D), as in other *Psittacosaurus* species (*Russell & Zhao, 1996*; *Zhou et al., 2006*; *Averianov et al., 2006*; *Sereno et al., 2007*; *You, Tanoue & Dodson, 2008*; *Sereno, 2010*; *Sereno, Zhao & Tan, 2010*; *Napoli et al., 2019*). In lateral view, the bar on the skull roof extends along the margin of the orbit, contributing to form the posterodorsal rim of the orbit (Figs. 1A, 1B) as in other *Psittacosaurus* species (*Russell & Zhao, 1996*; *Zhou et al., 2006*; *Averianov et al., 2006*; *Sereno et al., 2007*; *You, Tanoue & Dodson, 2008*; *Sereno, 2010*; *Sereno, Zhao & Tan, 2010*; *Napoli et al., 2019*; *Podlesnov et al., 2023*).

Squamosal—Both squamosals are preserved (Fig. S12). They are nearly complete except for the missing anterior ramus on the right squamosal (Figs S12B, S12D, S12F, S12H, S12J and S12L). The squamosal is a tetraradiate bone as in other *Psittacosaurus* species (*Zhou et al., 2006*; *Sereno et al., 2007*; *You, Tanoue & Dodson, 2008*; *Sereno, 2010*; *Sereno, Zhao & Tan, 2010*; *Napoli et al., 2019*) and is located at the upper posterolateral corner of the skull. Anteriorly, the squamosal contacts the postorbital to form a bar separating the upper and lower temporal fenestrae. The anterior ramus of the right squamosal is missing due to a local fracture (Figs. S12B, S12H), along with the cracked postorbital, both of which likely contributed to the ventral distortion of the right supratemporal fenestra. The anterior ramus of the left squamosal is completely preserved but short in dorsal view (Fig. 1E), as in *Psittacosaurus lujiatunensis* (*Zhou et al., 2006*), *P. major* (CAGS-IG-VD-004; *You, Tanoue & Dodson, 2008*) and *P. amitabha* (*Napoli et al., 2019*). This contrasts with most other *Psittacosaurus* species, in which the anterior-most end of the ramus reaches the anterior margin of the supratemporal fenestra (*Russell & Zhao, 1996*; *Sereno et al., 2007*; *Sereno, 2010*; *Sereno, Zhao & Tan, 2010*). The ventral ramus is relatively short and does not contact the quadratojugal (Figs. 1A–1C). Although *Hedrick & Dodson (2013)* argue that the ventral ramus of the squamosal and the dorsal ramus of the quadratojugal are almost always broken in *P. lujiatunensis*, *P. major*, and *Hongshanosaurus houi* (=*Psittacosaurus houi*, IVPP V12617), our CT images and rendered model show no evidence of such damages. The medial ramus projects medially to meet the parietal, with which it forms the posterior margin of the supratemporal fenestra (Figs. 1E, S12G, S12H). Further ventrally, the ventromedial ramus of the squamosal contacts the exoccipital (Figs. S12I, S12J).

Jugal—Both jugals are present (Fig. S13). The jugal is large and consists of infraorbital, infratemporal, and dorsal rami, and the jugal horn (Figs. 1A, 1B). The anteromedially

curved infraorbital ramus, the dorsal ramus, and the posteromedially sloped infratemporal ramus are plate-shaped. The ventral surface of the jugal is flat (Figs. S13E, S13F). A gentle ridge runs posterolaterally from the apex of the dorsal process to the lateral end of the jugal horn, dividing the lateral aspect of the jugal into anterior and posterior surfaces. In lateral view, the area of the anterior surface is much larger than that of the posterior surface, as in *Psittacosaurus major* (*Sereno et al., 2007*; *You, Tanoue & Dodson, 2008*) and *P. sibiricus* (*Podlesnov et al., 2023*), which is thought to be due to the shorter infratemporal ramus than in other *Psittacosaurus* (*Russell & Zhao, 1996*; *Sereno, 2010*; *Sereno, Zhao & Tan, 2010*; *Napoli et al., 2019*) (Figs. S13A, S13B). Additionally, ZMNH M12414 exhibits a relatively larger anterior proportion of the jugal than IVPP V12704 (holotype; *You, Xu & Wang, 2003*). The infraorbital ramus of the jugal is deeper dorsoventrally than the infratemporal ramus, leaving the ventral margin of the orbit slightly higher than the ventral margin of the infratemporal fenestra (Figs. S13A–S13D). The anterior surface of the jugal is smooth and slightly concave in the middle as in *P. lujiatunensis* (*Zhou et al., 2006*). The dorsal process of the jugal is overlapped by the ventral ramus of the postorbital, forming the bar between the orbit and infratemporal fenestra (Figs. S13A, S13B). The axes of the dorsal process of the jugal and maxillary tooth row are oriented at an angle of approximately 135 degrees in lateral view (Figs. 1A, 1B), resulting in the lateral surface of the jugal being gently inclined, in contrast to the more acute angle observed in all other species. As a result, the surface area of the jugal is very large in dorsal view (Fig. 1E), and this feature is only found in ZMNH M12414 and IVPP V12617 (*You & Xu, 2005*; *Bullar et al., 2019*) among *Psittacosaurus* species. The infraorbital ramus forms the ventral margin of the orbit and anteriorly contacts the lacrimal and the maxilla (Figs. 1A, 1B). The infratemporal ramus bifurcates posteriorly as in other species of *Psittacosaurus* (*Sereno et al., 1988*; *Sereno & Chao, 1988*; *Russell & Zhao, 1996*; *Zhou et al., 2006*; *Averianov et al., 2006*; *Sereno et al., 2007*; *You, Tanoue & Dodson, 2008*; *Sereno, 2010*; *Sereno, Zhao & Tan, 2010*; *Napoli et al., 2019*; *Podlesnov et al., 2023*), although this feature is not seen in the incomplete left jugal (Figs. S13D, S13L). The dorsal part of the bifurcated infratemporal ramus is large and overlaps the quadratojugal, but never reaches the quadrate (Fig. 1B). The infratemporal ramus forms the anteroventral and ventral margins of the infratemporal fenestra, without expanding posterodorsally to form the posteroventral margin (Figs. 1A, 1B) as in most other *Psittacosaurus* species (*Sereno et al., 1988*; *Sereno & Chao, 1988*; *Russell & Zhao, 1996*; *Zhou et al., 2006*; *Averianov et al., 2006*; *Sereno, 2010*; *Sereno, Zhao & Tan, 2010*; *Napoli et al., 2019*; *Podlesnov et al., 2023*). The stout jugal horn protrudes laterally, forming the well-developed horn that is sub-triangular in dorsal view. The sagittal section of the horn is also sub-triangular. The ventral surface of this horn is flat and lies more dorsal than the level of the maxillary tooth row (Figs. 1A, 1B, S13E, S13F).

Quadratojugal—Both quadratojugals are preserved, while the left one is partly fractured (Fig. S14). The quadratojugal is subtriangular in lateral view (Fig. S14B) and mediolaterally flattened in posterior view (Fig. S14L). The posterior margin of the quadratojugal extensively overlaps the quadrate, while the quadrate condyle is exposed in lateral view (Figs. 1A–1C, 1E) as in *Psittacosaurus lujiatunensis* (*Zhou et al., 2006*) and *P. gobiensis* (*Sereno, Zhao & Tan, 2010*). As in most other species of *Psittacosaurus* (*Sereno et al., 1988*;

*Sereno & Chao, 1988*; *Russell & Zhao, 1996*; *Sereno et al., 2007*; *You, Tanoue & Dodson, 2008*; *Sereno, 2010*; *Sereno, Zhao & Tan, 2010*), the dorsal process of the quadratojugal does not contact the ventral process of the squamosal, only slightly contributing to the posterior rim of the infratemporal fenestra (Figs. 1A–1C, 1E). In *P. meileyingensis* (*Sereno et al., 1988*), *P. sinensis* (BNHM BPV149; *Sereno, 2010*), and IVPP V12617 (*Bullar et al., 2019*), the quadratojugal prominence is present in the ventral part, whereas only a weak eminence (bump) is observed in the right quadratojugal of ZMNH M12414 (Figs. S14B, S14F). The ventral part of the quadratojugal extends anteriorly to the level of the posterior margin of the laterotemporal fenestra (Figs. 1A, 1B) as in *P. sinensis* (IVPP V738; *Sereno, 2010*), *P. neimongoliensis* (*Russell & Zhao, 1996*), *P. mongoliensis* (*Sereno, 2010*), *P. sibiricus* (*Averianov et al., 2006*; *Podlesnov et al., 2023*), *P. gobiensis* (*Sereno, Zhao & Tan, 2010*), *P. major* (CAGS-IG-VD-004; *You, Tanoue & Dodson, 2008*), *P. lujiatunensis* (*Zhou et al., 2006*) and IVPP V12617 (*You & Xu, 2005*; *Taylor et al., 2017*).

Quadrate—Both quadrates are preserved (Fig. S15). As in other *Psittacosaurus* species (*Russell & Zhao, 1996*; *Zhou et al., 2006*; *Sereno et al., 2007*; *Averianov et al., 2006*; *Sereno, 2010*; *Podlesnov et al., 2023*), the quadrate shaft is oriented anterolaterally in dorsal view (Figs. 2E, S15G, S15H) and the condyle expands transversely to form a broad articular surface in ventral view (Figs. 1F, S15E, S15F). The shaft is slightly arched along its posterior margin (Figs. S15A, S15B) as in *Psittacosaurus mongoliensis* (*Sereno, 2010*), *P. sinensis* (IVPP V738; *Sereno, 2010*), *P. neimongoliensis* (*Russell & Zhao, 1996*) and IVPP V12617 (*You & Xu, 2005*; *Bullar et al., 2019*), but unlike the strongly concave condition of *P. sinensis* (BNHM BPV149; *Sereno, 2010*), *P. meileyingensis* (*Sereno et al., 1988*) and *P. lujiatunensis* (*Sereno et al., 1988*). The quadrate shaft is oriented at an angle of approximately 45 degrees in lateral view, as is the postorbital jugal bar (Figs. 1A, 1B), a feature shared with ZMNH M12414, IVPP V12617 (*You & Xu, 2005*; *Bullar et al., 2019*) and *P. amitabha* (*Napoli et al., 2019*). Dorsal to the condyles, the quadrate is not exposed in lateral view just posterior to the quadratojugal-quadrate suture (Figs. 1A, 1B), as in *P. mongoliensis* (*Sereno, 2010*), *P. sinensis* (*Sereno, 2010*) and *P. meileyingensis* (*Sereno et al., 1988*), also in *P. major* (*Sereno et al., 2007*; *You, Tanoue & Dodson, 2008*) and *P. lujiatunensis* (*Zhou et al., 2006*) to a lesser degree. The pterygoid wing developed as a typically broad and thin bone and completes the medial wall of the laterotemporal fossa as far anterior as the main body of the postorbital (Figs. S15G–S15L). This medial wall obscures the ventral portions of the braincase in lateral view (Figs. 1A, 1B), including most of the cranial nerve foramina.

Palate—The palate consists of premaxillae, maxillae, vomers, palatines, pterygoid and ectopterygoid (Figs. 1F, S4, S5 and S16–S19). The short secondary palate is formed by the palatal processes of premaxillae and maxillae and is well-exposed anteriorly in ventral view (Fig. 1F). The choanae (internal nostrils) are bordered anteriorly and laterally by the maxillae, medially by the vomers, posteriorly by the palatines and posteromedially by the pterygoids (Fig. 1F). In ZMNH M12414, the choanae are fully exposed in ventral view as large elliptical openings into the oral cavity, which is one of the major differences in the palate between psittacosaurids and basal neoceratopsians (*Dodson, You & Tanoue, 2010*). Anterior to the internal naris, the rostral, maxillae and premaxillae form a gentle vault

(arched cavity), and the incisive foramen is bounded by the same three elements. The internal naris extends anteriorly to the level of the anterior edge of maxillary tooth row (Fig. 1F) as in other species of *Psittacosaurus* (*Zhou et al., 2006*).

Palatine—Both palatines are well-preserved in ZMNH M12414 (Fig. S17). The palatine contacts the maxilla and the jugal laterally, the pterygoid posteriorly, and the vomer medially. In some *Psittacosaurus* species, the palatine does not extend dorsally enough to contact the vomers, which are instead embraced by the pterygoids (*Dodson, You & Tanoue, 2010*; *Podlesnov et al., 2023*). In contrast, the palatine contacts the vomer in ZMNH M12414 (Fig. S17). However, it should be noted that the exact point of contact is not discernible. Whether this is due to the CT image resolution or obliterated sutures remains unclear.

Vomer—The long, fused vomer is a paired bone located along the midline of the skull, contacting the maxillae anteriorly and the pterygoids posteriorly and forming the medial margins of the choanae (Fig. 1F). The bone consists of a plate-like posterior vertical part and a rod-like, slightly arched vomerine bar (Figs. S16A, S16B). The anterior end of the vomer does not contribute to the formation of the secondary palate but lies dorsally, in a subnarial position (Figs. 1F, S16). This feature is one of the major differences in palate between psittacosaurids and basal neoceratopsians (*Dodson, You & Tanoue, 2010*). The vomer is in contact with the palatine laterally and the pterygoid posteriorly (Figs. S16C, S16D).

Pterygoid—Both pterygoids are preserved, and a well-developed pterygoid is triradiate, with a posterolaterally directed quadrate ramus, a posteroventrally directed mandibular ramus and an anterodorsally extending palatal ramus (Fig. S18). The quadrate ramus is Y-shaped and the largest part of the pterygoid. The quadrate ramus is thin and broadly meets the pterygoid ramus of the quadrate and the prootic posterolaterally (Figs. S18A–S18D). Medially, there is a large cup-like facet for the basipterygoid process of the basisphenoid, which is adjacent to the dorsal margin of the quadrate ramus (Figs. S18G, S18H). The pterygoid forms the posterior half of the palate. The mandibular ramus is one of the diagnostic characters of the genus *Psittacosaurus* (*Sereno, 2010*), where the ventral-most extent is elongated to form the mandibular process (*Sereno, 2010*). In ZMNH M12414, the mandibular process is short and mediolaterally broad, which can be seen from the lateral and ventral views (Figs. S18A–S18D). The mandibular ramus is in contact with the maxilla anterolaterally (Figs. S18A–S18B). The palatal ramus is larger than the mandibular ramus and contacts the vomers and the palatine anteriorly (Figs. S18A, S18B). As in other *Psittacosaurus* (*Zhou et al., 2006*; *You, Tanoue & Dodson, 2008*; *Sereno, 2010*; *Podlesnov et al., 2023*), the anterior end of the pterygoid (Figs. S18E, S18F) forms a very short median joint between the left and right pterygoids, which is located anteriorly, at approximately the midpoint of the skull (Fig. 1F). This feature is noted as one of the major differences in the palate between psittacosaurids and basal neoceratopsians (*Dodson, You & Tanoue, 2010*).

Ectopterygoid—Both ectopterygoids are well-preserved in ZMNH M12414 (Fig. S19). The outline of the ectopterygoid is a tall triangle in dorsal and ventral view, and it contacts the jugal anterolaterally and the maxilla ventrally (Figs. S19G–S19L). The ectopterygoid is

separated from the palatine by the maxilla as in *Psittacosaurus sinensis* (*Sereno, 1987*) and *P. neimongoliensis* (*Russell & Zhao, 1996*), while the bone meets the palatine in *P. lujiatunensis* (*Zhou et al., 2006*) and *P. mongoliensis* (*Russell & Zhao, 1996*). The maxillary articular surface is flat (Figs. S19E, S19F), while the dorsal surface forms a gentle ridge (Figs. S19C, S19D).

Braincase—The braincase of ZMNH M12414 includes a supraoccipital, partly fractured exoccipitals, basioccipitals, opisthotics, laterosphenoids, a basisphenoid and prootics (Fig. S20). In posterior view, the foramen magnum has a diameter of approximately 18.3 mm, bordered by the supraoccipital dorsally, the exoccipitals laterally and the basioccipital ventrally (Fig. S20D). The supraoccipital is dorsoventrally deep and subtriangular in posterior view (Fig. S20) unlike the diamond-shaped ones in other *Psittacosaurus* species (*Sereno et al., 1988*; *Napoli et al., 2019*; *Podlesnov et al., 2023*). Similarly, IVPP V12617 seems to have a tall subtriangular-shaped supraoccipital (*You & Xu, 2005*; *Bullar et al., 2019*), although *Bullar et al. (2019)* notes that the fused and obscured sutures between the supraoccipital and the parietal or the exoccipital make the exact location of the contact difficult to determine. Because the supraoccipital of ZMNH M12414 is undeformed and there is no evidence of fracture, we conclude that this character is an autapomorphy of ZMNH M12414 and IVPP V12617. The supraoccipital contributes only to the middle portion of the dorsal border of the foramen magnum and is covered dorsally by the parietals. The exoccipitals borders the foramen magnum laterally. The exoccipital and opisthotic are fused to form the paroccipital process, which extends posterolaterally but lacks the distal end (Fig. S22). As in other *Psittacosaurus* (*Zhou et al., 2006*; *You, Tanoue & Dodson, 2008*; *Sereno, 2010*; *Bullar et al., 2019*; *Podlesnov et al., 2023*), the basal tubera are present as paired processes projecting ventral to the occipital condyle (Fig. S23). The basioccipital borders the foramen magnum ventrally (Fig. S20D) and forms the occipital condyle that is directed posteriorly and slightly ventrally (Fig. S23) as in other *Psittacosaurus* species (*Zhou et al., 2006*; *You, Tanoue & Dodson, 2008*; *Sereno, 2010*; *Bullar et al., 2019*; *Podlesnov et al., 2023*). The condyle is about 12.5 mm wide in diameter with a smooth hemisphere. The basioccipital fuses with the basisphenoid anteroventrally and exoccipital-opisthotic dorsolaterally. The latter suture is almost completely fused and is difficult to discern. There is no foramen or fossa along the midline between the basal tubera and the occipital condyle (Fig. S20F), unlike those in *Psittacosaurus amitabha* (*Napoli et al., 2019*) and *P. lujiatunensis* (*Zhou et al., 2006*). The laterosphenoid contacts the prootic posteroventrally and the frontal anterodorsally (Fig. S24) as in other *Psittacosaurus* species (*Zhou et al., 2006*; *You, Tanoue & Dodson, 2008*; *Sereno, 2010*; *Napoli et al., 2019*; *Podlesnov et al., 2023*). The trigeminal foramen (CN V) is 5 mm in diameter and bounded by the laterosphenoid, the prootic and the basisphenoid, which is visible on both lateral sides of the bones (Figs. S20A–S20C). The basisphenoid is well-preserved and in contact with the prootic dorsally, the pterygoid anteriorly and the basioccipital posteriorly (Fig. S25). In *Hedrick & Dodson (2013)*, a separation of the braincase from the palate was shown as the evidence for dorsoventral compressive deformation. However, in ZMNH M12414, the braincase and palate are in contact (Fig. S25). The basal tubera has a subcircular and subvertical posterior surface, with a

round, rough ventral margin that is located slightly ventral to the occipital condyle (Figs. S20D, S20F, S23). The basipterygoid processes are elongated to reach 14.4 mm in length, which is subequal to the length of the basisphenoid body measured from the notch between the processes to the basal tubera (13.5 mm), as in *P. major* (*Sereno et al., 2007*; *Sereno, 2010*). Both prootics are preserved, although their boundaries to the exoccipital and basisphenoid are obscured (Fig. S26). The prootic forms the lateral wall of the braincase and appears to have a contact with the laterosphenoid dorsally, the parietal posterodorsally, the pterygoid ventrally and the exoccipital posteriorly. The shape of the prootic is sub-rectangular in lateral view (Figs. S26A, S26B).

Mandible—The mandible lacks the predentary and is composed of the following elements: left and right dentaries, left and right surangulars, left and right angulars, left and right splenials, left and right coronoids, left and right prearticulars and left and right articulars (Figs. 2, S27–S33). In lateral view, the mandible is strongly curved dorsally along its ventral margin, with the dentaries and angulars forming the anterior and posterior ends of the arc, respectively (Figs. 2A, 2B). The mandible is dorsoventrally deep relative to the anteroposterior length as in *Psittacosaurus lujiatunensis* (*Zhou et al., 2006*), *P. major* (*Sereno et al., 2007*; *You, Tanoue & Dodson, 2008*) and IVPP V12617 (*You & Xu, 2005*; *Landi et al., 2021*). Posteriorly, the articular and surangular form the retroarticular process (Figs. 2A–2D) as in other *Psittacosaurus* species (*Sereno et al., 1988*; *Russell & Zhao, 1996*; *Zhou et al., 2006*; *Averianov et al., 2006*; *Sereno et al., 2007*; *You, Tanoue & Dodson, 2008*; *Sereno, 2010*; *Sereno, Zhao & Tan, 2010*; *Podlesnov et al., 2023*). The height of the retroarticular process is approximately at the same level as the dentary tooth row (Fig. 2C) as in IVPP V12617 (*You & Xu, 2005*; *Landi et al., 2021*) and *P. neimongoliensis* (*Russell & Zhao, 1996*). In contrast to other *Psittacosaurus* species (*Sereno et al., 1988*; *Russell & Zhao, 1996*; *Zhou et al., 2006*; *Averianov et al., 2006*; *Sereno et al., 2007*; *You, Tanoue & Dodson, 2008*; *Sereno, 2010*; *Sereno, Zhao & Tan, 2010*; *Podlesnov et al., 2023*), the retroarticular process is reduced so that the mandibles appear dorsoventrally deep in lateral view (Figs. 2A–2D). The dentary is the largest bone of the mandible, extending for about half of its length (Figs. 2A–2D). The dentary contacts the prearticular and coronoid medially and the surangular and angular posteriorly. The dentary is convex lingually (Figs. S27E–S27H), causing the toothrow to be slightly convex medially in dorsal view (Figs. S27G and S27H). There are nine alveoli in the dentary, which are fully occupied by the erupted teeth on the left, while only six alveoli are occupied on the right (Figs. S27A–S27D, S27G and S27H). The ventral border of the dentary is strongly curved and possesses a prominent dentary flange at its posteroventral corner, as in *P. meileyingensis* (*Sereno et al., 1988*), *P. lujiatunensis* (*Zhou et al., 2006*) and *P. major* (*Sereno et al., 2007*; *You, Tanoue & Dodson, 2008*). Although this feature is particularly evident in the right dentary, the left dentary flange is poorly preserved (Figs. S27A, S27B). The surangular forms the posterodorsal portion of the lower jaw in lateral view (Figs. 2A, 2B), and contacts the dentary anteriorly with forming the posterior half of the coronoid process (Fig. S28). The surangular extends posteroventrally to the end of the mandible, covering the angular underneath. The posterior portions of the right surangular and angular are fractured (Figs. 2B, 2D), and the right mandibular condyle is somewhat tilted posterodorsally. In addition,

the distortion is indicated by the asymmetry between the height of the left and right mandibular condyles in posterior view (Fig. 2F). The obliteration of the sutural boundaries between the right surangular and articular, which led to the incomplete restoration of these bones in the CT images (Figs. S28, S29), may be a result of these deformations. Although the central sutural boundary between the dentary, surangular and angular is fenestrated in *P. meileyingensis* (*Sereno et al., 1988*), *P. mongoliensis* (*Sereno, 2010*) and *P. sinensis* (*Sereno, 2010*), forming the external mandibular fenestra, its presence remains unclear in ZMNH M12414. The articular is mediolaterally broad, but dorsoventrally thin where it articulates with the robust quadrate (Fig. S29). The articular contacts the prearticular anteriorly and the surangular and angular laterally, forming the retroarticular process. In lateral view, the anteroposterior length of the articular is relatively shorter than those of other *Psittacosaurus* species including *P. lujiatunensis* (*Zhou et al., 2006*), *P. major* (*You, Tanoue & Dodson, 2008*), *P. sibiricus* (*Podlesnov et al., 2023*) and IVPP V12617 (*Landi et al., 2021*). This difference appears to be due to post-depositional deformation or intraspecific variation in *P. houi* (see Discussion). The angular is well-exposed laterally with a sheet-like process that covers the ventral margin of the mandible. The angular contacts the dentary anteriorly, the surangular dorsally, and the splenial, prearticular and articular medially (Fig. S30). The posterior part of the right angular has some fractures, likely contributing to the medial displacement of the posterior part (Figs. S30B, S30D, S30F, S30H, S30J and S30L). In medial view, the angular is medially covered by the splenial anteriorly and the prearticular dorsally (Figs. 2C, 2D). The ventral margin of the angular is sinusoidal in lateral and medial views as in *P. meileyingensis* (*Sereno et al., 1988*), *P. lujiatunensis* (*Zhou et al., 2006*), *P. major* (*Sereno et al., 2007*; *You, Tanoue & Dodson, 2008*), *P. sinensis* (IVPP V738; *Sereno, 2010*), *P. gobiensis* (*Sereno, Zhao & Tan, 2010*), *P. sibiricus* (*Podlesnov et al., 2023*), and IVPP V12617 (*Landi et al., 2021*). The splenial is a thin plate-like bone on the medial side of the mandible. The splenial covers the dentary anteriorly and the angular posteriorly, contacting the prearticular posterodorsally and coronoid dorsally (Fig. S31). The anterior process of the left splenial is fractured and failed to reach the anterior-most part of the dentary, while the process of the right extends to cover the ventromedial margin of the bone (Figs. S31C, S31D). The depth of the splenial seems to be short in ZMNH M12414, while the depth is tall in *P. lujiatunensis* (*Zhou et al., 2006*), *P. major* (*You, Tanoue & Dodson, 2008*), *P. sibiricus* (*Podlesnov et al., 2023*), and IVPP V12617 (*Landi et al., 2021*). However, the splenials of ZMNH M12424 are asymmetrical (Fig. S31), suggesting that the thin plate-like bones may have been fractured and the splenial was not fully preserved. Both coronoids are preserved, but the left one is better preserved (Fig. S32). The coronoid process is a small thin plate, almost rhombic in shape, attaching by the dorsal part of its medial side to the coronoid process of the dentary (Figs. 2C, 2D). The position of the coronoid process is on the main axis of the dentary in dorsal view and posterior to the tooth row in lateral view as in *P. major* (*You, Tanoue & Dodson, 2008*) and *P. sibiricus* (*Podlesnov et al., 2023*). This condition differs from IVPP V12617 (*Landi et al., 2021*) but it appears to be influenced by post-depositional deformation or intraspecific variation in *P. houi* (see Discussion). The prearticular is more complete on the left side, while the right side is missing the anterior process

(Figs. S33A, S33B). The prearticular is a thin, strip-like bone that curves ventrally, extending along the medial side of the lower jaw. The prearticular contacts the splenial and the angular ventrally, the splenial anteriorly, and the coronoid dorsally (Fig. S33). The articular and splenial form the internal mandibular fenestra on the medial side of the lower jaw in the left mandible (Fig. 2C) as seen in *P. lujiatunensis* (*Zhou et al., 2006*), *P. major* (*You, Tanoue & Dodson, 2008*) and *P. sibiricus* (*Podlesnov et al., 2023*). However, due to the partially fractured splenial and prearticular, which fail to form the complete rim of the fenestra, it remains unclear whether the fenestra observed in ZMNH M12414 is homologous to those of other species.

## Phylogenetic analysis

The analysis yielded 36,696 MPTs with a length of 1,317 steps, a consistency index (CI) of 0.318 and a retention index (RI) of 0.680 (Fig. S36). In the strict consensus topology, *Psittacosaurus houi* is recovered as the latest-diverging species of *Psittacosaurus* together with *P. major* (Fig. S37). In addition, *P. houi* is found within a clade comprising the genus *Psittacosaurus*.

According to the strict consensus topology of the present analysis, the monophyly of *Psittacosaurus* is supported by 14 unambiguous synapomorphies, and *Psittacosaurus houi* presents 13 of them:

1. 10(1) posterolateral process of the premaxilla in contact with lacrimal, interfering with maxilla-nasal contact (convergent with some heterodontosaurids, *Jeholosaurus*, some iguanodontians and some neoceratopsians);

2. 11(1) premaxilla-lacrimal contact wide, and almost the entire anterior edge of the lacrimal contacting the premaxilla (convergent with some heterodontosaurids, *Jeholosaurus*, some iguanodontians and some neoceratopsians);

3. 17(1) premaxilla-prefontal contact present (convergent with *Heterodontosaurus*, some iguanodontians and *Liaoceratops*);

4. 19(2) premaxilla fully edentulous;

5. 23(2) ventral border of external nares located significantly above that of the infratemporal fenestra (convergent with *Herrerasaurus* and *Eoraptor*);

6. 24(1) dorsoventral depth of snout at external nares large and about or more than 60% relative to that of orbital region (convergent with *Herrerasaurus*, *Huayangosaurus*, *Chaoyangsaurus* and some neoceratopsians);

7. 26(0) deep elliptical fossa along sutural line of nasals absent (convergent with *Silesaurus*, *Eoraptor*, *Herrerasaurus*, *Lesothosaurus*, *Hypsilophodon*, some thyreophorans, *Thescelosaurus*, *Parksosaurus*, some pachycephalosaurians, some iguanodontians and some neoceratopsians).

8. 28(1) location of anterior end of nasal (internarial bar) far anterior to external naris (convergent with *Mosaiceratops*).

9. 32(0) external antorbital fenestra absent (convergent with *Herrerasaurus*, some thyreophorans and some iguanodontians);

10. 73(1) jugal process of postorbital extending to ventral margin of orbit (convergent with *Mosaiceratops*);

11. 88(1) anterior margin above quadrate wing transversely expanded, rounded and thickened (convergent with *Hualianceratops*);

12. 156(0) anterior end of the predentary rounded in dorsal view (convergent with *Chaoyangsaurus*, some iguanodontians, *Pinacosaurus* and some thyreophorans);

13. 173(1) minimum height greater than 50% of total length of dentary in lateral view (convergent with *Goyocephale*, *Hualianceratops* and some neoceratopsians).

*Psittacosaurus houi* also presents three unambiguous synapomorphies for *Psittacosaurus* diverging later than *P. sinensis*:

1. 56(1) jugal ridge divides the lateral surface of into two (convergent with *P. mongoliensis*, *P. major*, *P. gobiensis*, *P. lujiatunensis*, *P. sibiricus* and some neoceratopsians);

2. 164(1) first dentary tooth separated from predentary by short diastema (convergent with *Echinodon*, *Chaungchunsaurus*, *Haya*, *Thescelosaurus*, *Parksosaurus*, some iguanodontians, some thyreophorans, *P. mongoliensis*, *P. major*, *P. gobiensis*, *P. sibiricus* and some neoceratopsians);

3. 184(1) surangular length more than 50% of mandibular length (convergent with *P. mongoliensis*, *P. gobiensis*, *P. lujiatunensis*, *P. sibiricus* and some chaoyangsaurids).

*Psittacosaurus houi* also presents three unambiguous synapomorphies for *Psittacosaurus* diverging later than *P. sibiricus*:

1. 33(1) antorbital fossa present (convergent with *P. amitabha*, *P. mongoliensis*, *P. major*, *P. gobiensis* and *P. lujiatunensis*);

2. 84(1) lateral ramus of quadrate absent (convergent with some thyreophorans, *P. amitabha*, *P. major*, *P. gobiensis* and *P. lujiatunensis*);

3. 139(1) pterygoid-maxilla contact at posterior end of tooth row present (convergent with *P. amitabha*, *P. mongoliensis*, *P. major*, *P. gobiensis*, *P. lujiatunensis* and some neoceratopsians).

*Psittacosaurus houi* also presents two unambiguous synapomorphies for *Psittacosaurus* diverging later than *P. lujiatunensis* and *P. gobiensis*:

1. 102(1) anteroventral corner of infratemporal fenestra forms an acute angle (convergent with *Silesaurus*, *Eoraptor*, some heterodonsaurids, *Agilisaurus*, *Haya*, *Changchunsaurus*, *Hypsilophodon*, *Jeholosaurus*, *Orodromeus*, *Parksosaurus*, *Zephyrosaurus*, *Thescelosaurus*, *Gasparinisaura*, some pachycephalosaurians, some iguanodontians, *P. amitabha* and *P. mongoliensis*);

2. 126(0) ratio of maximum occipital width to maximum occipital height > 1.1 (convergent with some thyreophorans, *Ouranosaurus*, *P. amitabha* and *P. mongoliensis*);

*Psittacosaurus houi* also presents three unambiguous synapomorphies for *Psittacosaurus* diverging later than *P. amitabha*:

1. 12(2) ventral (oral) margin of premaxilla raised above maxillary tooth row (convergent with *P. mongoliensis*, *P. major*, *P. sinensis* and some neoceratopsians);
2. 61(1) infratemporal process of jugal strongly arched laterally (convergent with *P. mongoliensis*, *P. major*, some chaoyangsaurids and *Mosaiceratops*);
3. 82(1) quadratojugal facing posterolaterally (convergent with some thyreophorans, *P. mongoliensis*, *P. gobiensis* and some neoceratopsians).

*Psittacosaurus houi* also presents four unambiguous synapomorphies for *Psittacosaurus* diverging later than *P. mongoliensis*:

1. 37(2) ventral margin of antorbital fossa poorly delineated (convergent with *P. major*);
2. 93(0) medial quadrate condyle subequal to lateral condyle in size (convergent with *Agilisaurus*, *Haya*, *Changchunsaurus*, *Hypsilophodon*, *Jeholosaurus*, *Zephyrosaurus*, *Parksosaurus*, some pachycephalosaurians, some iguanodontians, *P. major*, *P. gobiensis*, *P. sinensis* and *Hualianceratops*);
3. 94(1) frontal involved in supratemporal fenestra (convergent with some *Herrerasaurus*, *Heterodontosaurus*, *Lesothosaurus*, some thyreophorans, *Ouranosaurus*, *P. major*, *P. sinensis*, *Yinlong*, and some neoceratopsians);
4. 167(1) anterior end of dentary tooth row (and edentulous anterior portion) downturned in lateral view (convergent with *Pegomastax*, some thyreophorans, some iguanodontians, *P. major*, *P. gobiensis* and *P. sibiricus*).

*Psittacosaurus houi* exhibits five diagnostic features. Among them, one diagnostic feature is autapomorphic within *Psittacosaurus*:

1. 125(2) supraoccipital subtriangular and widest near its ventral margin (convergent with *Herrerasaurus*, *Thescelosaurus* and some iguanodontians).

Based on the reconstructed phylogeny, the remaining four diagnostic features of *Psittacosaurus houi* are homoplastic within *Psittacosaurus*:

1. 2(0) preorbital skull length relative to basal skull length more than 50% (convergent with *Eoraptor*, *Herrerasaurus*, some thyreophorans, *Thescelosaurus*, *Parksosaurus*, *Prenocephale*, some iguanodontians and *P. amitabha*);
2. 96(1) arched smooth depression on posterior edge of frontal (convergent with *Heterodontosaurus*, *P. sinensis*, some chaoyangsaurids and some neoceratopsians);
3. 129(0) contribution of basioccipital to basal tubera restricted, not extending ventrally, and basisphenoid contribution to the tubera seen in posterior view (convergent with *Lesothosaurus*, some pachycephalosaurians, *P. lujiatunensis*, *P. sibiricus*, *P. sinensis* and *Yinlong*);

4. 160(1) ventral process of predentary bifurcated (convergent with *Haya*, *Changchunsaurus*, *Hypsilophodon*, *Thescelosaurus*, some iguanodontians, *P. lujiatunensis*, some chaoyangsaurids and some neoceratopsians).

For more details on unambiguous synapomorphies on all nodes of the strict consensus tree, see Fig. S38 and Appendix.

## DISCUSSION

The present study reveals that the three psittacosaurid specimens, IVPP V12704, ZMNH M12414 and IVPP V12617, are assignable to a single species, namely *Psittacosaurus* (formerly *Hongshanosaurus*) *houi* based on the shared autapomorphy: long jugal bar of the postorbital process nearly twice the length of the temporal bar of the bone (*You, Xu & Wang, 2003*; *You & Xu, 2005*). The following discussion pertains to ZMNH M12414 and IVPP V12617, the referred specimens of *P. houi* in terms of the taxonomic comparisons because IVPP V12704, the holotype, represents a juvenile that is inadequate for taxonomic comparisons.

### Notes on deformations in ZMNH M12414

ZMNH M12414 has some fractures in the following parts: both premaxillae and nasals, the right prefrontal, postorbital, squamosal, parietal, angular surangular, and the left maxilla-jugal contact, splenial and dentary. The ventral distortion of the right supratemporal fenestra (Fig. 1D) may be influenced by the fractures in the anterior ramus of the right squamosal (Figs. S12B, S12H), the central portion of the right postorbital (Figs. S11B, S11D), and the posterior portion of the right parietal (Fig. S10E). In addition, the right angular and surangular are fractured (Fig. 2B), and the right mandibular condyle is slightly tilted posterodorsally compared to the left (Fig. 2F). Here, we demonstrate that these minor post-depositional deformations have minimal effects to the taxonomic assessments in the present study.

*Psittacosaurus houi* is distinguished from other *Psittacosaurus* species by the following five autapomorphies:

(1) Prefrontal-premaxilla contact narrow

The contact between the prefrontal and premaxilla is present on both lateral sides of the skull (Figs. 1A, 1B), whereas the left lacrimal is fractured and missing most of its body. The bones surrounding this contact (lacrimal, nasal and maxilla) also show no signs of displacement, particularly on the right side.

(2) Ventral margin of premaxilla raised above maxillary tooth row

If this feature is more likely due to post-depositional deformation, it probably results from the bilateral compression, or localized deformation and dorsal displacement of the ventral margin of the premaxilla. However, both jugal horns extend straight outward on each side, suggesting that there was less compressive deformation bilaterally (Fig. 1E). In addition, there is no significant misalignment in the articulations between the right premaxilla and maxilla and the surrounding bones (rostral, lacrimal, nasal, prefrontal and jugal; Fig. 1B).

(3) Axes of maxillary tooth row and dorsal process of jugal oriented at an angle of about 135 degrees in lateral view

The cranium is minimally affected by the dorsoventral compressive deformation, as discussed below. In addition, the jugal horns, particularly the well-preserved right one, extend straight outward on both sides, suggesting minimal bilateral compressive deformation. Therefore, the orientation of the maxillary tooth row seems to be original. The jugal articulates with several bones (lacrimal, maxilla, quadratojugal, palatine, ectopterygoid and postorbital), and there is no displacement of their articulations. Therefore, the bones comprising the laterotemporal fenestra (postorbital, squamosal, quadrate, quadratojugal and jugal) are well-preserved, with only the ventral-most extent of the postorbital missing. There is no noticeable displacement between these bones (Fig. 1A), and the orientation of the dorsal process of the jugal also seems to be in its original position.

(4) Supraoccipital subtriangular and widest at its ventral margin

The supraoccipital of *Psittacosaurus*, except for *Psittacosaurus houi*, is diamond-shaped. If the subtriangular shape of the supraoccipital of *P. houi* results from the deformation of the originally diamond-shaped supraoccipital, it should have been due to significant bilateral compression. However, the lack of misalignment of the adjacent bones (parietals and exoccipitals) and the preservation of both jugal horns suggest that the compressive deformation was insufficient to affect the shape of the supraoccipital.

(5) Long jugal bar of the postorbital nearly twice the length of the temporal bar of the bone

The long jugal bar is completely preserved in the right postorbital, whereas the left one lacks the anterior-most tip and appears shorter than the right (Fig. S11). ZMNH M12414 bears some fractures in the central part of the right postorbital and the posterior part of temporal bar of the same bone, and exhibits a ventral distortion in the right laterotemporal fenestra, although it is unlikely that these factors have caused an extension of the right jugal bar. The short temporal bar is better preserved in the left postorbital, and the effect of deformation on the left laterotemporal fenestra appears minimal, as mentioned in (3). Additionally, the left supratemporal fenestra is less distorted than the right side. Therefore, the length of the temporal bar likely reflect its original state.

Additionally, *P. houi* is distinguished from other *Psittacosaurus* species by the combination of the following characters:

- Preorbital length about one half of skull length

This character is valid only if the dorsoventral compressive deformation is minimal as suggested by the previous studies (*Sereno, 2010*; *Hedrick & Dodson, 2013*). The following lines of evidence falsify that major dorsoventral compression occurred to affect the PLP. To begin with, the bones comprising the preorbital region of ZMNH M12414 (rostral, premaxilla, prefrontal, lacrimal, maxilla and jugal) are generally well-preserved on both sides (Figs. 1A, 1B). The right lateral surface shows a better preservation than the left one with only some fractures in the premaxilla and lacrimal, and no displacement of these

bones are present. Similarly, there is no displacement of the bones that contribute to the skull length (rostral, premaxilla, maxilla, jugal, pterygoid, ectopterygoid, vomer, palatine, quadratojugal and quadrate; Figs. 1A, 1B and 1F). In addition, the connection between the braincase and the palate is completely preserved in ZMNH M12414 (Figs. S4, S5 and S16–S20), whereas in IVPP V12617 (referred specimen of *Psittacosaurus houi*), a broken process in the braincase-palate contact has been cited as evidence of the dorsoventral compression (*Hedrick & Dodson, 2013*).

- Laterotemporal fenestra dorsoventrally elongated and oriented at an angle of about 45 degrees in lateral view

The cranium appears minimally affected by the dorsoventral compressive deformation as noted above. The left laterotemporal fenestra shows better preservation, whereas the fractured anterior ramus of the right squamosal (Figs. S12B, S12H) and central portion of the right postorbital (Figs. S11B, S11D) form an incomplete dorsal rim of the right laterotemporal fenestra (Fig. 1B). In posterior view, only the right supratemporal fenestra is distorted ventrally (Fig. 1D), suggesting that this deformation is localized. On the left side, the bones comprising the laterotemporal fenestra (postorbital, squamosal, quadrate, quadratojugal and jugal) is well-preserved, with only the ventral-most extent of the postorbital missing, but without any displacement between each bone (Fig. 1A).

- Posterior margin of parietal nearly linear and perpendicular to sagittal crest with no indentation on midline

Except for *Psittacosaurus houi* and *P. meileyingensis* (*Sereno et al., 1988*), there is a distinctive indentation on the midline of the posterior margin of parietals in *Psittacosaurus* (*Zhou et al., 2006*; *Averianov et al., 2006*; *Sereno et al., 2007*; *You, Tanoue & Dodson, 2008*; *Sereno, 2010*; *Sereno, Zhao & Tan, 2010*; *Napoli et al., 2019*; *Podlesnov et al., 2023*). However, for this "indentation" to be formed by post-depositional deformation, only the center of the posterior margin would have to be locally deformed, which is inconceivable. Moreover, the presence or absence of the indentation among *Psittacosaurus* species in different conditions is clearly distinguished in the literature.

- Anterior margin of rostral and nasal gently sloped posterodorsally at 30 degree

As noted above, the right preorbital region is minimally fractured and less affected by dorsoventral compressive deformation. The bones comprising the anterior margin of the preorbital region (nasal, rostral, prefrontal and premaxilla) have a partial breakage in the premaxilla, but without displacement of each element.

## Notes on phylogenetic analysis

In our phylogenetic analysis, *Psittacosaurus houi* was recovered as the latest-diverging *Psittacosaurus* species along with *P. major*. Although *P. mongoliensis* has often been assigned to the basal species in previous studies focusing on the intraspecific relationships (*Sereno, 1987*; *Russell & Zhao, 1996*; *Xu, 1997*; *Averianov et al., 2006*;

*You, Tanoue & Dodson, 2008*; *Sereno, 2010*), it was recovered as a derived taxon next to the node of *P. major* and *P. houi*, in which the phylogenetic position of *P. mongoliensis* and *P. major* is consistent with *Napoli et al. (2019)*.

Compared to *Napoli et al. (2019)*, the addition of *Psittacosaurus houi*, *P. sinensis* and *P. sibiricus* results in a slight change in their phylogenetic relationships. For example, the present analysis finds *P. sinensis* as the most basal taxon, followed by *P. sibiricus*. Additionally, *P. lujiatunensis* and *P. gobiensis* form a sister group of the clade comprising *P. amitabha* and other later diverging species. This result contrasts with the one in *Napoli et al. (2019)* where *P. amitabha* is the most basal species of *Psittacosaurus*. This result could be attributed to the six shared characters between *P. houi* and *P. amitabha* (2, 9, 21, 30, 102 and 114).

## Attribution of ZMNH M12414 and IVPP V12617 to the same species

While we interpret that ZMNH M12414 and IVPP V12617 are assignable to *Psittacosaurus houi*, we address these morphological differences and present lines of evidence that these specimens belong to the same species. *Sereno (2010)* suggests that the presence of the lacrimal foramen is one of the diagnostic characters for the genus *Psittacosaurus*. Photographs and descriptions in the literature confirm the presence of lacrimal foramen in all *Psittacosaurus* species (*Psittacosaurus meileyingensis*: CAGS-IG-V-330 in *Sereno et al., 1988*; *P. neimongoliensis* in *Russell & Zhao, 1996*; *P. major* in *Sereno et al., 2007* and *You, Tanoue & Dodson, 2008*; *P. lujiatunensis* in *Zhou et al., 2006*; *P. gobiensis*: LH PV2 in *Sereno, Zhao & Tan, 2010*; *P. sinensis*: IVPP V738 in *Sereno, 2010*; *P. mongoliensis* in *Sereno, 2010*; *P. amitabha* in *Napoli et al., 2019* and *P. sibiricus*: KOKM 22985/2 in *Podlesnov et al., 2023*), except for *P. xinjiangensis* with incomplete skull elements (*Sereno & Chao, 1988*). *You & Xu (2005)* notes that IVPP V12617 has no openings or canals in the lacrimal. Although the lacrimal foramen found in the right lacrimal of ZMNH M12414 is slightly offset from the typical position described previously in *Psittacosaurus*, the coronal slices of the CT images show that the lacrimal canal is connected to this foramen (Fig. S35), supporting this identification. The distinct position of the lacrimal foramen in ZMNH M12414 is caused by a postmortem fracture, which peeled off the lateral surface of the right lacrimal. *Sereno (2010)* suggests that the presence of the lacrimal canal fenestra is one of the diagnostic characters for the genus *Psittacosaurus*. However, the fenestra cannot be identified between the lateral wall of lacrimal and premaxilla in ZMNH M12414 using CT techniques. This observation suggests that the lacrimal canal fenestra may be a taphonomic artefact and is therefore inappropriate as a diagnostic feature for the genus, as pointed out by *Napoli et al. (2019)*. On the other hand, while there is no lacrimal foramen in IVPP V12617 (*You & Xu, 2005*), other *Psittacosaurus* species have the foramen, suggesting that its absence in IVPP V12617 is likely a preservational artefact.

Other morphological features of ZMNH M12414 that are absent in IVPP V12617 (*Landi et al., 2021*) include reduced retroarticular and coronoid processes of the mandible positioned close to the main axis of the dentary. In ZMNH M12414, the posterior part of right mandible is partially fractured with a misalignment between the surangular and angular. On the left side, however, the bones comprising the retroarticular processes

(surangular, angular and articular) and the coronoid process (coronoid, surangular and dentary) are almost completely preserved, and no misalignment between the bones can be seen, suggesting that the effect of the deformation is small. Thus, future studies may clarify whether this morphological difference between ZMNH M12414 and IVPP V12617 are due to the post-depositional deformation or intraspecific variation (Figs. 2A, 2C and 2G).

Still, these specimens share multiple characters unique to them among *Psittacosaurus* species and suggest that they belong to the same taxon. Namely, ZMNH M12414 and IVPP V12617 share the following features: (1) the narrow prefrontal-premaxilla contact, (2) the ventral margin of the premaxilla raised above the maxillary tooth row, (3) the axes of maxillary tooth row and dorsal process of jugal oriented at an angle of about 135 degrees in lateral view, (4) the subtriangular supraoccipital widest at its ventral margin, (5) the jugal bar of the postorbital process nearly twice as long as a temporal bar, (6) a long preorbital length being about one half of the skull length (as in *Psittacosaurus amitabha*; *Napoli et al., 2019*), (7) the posterodorsally-elongated laterotemporal fenestra oriented at an angle of about 45 degrees in lateral view (as in *P. amitabha*; *Napoli et al., 2019*), (8) the posterior margin of the parietal nearly linear perpendicular to the sagittal crest with no indentation on the midline (as in *P. meileyingensis*; *Sereno et al., 1988*), (9) the anterior margin of the rostral and nasal gently sloped posterodorsally at an angle of 30 degrees from the vertical (as in *P. amitabha*; *Napoli et al., 2019*), and (10) the height of the retroarticular process approximately at the level of the dentary tooth row (as in *P. neimongoliensis*; *Russell & Zhao, 1996*). Considering a greater number of similarities than differences between the specimens, it is concluded that ZMNH M12414 and IVPP V12617 are assignable to the same species, *Psittacosaurus houi*.

## Synonymy of *Hongshanosaurus* with *Psittacosaurus*

The synonymy of the genus "*Hongshanosaurus*" with *Psittacosaurus* has been proposed in previous studies (*Sereno, 2010*; *Hedrick & Dodson, 2013*). The present study supports this hypothesis based on the observation that the most apparent difference between these two genera, the magnitude of the PLP, is no longer supported and recovery in *Psittacosaurus* clade.

The genus "*Hongshanosaurus*" is originally diagnosed by the PLP nearly equal to 50%, the elliptical shape of the orbit and external naris, and the posterodorsally-elongated laterotemporal fenestra (*You & Xu, 2005*). *Sereno (2010)* coins the PLP lower than 40% as a diagnostic feature of *Psittacosaurus*. *Sereno (2010)* further argues that, for "*Hongshanosaurus*", the diagnostic features listed above comes from the post-depositional deformation of the skull, concluding that "*Hongshanosaurus*" is a junior synonym of *Psittacosaurus*. In support of *Sereno (2010)*, *Hedrick & Dodson (2013)* performed the three-dimensional geometric morphometric analyses and found that "*Hongshanosaurus*" forms a morphological cluster with *Psittacosaurus lujiatunensis* and *P. major*. The same study also posits that "*Hongshanosaurus*" appears to have a large PLP due to dorsoventral compressive deformation. Following these studies, more recent analyses on the family Psittacosauridae have regarded IVPP V12617 as a species of *Psittacosaurus* (*P. lujiatunensis* in particular) (*Erickson et al., 2009*; *Zhao et al., 2013*; *Han et al., 2016*;

*Taylor et al., 2017*; *Han et al., 2018*; *Bullar et al., 2019*; *Zhao et al., 2019*; *Landi et al., 2021*; *Sakagami et al., 2023*; *King et al., 2024*).

These studies regard "*Hongshanosaurus*" as a junior synonym of *Psittacosaurus* based on the interpretation that all purported "*Hongshanosaurus*" specimens in fact have the PLP < 40%, and *Psittacosaurus* among basal ceratopsians is diagnosed by the PLP < 40%. However, a recent description of the psittacosaurid *Psittacosaurus amitabha* demonstrates that the PLP of the taxon is larger than 40% without significant post-depositional deformation of the skull (*Napoli et al., 2019*), suggesting that the PLP < 40% may not be diagnostic to the genus *Psittacosaurus*. ZMNH M12414 in the present study provides additional evidence that the PLP can exceed 40% in *Psittacosaurus*.

According to *Hedrick & Dodson (2013)*, the connection between the braincase and the palate are reconstructed by plaster in IVPP V12617 (referred specimen of "*Hongshanosaurus houi*"), suggesting that the specimen has undergone dorsoventral post-depositional compression, leading to the conclusion that the PLP exceeding 40% is not the original feature of "*H. houi*". On the other hand, ZMNH M12414 possesses a complete connection between the braincase and the palate and shows no misalignment in the bones comprising preorbital region and skull length, indicating that the deformation had little effect on the PLP. Thus, unlike IVPP V12617, the PLP in ZMNH M12414 is likely original and exceeds 40%. Furthermore, ZMNH M12414 and IVPP V12617 likely represent fully mature skulls based on the histological evidence for IVPP V12617 (*Zhao et al., 2013*) and their sizes, suggesting that the PLP would not undergo significant changes with further ontogeny. Together with *Psittacosaurus amitabha* (*Napoli et al., 2019*), the fact that more than one example of mature psittacosaurid specimens exhibits the PLP of nearly 50% means that this feature cannot be used as a criterion to diagnose either "*Hongshanosaurus*" or *Psittacosaurus*. This observation would leave two features to distinguish "*Hongshanosaurus*" from *Psittacosaurus*: the elliptical shape of the orbit and external naris, and the posterodorsally-oriented laterotemporal fenestra, although these may be affected by post-depositional deformation (*Sereno, 2010*; *Hedrick & Dodson, 2013*). Numerous features are shared between ZMNH M12414 and IVPP V12617, and the genus *Psittacosaurus* by *Sereno (2010)* (See Table S3). Additionally, our phylogenetic analysis placed the OTU comprising ZMNH M12414 and IVPP V12617 deep within the *Psittacosaurus* clade. For these reasons, we argue that there is insufficient evidence to support the genus "*Hongshanosaurus*" and suggest that it is synonymized with *Psittacosaurus*.

### *P. houi* as a distinct species within *Psittacosaurus*

To establish *Psittacosaurus houi* as a distinct species within the genus *Psittacosaurus*, the presence or absence of diagnostic characters of all 10 valid species defined in previous studies should be confirmed (*Sereno, 2010*; *Sereno, Zhao & Tan, 2010*; *Napoli et al., 2019*). However, following *Hedrick & Dodson (2013)*, the diagnostic characters of the Lujiatun psittacosaurids, *Psittacosaurus lujiatunensis* and *P. major*, are considered invalid and therefore not used in this comparison. Instead, we have listed and compared characters among *P. houi*, *P. lujiatunensis* and *P. major* recognized in the present study (Table S4).

The validity of *P. houi* is supported by nine characters commonly found in ZMNH M12414 and IVPP V12617 but absent in *P. major* and *P. lujiatunensis*, species with which "*Hongshanosaurus houi*" were previously synonymized. Furthermore, although several diagnostic characters of other *Psittacosaurus* species are also present in *P. houi*, the discordance of their combination with any other *Psittacosaurus* species supports the distinctiveness of *P. houi* (Table S5). The conclusion that *P. houi* is distinct from *P. lujiatunensis* is further supported by the present phylogenetic analysis, which demonstrates that the two species do not form sister clades (Fig. S37). *Napoli et al. (2019)* argues that *P. lujiatunensis* and *P. major* are separate species, based on the phylogenetic hypothesis in which they fail to form a sister group. The present phylogenetic analysis also does not support the sister relationship between *P. lujiatunensis* and *P. major*.

## CONCLUSIONS

In the present study, a new specimen of a psittacosaurid skull, ZMNH M12414, was described, leading to a taxonomic reevaluation of the previously coined species "*Hongshanosaurus*" *houi*. Anatomical observations and phylogenetic analyses involving ZMNH M12414 and other previously described "*Hongshanosaurus*" specimens suggest the following: (1) ZMNH M12414 exhibits the features most consistent with those found in the previously described psittacosaurid "*Hongshanosaurus houi*", (2) "*Hongshanosaurus*" is to be synonymized with *Psittacosaurus*, and (3) *Psittacosaurus houi* is a valid species. *Psittacosaurus* is known from numerous specimens from many localities, but its diversity remains controversial. The recognition of *Psittacosaurus houi* supports the highly diverse nature of the genus. The present study leaves some questions for future work regarding the taxonomy and morphological characters of the genus, such as intraspecific variation and ontogeny. Revisiting other previously described species of *Psittacosaurus* based on additional specimens is necessary, and disputed anatomical questions may be addressed through CT analyses as demonstrated in the present work.

## ABBREVIATIONS

**AMNH**      American Museum of Natural History, New York, U.S.A.
**CAGS-IG**   Chinese Academy of Geological Sciences, Institute of Geology, Beijing, China
**FPDM**      Fukui Prefectural Dinosaur Museum, Fukui, Japan
**IGM**       Mongolian Institute for Geology, Ulaanbaatar, Mongolia
**IVPP**      Institute of Vertebrate Paleontology and Paleoanthropology, Beijing, China
**KOKM**      Kuzbass State Museum of Local Lore, Kemerovo, Russia
**LH**        Long Hao Institute for Stratigraphic Palaeontology, Inner Mongolia Autonomous Region, China
**PM TGU**    Paleontological Museum, Tomsk State University, Tomsk, Russia
**ZMNH**      Zhejiang Museum of Natural History, Zhejiang, China

## ACKNOWLEDGEMENTS

Yoichi Azuma (Fukui Prefectural University) and Yoshikazu Noda (Fukui Prefectural Dinosaur Museum) provided AI with constructive comments and support during the

specimen observation. The authors would like to thank ZMNH and FPDM staff for preparing and preserving the specimen for study. Faculty members and graduate students at Fukui Prefectural University provided constructive comments to improve the present manuscript.

### Funding
The authors received no funding for this work.

### Competing Interests
The authors declare that they have no competing interests.

### Author Contributions
- Asato Ishikawa conceived and designed the experiments, performed the experiments, analyzed the data, prepared figures and/or tables, authored or reviewed drafts of the article, and approved the final draft.
- Wenjie Zheng performed the experiments, authored or reviewed drafts of the article, and approved the final draft.
- Takuya Imai conceived and designed the experiments, analyzed the data, prepared figures and/or tables, authored or reviewed drafts of the article, and approved the final draft.
- Soki Hattori analyzed the data, authored or reviewed drafts of the article, and approved the final draft.
- Masateru Shibata conceived and designed the experiments, authored or reviewed drafts of the article, and approved the final draft.
- Soichiro Kawabe analyzed the data, prepared figures and/or tables, authored or reviewed drafts of the article, and approved the final draft.
- Xingsheng Jin conceived and designed the experiments, authored or reviewed drafts of the article, and approved the final draft.

### Data Availability
The skull 3D model is available at Morphosource: DOI 10.17602/M2/M594298.

The CT-data is available at Morphosource: DOI 10.17602/M2/M739493.

### Supplemental Information
Supplemental information for this article can be found online at http://dx.doi.org/10.7717/peerj.19547#supplemental-information.

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
