# Peer review of "Psittacosaurus houi, a longer snouted psittacosaurid from the Lower Cretaceous Lujiatun Unit of Yixian Formation, China, with the synonymy of the unresolved genus Hongshanosaurus revisited"

_PeerJ, doi:10.7717/peerj.19547_

## Round 0.1 · original submission · Major Revisions

This manuscript provides a detailed description of a Psittacosaurus skull from the Lujiatun Unit in the Yixian Formation of China, anchored on CT scan data, along with a reassessment of "Hongshanosaurus." The anatomical data are a major contribution to the literature, and will undoubtedly be useful for many future researchers. As noted by the reviewers, the taxonomic interpretations require some additional consideration before the manuscript is ready to be considered for publication.

The reviewers provide extensive and detailed commentary; although they all brought different perspectives, major themes to address in revision include:

1) The research published by Hedrick and Dodson (2013) must be more thoroughly addressed here, particularly around taphonomic effects.
2) The phylogenetic analysis should be expanded to include the relevant species P. sinensis and P. sibiricus, and potentially additionaly sampling.
3) Ontogenetic staging of specimens should be more thoroughly considered, especially through comparison with P. lujiatunensis.

Furthermore, I add some additional considerations:
1) Are the CT scans reposited and available for other researchers to access?
2) Consider adding a sagittal cross-section of the skull -- given the importance of palatal morphology relative to the rest of the skull in ceratopsian evolution, this sort of imagery would be very useful. (this is an optional suggestion)
3) I strongly advise adding some additional measurements for the skull, matching others published for ceratopsians. This could include occipital condyle dimensions, basic dimensions of major bones (e.g., the quadrate), orbital dimensions, tooth row length, etc.
4) How many teeth or alveoli are in each maxilla and dentary? This is an important metric of comparison, especially for ontogenetic comparisons.

Thank you for your consideration of these comments.

·

Excellent Review

This review has been rated excellent by staff (in the top 15% of reviews)
EDITOR COMMENT
This review is thorough, with many specific comments and general suggestions to improve the manuscript. The reviewer went above and beyond in providing some copy editing assistance, too. Thank you to the reviewer for their considerable efforts!

Basic reporting

The article uses English that is easy to follow, but does require work to be made into 'professional English' since there are numerous grammatical errors. Since the mistakes are small and I, personally, do not like continuously correcting non-native speakers nor like making authors pay for English checks, I have provided a list of all that I could find on my first pass through the manuscript draft. If accepted for major revision, I am happy to help the authors with their English writing considering how close to acceptable it already is in this draft.

The references used throughout the manuscript are relevant and do support the arguments made by the authors. However, they are too few in quantity. Considering the depth of the research record for all of Psittacosauridae, more citations and references are needed to build a framework where the researchers' questions and hypotheses properly fit. There are many instances of statements that were not developed by the authors being uncited throughout the manuscript, too. These issues are easily fixed by developing the background for your work more and more thoroughly citing the depth of previous research.

The figures look great, but could use some improvement. Some of the figures could be merged (e.g. Figs 3, 4; Figs 10, 11) to better compare the anatomy that they show. Some parts of the described anatomy, such as the rostral bone, do not have individual images where the views that are described in the text are shown. Similarly, the vomers - which play a critical role in the authors' conclusions - are very difficult to see in Figs 3 and 4. Every element described should have its own image, in my opinion. Those that do have their own figure, such as the mandibles, need to be shown in standard views. The oblique views of Figure 10 make the anatomy difficult to see in their angled state. As for the phylogenetic trees, a geological time scale would help add an extra layer of detail. I have noted that the added detail to the tree that includes all of the taxa may make it hard to read; however, a time scale would be a great fit for the ceratopsian tree.

Experimental design

The research question, methods (with exceptions), and intended audience fit PeerJ. I think the journal is a sufficient fit for the authors' work.

The reseach goals are well-defined in the introduction, but the broader significance needs to be more accutely addressed early on. Many researchers (myself included) have had many questions about the taxonomy of psittacosaurs for many years, so I do understand the importance of testing what others have proposed in terms of taxonomy. Saying this, I do not think the broader importance of this would be clear to non-ceratopsian specialists, casual readers, or students early in their career. I suggest the authors develop the introduction further with a broader history of psittacosaur research and the current questions surrounding the validity of Hongshanosaurus and the other Lujiatun psittacosaurs.

The methods used are sufficiently described in basic details, with one notable exception. In the 'Observations and measurements' section, the citations are not given for which manuscripts were used for data gathering. This must be shared in the next round of review, preferably in a table format with specimen names, collection numbers, and citations listed. As of right now, I do not know which nor how many specimens the authors compared the described specimens to for their results. Furthermore, the location of the 'soft' data (i.e. the digital data) does need to be addressed. I do not see a repository location for the scan data or digital models listed in the manuscript. This needs to be addressed prior to publication.

In terms of methods, I do have one major concern - deciding the ontogenetic stage of the specimens described. The work of Zhao Qi, Claire Bullar, and their coauthors, while cited, are not utilized enough to confidently estimate the ontogenetic stage of the specimens described in this project. The authors of the manuscript primarily use sutural fusion as the indicator of approximate ongenetic stage. To their defense, this kind of works, but the reliability of this method is too easily questioned. The authors absolutely must defend their claims of maturity with more methods. I advise the authors spend time comparing elements of the braincase to those of somatically immature P. lujiatunensis specimens in Bullar et al. (2019) to see what trends they can find between P. lujiatunensis and the ZMNH and IVPP specimens. Then, they can either defend their claims of near somatic maturity or rethink the ontogenetic stage of the specimens the authors are describing. My main concern here is that an ontogenetic signal may be reflected in their phylogenetic analysis and skewing an otherwise plausible result.

Validity of the findings

The data the authors have provided is reproducible and uses tried and tested methods. The results are not necessarily unexpected - but that is not a bad thing. Psittacosaur phylogeny and interspecific comparative anatomy is in dire need of reinvestigation. Their results support other previous synonymies, which is reassuring.

My main 'complaint' here is that the authors have not delved deeply enough into their phylogenetic analysis to really investigate what their results mean. This issue is compounded by the lack of depth the authors have assessed their phylogenetic results in both a historical and current lense. How does their phylogenetic analysis compare with historical ideas of psittacosaur taxonomy? How does a synonymizing Hongshanosaurus and Psittacosaurus (and erecting a new species within in Psittacosaurus) change our understanding of psittacosaur biodiversity or basal ceratopsian speciation? These questions can be easily answered and my complain remedied by starting with a stronger literature review into the previous phylogeniese of psittacosaurs, better explaining the results of this manuscript's phylogenetic analysis, and then combining the new results with the broader research picture. As it is currently written, the validity of the phylogenetic analysis is difficult to easily understand.

The descriptive portion of the manuscript is more solidly explained and is much stronger than the phylogenetic analysis section. I do recommend that the authors spend time making their descriptions more concise and better supported with citations to help ground their comparisons. A section that delves deeper into the current tables and more detailed comparisons between the genera, species, and ontogenetically immature specimens would be incredibly helpful in sorting out what are strong characters that can be used, what characters may not be as important as once thought, and which ones are not useful due to ontogenetic variability.

Additional comments

I want to add that, while I have pointed out what seem to be fundamental flaws, I am broadly positive about this manuscript. Its major strength is that it tests previous claims that a synonymy within Psittacosauridae is valid. Please see my specific comments below:

Line 23: Change ‘validities’ to ‘validity’

Line 27: Can you say if the skull has been formerly attributed to Hongshanosaurus or Psittacosaurus?

Line 27: Perhaps doublecheck to see if including referencing a specimen by its specimen number is allowed by PeerJ or has been done in past publications. Typically, this is non-standard.

Line 29: Change ‘northeastern China, the locality that has’ to ‘northeastern China – the same locality that has’. It will be easier to read this way.

Line 30: It is more accurate to say ‘exhibits many features that are also present in’ than ‘exhibits many features present in’

Line 31: Change ‘indicating they are assignable to the same’ to ‘indicating that both specimens are attributable to the same’

Line 32-34: The sentence beginning with ‘Large proportional length of the snout’ raises more questions for me than anything. It is difficult to understand what the authors mean from this sentence considering that they state a shared trait between two specimens they are comparing (and an additional, separate specimen) is ontogenetically variable. If you look at Bullar et al (2019), the braincase of P. lujiatunensis has many points that are ontogenetically variable. As written, the authors seem to suggest that this trait is the backbone of their argument for synonymizing two genera when more was done. Alternatively, do the authors mean that previous authors have split the two genera based on this character? Please clarify the meaning of this statement.

Line 34: Consider replacing ‘On the other hand’ with ‘Moreover’

Line 36: Make ‘leading to an establishment of P. houi’ its own sentence for easier reading. I suggest something like ‘...including P. lujiatunensis. Because of these features, the specimen in question better fits as its own species within Psittacosaurus: P. houi’

Line 36: Change ‘The’ to ‘A’ at the end of this line

Line 38: ‘forms should be ‘form’

Line 38: Should ‘clade’ not be ‘species’?

Line 38-41: Can you explain the sentence ‘The computed tomography techniques......diversity of Psittacosaurus’ in a little more detail? How was it helpful? What extra insight did the process give?

Line 41-42: Very small point, but ‘inter-specific’ and ‘intra-specific’ are most frequently used without a hyphen. Please change this throughout the manuscript

Line 48-72: After some consideration, I think the introduction needs to be expanded a bit. The overview is missing quite a few hallmark psittacosaur studies, all of which help support the authors’ conclusions. For example, ontogeny is brought up in the abstract, but no ontogenetic work is mentioned in the introduction. In the first sentence, the authors talk about bipedal posture, but posture in psittacosaurs – specifically in P. lujiatunensis – has been shown to be a factor of ontogeny; see Zhao et al (2013). As mentioned earlier, Bullar et al (2019) has also described the ontogenetically plastic nature of the braincase, too. I recommend the authors more deeply explore the background of psittacosaur work. Particularly, I think comments on ontogeny, the establishment of Hongshanosaurus historically, and defining characteristics of the psittacosaur skull need to be made based on prior publications.

Line 48-49: A few notes about your first sentence. As noted above, be careful not to over simplify the clade. Secondly, there are more recent citations that can be made than Osborn (1923). It is good to have to first instance, but I would add more recent citations considering how many times Psittacosauridae has had species added to it. Lastly, here is a suggestion on how to rewrite the sentence for accuracy and easier reading: ‘The Psittacosauridae represent a family of basal ceratopsian dinosaurs with bipedal posture and characteristic upper and lower jaws that form a beak-like rostrum similar to a parrot’s (Osborn, 1923)’

Line 49-51: Again, psittacosaurids have been reported from eastern Asia in many different publications; you even list the species in this same paragraph. This sentence should be heavily cited

Line 53: The authors are correct in that Psittacosaurus is the most speciose non-avian dinosaur, but this needs to be cited

Line 53: Replace ‘So far’ with ‘To date’

Line 56-57: I suggest changing ‘new species, and the total number of Psittacosaurus species remains debated’ to ‘new species, with the total number of Psittacosaurus species still debated’

Line 58-59: The first sentence of your second paragraph needs to be broken into two sentences; one that introduces Hongshanosaurus and another that explains the differences between the two genera

Line 58: Remove ‘hypothetically’

Line 60: Change ‘Within the genus’ to ‘Within Hongshanosaurus’

Line 60: Change ‘holotypic’ to ‘holotype’

Line 62: Replace ‘On the other hand, lack of’ with ‘However, a lack of’

Line 63: Can you clarify which genera you mean by ‘these genera’? Are you talking about Hongshanosaurus or Psittacosaurus or both?

Line 64: Replace ‘argues’ with ‘argued’

Line 65: Change ‘postmortem’ with ‘taphonomic’

Line 65: Change 'synonymizes’ to ‘synonymized’

Line 66: Change 'In addition, Hedrick and Dodson (2013) supports’ to ‘Additionally, Hedrick and Dodson (2013) supported’

Line 67: Change ‘the three-dimensional’ to ‘a three-dimensional’

Line 68: Change 'forms’ to ‘formed’

Line 68-70: Rewrite this sentence to say ‘Napoli et al. (2019), however, criticized Hedrick and Dodson (2013) because the study failed to take ontogenetic and intraspecific variation into account’

Line 70: Replace ‘Nonetheless, the series’ to ‘A series’. Also, this sentence would make more sense further up in this section – probably after the You and Xu (2005) citation

Line 73: Change ‘Psittacosauridae’ to ‘a psittacosaur’

Line 74-75: Replace ‘address the Psittacosaurid taxonomy based on morphological comparisons and phylogenetic analysis’ with ‘address current psittacosaur taxonomy based on morphology and phylogenetic analysis’

Line 76-78: The sentence ‘the present analyses suggests that Psittacosaurus (Hongshanosaurus) houi is a valid species and can be distinguished from P. lujiatunensis or any other species of Psittacosaurus is confusing. I recommend restructuring it as follows: ‘our study supports Psittacosaurus (Hongshanosaurus) houi as a valid species that can be distinguished from all other psittacosaur species – including the coeval P. lujiatunensis’

Line 78: Change ‘The present study sorts out’ with ‘This study helps to clarify’

Line 79: Remove ‘continuously’

Line 78-80: Your last sentence is a bit repetitive. I would first half of the sentence since the bulk of your study is specifically about Psittacosaurus rather than all of Psittacosauridae

Line 82-87: The abbreviations of museums should be bold-face for easier reading. I would also change the ‘=’ to a ‘–‘

Line 84: Remove the extra ‘;’ after the word ‘Japan’

Line 92: Remove ‘The specimen is stored at ZMNH’

Line 92-93: Change ‘The caudal surface of the skull is covered by sandstone matrix’ to ‘The caudal surface of the skull is obscured by sandstone matrix’. Also, please indicate that this is viewable in your first figure

Line 93-95: This sentence could be more concise (e.g. ‘The specimen was collected from the Lower Cretaceous-aged Lujiatun Unit, the lowermost unit of the Yixian Formation, from the Jehol Group. The locality is located near the village of Lujiatun, Beipiao, Western Liaoning, in northeastern China’)

Line 95-97: Can you give a little more geological detail here? How are the rock types of the unit related to one another (i.e. are they interbedded, not interbedded, or something else?)

Line 98: You say ‘lake sediments’ here but say ‘flooding’ in the prior sentence. Flooding is usually attributed to rivers. Perhaps reword either sentence to correct this or doublecheck the literature that was cited for clarification

Line 104: Change ‘Skull and mandibles’ to ‘The skull and mandibles’

Line 104: Change 'analyzed’ to ‘scanned’

Line 105: Remove ‘technique’

Line 106: Add ‘the’ after ‘Japan)’

Line 107: Replace ‘under’ with ‘using’

Line 107-108: Add ‘a’ after ‘kV,’; ‘an’ after 'µA’; and a space after ‘1,920’

Line 108: Correct ‘The parameter’ to ‘These parameters’

Line 109-110: Please define what you mean by ‘The resulting CT images’. Are they TIFF files? DICOM files? Something else?

Line 114-115: If you used the digital specimen to make measurements from, what program were the measurements made in? Amira?

Line 116-117: Change ‘literatures’ to ‘literature’

Line 118: Change ‘followed’ to ‘follow’

Line 121: Add ‘of species’ after ‘positions’

Line 122: Add ‘an’ after ‘V12617(‘

Line 121-126: Why were the two specimens scored differently if they are, assumedly, representative of the same species – especially if there are dissenting opinions on psittacosaur taxonomy in past publications? Further explanation of how and why the data matrices were chosen for the phylogenetic analysis would help clarify some of this confusion, I think

Line 125: Change ‘literatures’ to ‘literature’

Line 135: Remove () from citation or add them to the rest of the citations under this section

Line 136: Is ‘Figures 1-11’ a mistake? If so, please delete

Line 138-139: Remove () from citations or add them to the rest of the citations under this section

Line 151: Add ‘the’ after (1); ‘is’ after ‘length; and replace ‘a half’ to ‘one half’

Line 151-152: Add ‘the’ after (2) and ‘is’ after ‘nasal’; also, do you mean ‘rostral’ or ‘rostrum’?

Line 152-153: Reword your (3) statement to say ‘the laterotemporal fenestra is dorsoventrally elongated and oriented at an angle of about 45 degrees in lateral view’

Line 154: Add ‘the’ after ‘margin of’ and ‘is’ after ‘premaxilla’

Line 154-155: Add ‘amounts of’ after ‘(5) large’ and ‘is’ after ‘jugals’

Line 155: Add ‘is’ after ‘ramus of the’

Line 155-156: Change ‘process in contact’ to ‘process coming into contact’

Line 156: Add ‘is’ after ‘supraoccipital’

Line 160: Rephrase ‘measures a length of 160 mm’ to ‘is 160 mm long’. The wording needs to be more concise throughout your ‘Description and comparisons’ section

Line 160-161: You should be specific with your terminology. For example, ‘tip of the snout to the back of the quadrate’ should read ‘the anteriormost extent of the rostral bone to the posteriormost extent of the quadrate’. There are multiple examples of this throughout this section. Please fix this

Line 161: Add ‘has’ after ‘quadrate,’

Line 161: Add ‘has’ after 'horns and’

Line 162-163: Site your claim for somatic maturity based on the obliteration of suture lines

Line 164: When you say that the snout is relatively long, what are you comparing it to? The rest of the skull? Also, please define what you are calling the ‘snout’ region

Line 164-167: This last sentence is a bit confusing. I recommend the following corrections: ‘The proportion of the preorbital length against the total skull length is approximately 53%, being similar to the same ratios in IVPP V12716 and P. amitabha, 52% and 46%, respectively, and unlike other known Psittacosaurus species which exhibit less than 40% of the proportion’. Also, please remove the space between numbers and a % sign throughout the manuscript

Line 169: Why do some of the singular elements you describe not have accompanying figures? Please fix this

Line 169: Add ‘a’ after ‘subtriangular with’

Line 170: Add ‘a’ after 'having’

Line 171: Add ‘the’ after ‘contact with’

Line 172: Add ‘the’ after 'articulation between’

Line 174: Do you mean the anteriormost tip of the rostral when you say ‘the rostral tip is rounded’?

Line 179-180: You point out that the premaxilla meets the jugal in P. sinensis, is this not the case for the specimen you are describing? It looks like the jugal meets the premaxilla in Figure 3A

Line 185-186: Add ‘of the premaxilla’ after ‘The lateral surface’

Line 186: Do you mean ‘rugose’ when you say ‘roughed’?

Line 187: You say ‘which is particularly apparent on the right premaxilla’. Is this because of erosion on the left premaxilla? Or do you think this is taphonomic or due to a disease or injury?

Line 191: Add ‘the’ before ‘position’

Line 192: Do you mean other features are similar or the same when you say ‘While similar features are present in P. mongoliensis and P. major’? If they are similar, but different, you need to explain the similarities in a table or later in your discussion

Line 201: Add ‘all’ before ‘other psittacosaurids’

Line 201-202: Add ‘a’ before ‘sub-triangular’

Line 203: Replace ‘lies’ with ‘located’

Line 206: I am unsure what you mean by ‘are not confirmed in ZMNH’. Do you mean that you cannot confirm the presence of neurovascular foramina due to damage or scanning resolution? Please clarify

Line 207: Change ‘Psittacosaurs’ to ‘psittacosaurs’. This occurs a few times throughout the manuscript. Please correct all instances

Line 207-209: The sentence ‘Like most Psittacosaurs, posterior to the maxillary fossa, the maxillary protuberance is found on the posterior end of the rim of the cheek emargination near the maxilla-jugal suture as mentioned in Sereno (2010)’ is confusing. I recommend this wording: ‘Like most psittacosaurs (Sereno, 2010), a maxillary protuberance is found posterior to the maxillary fossa at the posterior margin near the maxilla-jugal suture’

Line 209: Replace ‘end’ with ‘margin’

Line 212: Change ‘lacrimal’ to ‘lacrimals’

Line 214: Change 'bounded’ to ‘bound’

Line 215: What is the size of the trapezoid shape small relative to in this sentence?

Line 218: Add ‘species’ after ‘Psittacosaurus’. Also, the state of both lacrimals is very hard to see in Figure 2. Why not make their own figure?

Line 222-225: The second sentence here is exceptionally long. Please break it up into at least two separate thoughts. Also, I just realized that the degree of fusion between the craniofacial bones has not been discussed. Can you elaborate somewhere how well the craniofacial bones are fused together or how well they can be separated from one another in the CT data?

Line 228: Replace ‘frontal’ with ‘frontals’ throughout this section since frontal bones are medially fused in dinosaur skulls. The correct way to word this is either ‘frontal bones’ or ‘frontals’

Line 232: Is the postorbital-frontal suture absent or is it not preserved? ‘Absent’ could mean not visible or obliterated

Line 235: Replace ‘It is’ with ‘They are’. Make this sentence plural

Line 241: Nothing to correct here. I just want to point out that the second sentence of this section is well done and the rest of your description sections should be as detailed

Line 246: In what way is this feature remarkable?

Line 252: What do you mean ‘skull roof process’? Do you mean it contributes to the total construction of the skull roof itself?

Line 262-263: I am confused what is meant by the last sentence. Can you clarify this further?

Line 265: Add ‘they are’ after ‘except’

Line 269: What do you mean ‘while differing from others’?

Line 265-277: You talk a lot about the length of certain rami here. Is there any connection between rami length and ontogeny? Or are the rami all short in the ZMNH and IVPP specimens when compared to other psittacosaur species?

Line 279: Replace ‘forms’ with ‘is’

Line 279-280: Can you describe the horn and rami three dimensionally?

Line 283: What do you mean by ‘the anterior surface is much larger than the posterior one’? Are you talking about height, width, or length?

Line 293: Can you quantify how large the surface area is? How does the surface area compare between the ZMNH and IVPP specimens?

Line 322: Break this up into individual bones. Unlike other parts of the skull (e.g. the braincase) the bones that make up the palate are not frequently lumped together. Also, considering that you later single out the vomer, having everything mixed together like this will be confusing for the reader. If they are fused together as you say and cannot be separated, describe them as separate bones and you can show the complex in its own figure with the labeled bones

Line 322: Change ‘well exposed’ to ‘well-exposed’

Line 323: What do you mean by ‘gentle vault’?

Line 324: Change ‘Long fused’ to ‘Long, fused’

Line 325-326: How does the location of the internal naris contrast with other species?

Line 340-341: A word of warning about the shape of the supraoccipital. Its shape is highly dependent on the ontogenetic stage of the specimen, as evidenced by P. lujiatunensis Bullar et al (2019). Consider this when you make claims about the ontogenetic stage of the specimens you are describing!

Line 342: Change ‘by parietal’ to ‘by the parietals’

Line 343: Change ‘extending’ to ‘and extends’

Line 344: Add ‘of the’ before ‘laterosphenoids’

Line 345: I just noticed that there have not been many measurements in your description. I strongly suggest adding a table that incorporates the measurements of the different elements you have been describing

Line 345: Remove ‘and the trigeminal ganglion is housed in the trigeminal fossa’

Line 346: Add ‘the’ after ‘distal ends of’

Line 347-348: What is the shape of the occipital condyle? How does it compare with other psittacosaur species? Also, change ‘slightly downward’ to ‘slightly ventrally’

Line 352: Add ‘a’ before ‘predentary’

Line 356: Change ‘with predentary and angular forming’ to ‘with the predentary and angulars forming’

Line 357: What is the mandible deep relative to?

Line 359: Is the shape of the predentary the same in dorsal view?

Line 361: Is the dentary flange only present on the right lateral surface because of poor preservation?

Line 362: Change ‘well exposed to ‘well-exposed’

Line 365: What do you mean by ‘it’ when you say ‘It contacts the dentary’?

Line 366: Add ‘the’ before ‘coronoid process’

Line 367: Add ‘the’ before ‘main axis’ and ‘dentary’

Line 367: It is not a bad idea to add the collections number to all of the psittacosaur species that you compare with throughout the manuscript. Please do this. Saying this, please cite where you got this information if you did not measure it or make the observations yourself

Line 372-385: The first half of this section needs to be worked into the similar section in your materials and methods section. The second half of this section needs to be heavily expanded. How do your results compare with others? How do your results compare with the historical ideas of psittacosaur phylogeny? What does this mean for your ideas about a “Hongshanosaurus”/Psittacosaurus synonymy? Later on, in the discussion, you need to explain what this means for psittacosaur biodiversity, speciation, and our understanding of psittacosaur taxonomy. As it is currently written, your discussion barely touches your phylogenetic analysis results at all when it is half of your data analysis!

Line 388-393: I agree with the idea that a juvenile should not be compared with other taxa that have been described based on adults. However, is a subadult any better than using a juvenile? The authors have noted that ZMNH M12414 is ‘nearly at its somatic maturity’, thus making it a subadult. How does ZMNH M12414 compare with other psittacosaur subadults? Will using a subadult to help support the synonymy of Hongshanosaurus and Psittacosaurus skew your results?

Line 397: Remove ‘On the other hand’

Line 399: Change ‘and its position’ to ‘but its position’

Line 401: Remove ‘Nevertheless’

Line 402: Remove ', a pathway innervating up to the rostrum,’

Line 403-405: I agree with ‘These observations indicate that the presence or absence of the lacrimal foramen may be a preservational artefact and should not be considered an apomorphy’. Saying this, can you comment on this further since you have CT scanned a specimen that preserves what may be a lacrimal foramen? Do you see evidence that the foramen continues in the bone or is the existence of this foramen still inconclusive?

Line 410-411: List all of these characters out or make a table out of them. You need to show strong support for grouping both specimens as the same species – especially if they are not both at the same ontogenetic stage

Line 412: Two points to make here. First, add ‘species’ after ‘than other Psittacosaurus’. Secondly, at some point in the manuscript, you need to be explicit with which specific species you are comparing ZMNH M12414 and IVPP V12617 against. You do not say it specifically, but I assume you are comparing the two specimens between other psittacosaur species of the Lujiatun Unit or at least the Yixian Formation. However, as you have stated, there are many species of Psittacosaurus. There needs to be a section where an in-depth comparison between the proposed P. houi and all other psittacosaurs is made

Line 413: You state ‘the parietal shape is parallel to the frill in dorsal view’ but it looks like there is at least a little bit of deformation near this area. For example, the right supratemporal fenestra seems to be more anteriorly oriented than the left. Is the parietal shape actually parallel to the frill or is this due to taphonomic deformation?

Line 414: Replace ‘is not in contact with’ with ‘does not come in contact with’

Line 416-417: Here, you state ‘Considering a greater number of similarities than differences between the specimens’. Can you explain the morphological differences between the two specimens?

Line 422: Replace ‘has been raised’ with ‘has been proposed’

Line 426: What is meant by ‘large proportion of preorbital length (>40 %)’? What is the preorbital length large in proportion to? Total cranium length?

Line 426: Add ‘shape of the’ after ‘elliptical’. Also, what is meant by ‘external naris’? Are you meaning that the external naris shares the same elliptical shape as the orbits?

Line 427-428: Reword ‘laterotemporal fenestra oriented postero-dorsally’ to say ‘both laterotemporal fenestrae are oriented posterodorsally’

Line 428-429: Much of ‘= 40 % or less, measured from the anterior end of the rostrum to the anterior margin of the orbit’ can be deleted. I would rewrite this as simply ‘short preorbital length (≤40%) as diagnostic’

Line 431: What do you mean when you say ‘as well as other diagnostic features’? If they are diagnostic, they should be listed – especially if Sereno (2010) is making an explanation for why they should not be considered diagnostic

Line 432: Change ‘postmortem’ to ‘taphonomic’. Postmortem is more frequently associated with the point after death but before burial in biology. Please change throughout the manuscript

Line 433: Remove ‘the’

Line 434: Change ‘performs’ to ‘performed’

Line 435: Change ‘forms’ to ‘formed’

Line 439-441: Not a problem, but I wanted to point out how well-supported your statement is here by all of the citations. More of your statements should look like this!

Line 441-443: ‘While the present study concurs with the previous hypothesis that “Hongshanosaurus” is to be synonymized with Psittacosaurus, its reasoning needs to be revisited’ can be deleted

Line 444: Add commas after ‘psittacosaurid’ and after ‘amitabha’

Line 445: Remove ‘apparently’

Line 453: Remove ‘a’

Line 455: Here, you state ‘indicating negligible (if any) deformation’ for ZMNH M12414. For the specimen as a whole, this seems to be untrue (see the supratemporal fenestrae of Figure 4B). If you mean the vomers specifically, this needs to be specified

Line 458: Bullar et al (2019) notes that the somatically mature specimen of P. lujiatunensis – which was aged with osteohistological sectioning – had sutures that were fully fused together. Furthermore, size is a poor indicator of age. See my broader comments for more details on how to better explain this

Line 460: Here, you say ‘through further maturity’. If the specimen is fully mature, then it cannot mature further and can only get larger. If the specimen is not yet fully mature, then it is a subadult

Line 461: Change ‘examples’ to ‘example’

Line 463: Remove ‘the’ before ‘ontogenetic change’

Line 465-466: For ‘the elliptical orbit, external naris and laterotemporal fenestra oriented postero-dorsally’, see my earlier comment for lines 426-428

Line 466: First, remove ‘On the other hand’. Secondly, again, numerous shared features need to be explained further. Here, you are drawing similarities between the ZMNH and IVPP specimen and other psittacosaur species. It is critical to name the shared characters here

Line 477: Delete ‘(if any)’

Line 472-489: In my opinion, you have not addressed if P. houi is separate from P. lujiatunensis well enough here. You need to make some summary comments on how different ontogenetic stages of P. lujiatunensis compares with P. houi, too. Furthermore, the Lujiatun Unit has also produced P. major (as you have stated earlier). How does P. houi compare with P. major? You have listed the traits of P. major in your Table 1, so why not explain the differences here? I would rename this section ‘Is P. houi a separate species from other Lujiatun psittacosaur species?’ and go into deep comparisons between each species

Line 493: Remove ‘to date’

Line 494: Remove ‘widely’

Line 494-496: The sentence ‘The validity of these species and taxonomic diversity within the genus have received much attention to understand the role of this herbivorous dinosaur in the ecosystem of Asia during the Early Cretaceous’ is a little out of place considering that you did not really go into this topic in your manuscript

Line 493-496: This section needs to be expanded or deleted. A conclusion should summarize your work and how it contributes to the broader body of knowledge about the subject

Line 503: ‘Psittacosaurus houi is a valid species, being distinguished from P. lujiatunensis’. Again, yes, but what about P. major?

Line 505: You say ‘The present study demonstrates the importance of the CT techniques to revisit the anatomy and taxonomy of previously described dinosaur taxon’, but I am left wondering why. I assume you say this because CT scanning allowed internal anatomy, such as the vomers, to be seen and compared with other psittacosaur species. You need to explain how technology, such as CT scanning and segmentation, has allowed you to reassess the species in ways that were previously unused

Line 505-510: Can you add a little more that explains why supporting a synonymy between “Hongshanosaurus” and Psittacosaurus and the establishment of a new species has helped our understanding of psittacosaur biodiversity specifically? I do not disagree with what you have said here, it just needs to be much more specific in how your study has done this

Figure 3 and Figure 4: Is there a way to merge these two images so comparisons between the different views are easier?

Figure 5: The white dotted lines are explained, but what are the green dotted lines?

Figure 6: Are you specifying the extent of the maxillary fossa with the green dotted line and shaded area?

Figure 7: This figure has a good explanation of the shaded area and dotted lines. Please follow this format for your other figures

Figure 9: Why only one view of the braincase? More views need to be shown considering how the braincase is a complex piece of anatomy that is comprised of many bones. Also, what are the shapes of the paroccipital processes based on if they are incomplete in the specimen?

Figure 10: Are A and B in oblique views? Please make the views from a perpendicular point of view

Figure 10 and Figure 11: Merge these two figures, as well

Figure 12 and Figure 13: First, is it possible to add a geological time scale to all of your trees to give a temporal reference for your taxa? I will appreciate if you think this makes Figure 12 too hard to read, but I would strongly recommend a time scale for Figure 13. Secondly, the ZMNH and IVPP specimens you described should come out as the same species here, correct? Or did you code some ontogenetically variable characters by accident? Moreover, why do the referred and holotype specimens of P. major form their own polytomy?

Table 1: You have stated there are 12 valid species of Psittacosaurus. Why only compare to nine? Moreover, why only compare with the ZMNH specimen and not all of the specimens you have figured in Figure 14?

Table 2: Same here. Why not include all three specimens you have in Figure 14? This would be a great way to show ontogenetic characters vs stable characters. More broadly, why is this comparing the ZMNH specimen against the genus Psittacosaurus? It would be a better idea to include “Hongshanosaurus”, too, so the reader can compare and contrast the described specimens between the two genera.

·

Excellent Review

This review has been rated excellent by staff (in the top 15% of reviews)
EDITOR COMMENT
This reviewer provided a heavily referenced assessment of the manuscript, with many helpful suggestions for edits and improvements. I particularly value the detailed comments in multiple areas for the specimen under consideration. Thank you!

Basic reporting

I only intend to give constructive comments for the manuscript - I apologize in advance if any part of this review sounds rude.

The article describes a new skull of Psittacosaurus from the Yixian Formation of northwestern China, element by element based on CT segmentation, and claims the validity of P. houi (new combination; formerly Hongshanosaurus houi). This level of detailed description was much needed for comparative works on ceratopsian evolution. This manuscript is generally well-written, but I have concerns on the taxonomic utility of the proposed diagnostic characters, as addressed below and in the annotated pdf. The MorphoSource file looks good.
The logic of the manuscript is as follows(, which I comment on below): a species of Psittacosaurus (that is distinct from the co-occuring P. lujiatunensis) is erected(/defended) based on a new skull. The diagnostic features in this new skull are shared with two more skulls that were previously assigned to Hongshanosaurus houi. Previously, the diagnostic features of Hongshanosaurus houi were considered to be due to taphonomic deformation and thus the taxon was synonymized. However, since the new skull shows these features while not being deformed, the features are likely of taxonomic value (and not due to postmortem deformation, even for the two skulls thought to have been deformed). Since there are now two adult skulls, more diagnostic features can be reliably established for the taxon. For the three specimens, the old name Hongshanosaurus houi is chosen to represent them(, while keeping the holotype that is morphologically immature and does not show all the diagnostic features of the taxon). Therefore, Hongshanosaurus houi is no longer a junior synonym of P. lujiatunensis, although it should be distinguished only at the species level and not the genus level, making it P. houi.
I am not convinced that Psittacosaurus houi is distinct from Psittacosaurus lujiatunensis and can be reliably identified from a pool of Lujiatun psittacosaurs – the new skull shows signs of deformation and all seven diagnostic features proposed seem to be prone to taphonomic deformation and/or ontogenetic variation (which admittedly assumes that the specimens belong to a single species, but I expect this to be the null hypothesis for specimens from geographic and stratigraphic proximity; elaborated below). Unless the authors can support the diagnosis better, I think the manuscript is better off just describing a well-preserved skull using CT segmentation, noting the probable intraspecific variations without arguing for taxonomic distinctiveness.
Since individual elements are already rendered, I think it would be great if the authors could provide figures of each element from multiple (six would be great) views, perhaps in a supplementary file. These will be useful for the readers trying to compare disarticulated elements. A table of standard measurements should be added, also perhaps in the supplementary file.
I get that the lower jaw is presented in an angle (Figs. 10A, and B) so that it would articulate as seen with the skull, but since it is so broad at the jaw joint it is hard to see the relative length of the elements, making it harder to compare with other published specimens. Since Fig. 1D already shows the jaw in articulation, perhaps presenting the jaw so that the lateral surface is parallel to the screen, as an isolated jaw would be conventionally presented, would be better. Alternatively, you could add figures from more views.

Experimental design

The authors utilized CT scanning to digitally prepare the specimen element by element and described the anatomy. The scan parameters are adequately provided.
The following issues that could alternatively explain the observed morphology of specimens here referred to P. houi and the phylogenetic position of P. houi should be addressed:

1. Taphonomic deformation should be accounted for.
It would help support that the data are biological and not biased taphonomically if a section addressing the effect of taphonomic deformation and how they may or may not impact the proposed diagnostic characters and other morphological differences from the holotype of P. lujiatunensis is added here. The variation among Lujiatun psittacosaurs has been quantitatively addressed by Hedrick and Dodson (2013) including comments on IVPP V12617, the referred specimen of P. houi in this study.
The authors state that the skull is "undeformed" because the vomer is intact, unlike IVPP V12617, but lines of evidence suggest some degree of deformation (mostly dorsoventral compression) that is likely relevant to the overall configuration of the preserved skull. First, the skull is asymmetrical as shown in Fig. 3C and 4C and the articulation with the mandible is off (Fig. 1F). The maxillary teeth laterally overlap the dentary teeth (only) on the right side. Also the nasal, right postorbital and squamosal, parietal, predentary, and right surangular shows breakage. The foramen magnum is wider than tall (rather than circular), which Hedrick et al. (2014) interpreted as being due to (dorsoventral) compression in another psittacosaur skull from the Lujiatun beds.
Many of the diagnostic features of P. houi can alternatively be explained by the shared deformation pattern between IVPP V12617 and ZMNH M12414 – i.e., if these features depended on the direction of the crushing of the skull during sediment compaction, as examined by Hedrick and Dodson (2013). If the animal lay with the ventral margin of the angulars parallel to the horizontal surface (as in IVPP V12617), the following crushing from the vertical compaction of sediments would result in the anterior side of the jugal facing more dorsally than in life position (although this is also seen in some P. specimens that are mostly mediolaterally crushed, due to shearing), as well as the anterior edge of the snout and the posterior margins of the orbit and infratemporal fenestra inclined more anteroventrally. This would also match the breakage seen on the nasal (where it would have been pointing the top), postorbital, and predentary (broken along the midline meaning that considerable force was applied). The positioning of preservation for IVPP V12617 with the ventral margin of the angulars parallel to the ground and the anterodorsal angle of the nasal pointing upwards can be supported by its detached supraorbitals (palpebrals) preserved on top of its quadrate (M. Son pers. obs.).

Hedrick, B. P., & Dodson, P. (2013). Lujiatun psittacosaurids: understanding individual and taphonomic variation using 3D geometric morphometrics. PLoS One, 8(8), e69265.
Hedrick, B. P., Chunling, G., Omar, G. I., Fengjiao, Z., Caizhi, S., & Dodson, P. (2014). The osteology and taphonomy of a Psittacosaurus bonebed assemblage of the Yixian Formation (Lower Cretaceous), Liaoning, China. Cretaceous Research, 51, 321-340.


2. Ontogenetic assessment of the new specimen ZMNH M12414
In line 458, ZMNH M12414 is assessed as a mature skull based on cranial suture closure while in line 145 it is referred to as belonging to a subadult. What criteria and definition was used to assign the ontogenetic stage as a 'subadult' (Hone et al., 2016)? This is important especially given that the manuscript argues for a separate species and ontogenetic variation has huge impact on the morphology of an individual (as well as taphonomic deformation) (Griffin et al., 2021).

Griffin, C. T., Stocker, M. R., Colleary, C., Stefanic, C. M., Lessner, E. J., Riegler, M., ... & Nesbitt, S. J. (2021). Assessing ontogenetic maturity in extinct saurian reptiles. Biological Reviews, 96(2), 470-525.
Hone, D. W., Farke, A. A., & Wedel, M. J. (2016). Ontogeny and the fossil record: what, if anything, is an adult dinosaur?. Biology letters, 12(2), 20150947.


3. Diagnosis and phylogenetic analysis
The validity of the diagnostic characters relies on them not being due to ontogenetic or taphonomic effects, as argued from the mature and undeformed new skull (ZMNH M12414). If this is the case, shouldn’t ZMNH M12414 be assigned as the holotype (perhaps a species that is not H./P. houi), especially when IVPP V12704 is morphologically immature and does not show all the diagnostic features of the taxon?
Of the seven characters used for Diagnosis (lines 150-157) of P. houi, character (1) is ontogenetically variable among Psittacosaurus specimens as mentioned in the main text, (7) seems ontogenetically variable according to data from Bullar et al. (2019), (6) is ontogenetically variable in P. lujiatunensis (long in juveniles and short in adults; Zhou et al., 2006), and characters (1)-(3) and (5) can be explained by taphonomic deformation (especially for IVPP V12617, as I described above). (4) and (6) are likely continuous characters (albeit ontogenetic and taphonomic influences) and ideally should be tested statistically (e.g., Czepiński, 2020; Powers et al., 2021).
The phylogenetic analysis used the character matrix of Han et al. (2018) as modified by Napoli et al. (2019). Since the Han et al. (2018) matrix was developed for broad Ornithischian interrelationships and Napoli et al. (2019) did not add characters pertinent to distinguishing Psittacosaurus species, it would be helpful if the authors added how many characters are applicable for Psittacosaurus species, how many/which characters differ among Psittacosaurus OTUs, and which characters are uniting the OTUs.
Coding specimens from the same taxonomic unit as separate OTUs is not recommended and may give spurious results (Hennig, 1999; Sharma et al., 2017). With that said, I am curious what would happen if the authors added a paratype of P. lujiatunensis, perhaps ZMNH M8138 from the same collection as the newly described skull, since the authors put both the holotype and paratype for P. houi and P. major but only the holotype for P. lujiatunensis.

Bullar, C. M., Zhao, Q., Benton, M. J., & Ryan, M. J. (2019). Ontogenetic braincase development in Psittacosaurus lujiatunensis (Dinosauria: Ceratopsia) using micro-computed tomography. PeerJ, 7, e7217.
Czepiński, Ł. (2020). Ontogeny and variation of a protoceratopsid dinosaur Bagaceratops rozhdestvenskyi from the Late Cretaceous of the Gobi Desert. Historical Biology, 32(10), 1394-1421.
Hennig, W. (1999). Phylogenetic systematics. University of Illinois Press.
Powers, M. J., Fabbri, M., Doschak, M. R., Bhullar, B. A. S., Evans, D. C., Norell, M. A., & Currie, P. J. (2021). A new hypothesis of eudromaeosaurian evolution: CT scans assist in testing and constructing morphological characters. Journal of Vertebrate Paleontology, 41(5), e2010087.
Sharma, P. P., Clouse, R. M., & Wheeler, W. C. (2017). Hennig's semaphoront concept and the use of ontogenetic stages in phylogenetic reconstruction. Cladistics, 33(1), 93-108.


4. Interpretation of the anatomy of ZMNH M12414
Some of the bone boundaries are different from the traditional interpretation of Psittacosaurus anatomy, including the lacrimal-jugal contact and around the articular. The posterior processes of the jugal are bilaterally asymmetric – is this due to breakage, or because it was hard to see the boundary of the thin jugal processes in CT images? The retroarticular process looks too short - is the posterior end of the articular broken? It’s more elongate in IVPP V12617.

Validity of the findings

The osteological description of individual cranial elements is itself of high value. However, the characters listed as diagnostic of Psittacosaurus houi are not necessarily unique for the proposed taxon or biological, and instead seem to fall within the observed range of ontogenetic and taphonomic variation of Psittacosaurus lujiatunensis. The authors should show that the characters are valid even after more taphonomic considerations, at least based on the work by Hedrick and Dodson (2013) - autapomorphies that cannot be attributed to ontogeny/taphonomic deformation are needed (best if shared with the juvenile holotype).

Additional comments

Some minor errors that can easily be fixed are as follows:
The manuscript keeps referring to IVPP V12617 as the "paratype" of Hongshanosaurus houi, but in the original description, You and Xu (2005) only assigned IVPP V12617 as a "Referred specimen", with the word "paratype" never mentioned in the paper.
The authors should use "posterior" throughout the manuscript replacing the "caudal" to be consistent, especially since "anterior" is used instead of "rostral."
In Figure 7, the "white dotted line" in the caption is missing in the figure.
Tables 1 and 2 are not cited in the main text.
Sakagami et al. (in press) in the reference is not cited in the main text.
Please refer to the annotated pdf attached for additional minor errors (typos, grammar, clarity, etc) and suggested alternatives.

·

Excellent Review

This review has been rated excellent by staff (in the top 15% of reviews)
EDITOR COMMENT
Thank you to this reviewer for their thorough assessment of the manuscript, particularly through reference to very relevant papers in the literature and a suite of helpful suggestions for improvement. The reviewer addressed multiple research domains (phylogeny, taphonomy, anatomy), which is particularly laudable.

Basic reporting

PeerJ Review submission 94217

Psittacosaurus houi, a longer snouted psittacosaurid from the Lower Cretaceous Lujiatun Unit of Yixian Formation, China, with the synonymy of the unresolved genus Hongshanosaurus revisited

Ishikawa et al.

Reviewer: Eric Morschhauser

The authors describe the anatomy of a specimen of Psittacosaurus from the Lujiatun Unit of the Yixian Formation, ZMNH M12414, using segmented CT scan data. The specimen shows some of the characters that had previously used to erect the distinct genus and species ‘Hongshanosaurus houi’. The species Psittacosaurus luijiatunensis and P. major have also been named from the Lujiatun Unit. The authors propose that, while the differences seen in this specimen and others previously referred to ‘Hongshanosaurus’ do not rise to the level of a new genus, this new specimen and the referred adult specimen of ‘Hongshanosaurus’ (IVPP V12617) represent a diagnosable species of Psittacosaurus. They propose the new combination P. houi for this taxon.

Basic Reporting

Overall, the article is clearly written. There are a few areas where the language usage is non-standard. For example in lines 116-118 the phrase, “the descriptions of original literatures
or from the figures within literatures,” is used. Typically, one would say “the descriptions in the scientific literature or from the figures within the articles, if no written description was included.” Today, the language “descriptions from the original peer-reviewed articles” might be more appropriate. But the scientific literature is typically treated as a singular.

In the section “Are ZMNH M12414 and IVPP V12617 assignable to the same species?” there is a passage that is difficult to follow. Lines 411-416 seems to be a list of features shared by these two specimens and not seen in other species of Psittacosaurus. But in some sentences, they are only described as being shared between these two specimens. For example, line 414-415 “Furthermore the absence of rugosity in the quadratojugal and the dorsoventrally deep mandibles are shared between specimens.” Then there is a sentence fragment in line 413-414, which does not even state what the relationship is between these characters. (“In addition, the parietal shape is parallel to the frill in dorsal view, and the ventral ramus of the squamosal is not in contact with the dorsal process of the quadratojugal.”) This passage would be clarified if we knew what specimens/species had these characters. The arguments in this passage would be stronger if the authors included explicit statements for characters that were 1) shared by these two specimens and not seen in any other Psittacosaurus, and 2) shared by these two specimens and also seen in some other species of Psittacosaurus. If characters were shared by the two specimens and other Psittacosaurus, the other species should also be clearly specified for each character.

The review of the taxonomic situation of the unusually speciose genus Psittacosaurus is generally comprehensive and able. I think the introduction does not adequately summarize the taxonomic statements of Hedrick and Dodson, 2013. There is an entire section “Taxonomic-Based Results” as well as Table 2 of Hedrick and Dodson, 2013 that directly address the same question as the authors, with important direct anatomical and taphonomic observation of specimens from the Lujiatun Unit. The reaction to Hedrick and Dodson (2013) here seems similar to that in Napoli et al, (2019). Napoli et al (2019) critiqued the morphometric methods of Hedrick and Dodson (2013) but completely ignored the independent taxonomic observations of the same paper. I will bring this up again as Hedrick and Dodson (2013) has direct bearing on the taxonomy of Lujiatun psittacosaurs.

Experimental design

The research question of the paper is well stated. The authors use CT scanning and manual segmentation to recover the anatomy of ZMNH M12414 that is still buried in the matrix of the original specimen. The anatomy of these elements is described in a standard and able fashion. I like the large number of figures in multiple views. The figures explaining the character of preorbital length (Figure 2) and the changes in preorbital length through ontogeny (Figure 14) were very well done. The figures of the CT segmentation, the phylogenies, and the illustrations are of sufficient resolution.

Given that the paper describes the cranial sutures as nearly obliterated, I would have liked a little bit more description of the segmentation methods and standards used to determine where the edges of the individual elements are, if possible.

The phylogenetic methods represent accepted methods and settings. There are a few items of note with the matrix and the methods described. Looking at the TNT matrix I see that several of the characters are treated as additive (ordered). That is fine, but the methods should explicitly specify how the characters (referenced by character number) are treated (ordered, unordered, etc.) in the analyses used to produce the trees. This ensures that future workers can double-check to faithfully duplicate the same settings in their own analyses. It also ensures that all readers are aware of the assumptions made in coding and analyzing the matrix.

The phylogenetic matrix from Naopoli et al., 2019 was used. This is a recent, generally appropriate matrix with extensive outgroup sampling.

The phylogenetic matrix comes from Napoli et al. 2019, but that paper is not the original paper coding this matrix. While Napoli et al. 2019 did change some of the codings, the bulk of the work coding the terminals and defining the characters was done by Han et al., 2018. (Han, F., C.A. Forster, X. Xu, and J.M. Clark. 2018. Postcranial anatomy of Yinlong downsi (Dinosauria: Ceratopsia) from the Upper Jurassic Shishugou Formation of China and the phylogeny of basal
ornithischians. Journal of Systematic Palaeontology 16 (14): 1159–1187.) The third supplementary file of that paper has the extensive character descriptions and figures of examples of the different character states. Now that many systematics workers are making detailed descriptions of characters and character states, I think it’s important to cite them every time one uses an iteration of their matrix. It keeps the codings from changing due to different researchers using different definitions of the same characters and character states in future without explicitly discussing any changes to character state definitions in the main body of their articles. It also helps people using the matrix in the future remember where it originally came from. After two or three citation cycles (people citing papers that cited papers that cited the original paper for the matrix), these trails of “matrix ancestry” can be pretty hard to tease out after the fact. Referring to the matrix as that of Han et al. 2018 (incorporating the modifications of Napoli et al. 2019) minimizes that confusion.

Additionally, the taxon sampling in the matrix from Napoli et al. 2019 is rather light to address the question of relationships within Psittacosaurus. Napoli et al. 2019 did increase the sampling of species of Psittacosaurus over that found in Han et al. (2018). But Napoli et al 2019 carefully tailored their taxon sampling to only include taxa that could be considered conspecific with P. amitabha on other grounds. It oddly excludes well-studied and well-represented species of Psittacosaurus like P. sinensis, and P. sibiricus. Given the current question, I would expand the coding of species of Psittacosaurus to at least include individuals from those two taxa. Ideally, the holotypes of all valid species of Psittacosaurus would be included. Also, since the current matrix is seeking to determine the relationships between Psittacosaurus specimens from the Lujiatun Unit, would it be possible to code more specimens from that unit into the matrix? P. lujiatunensis is only represented by a single specimen, but multiple were referred in the original description, including others from the ZMNH. Perhaps one or more of the referred specimens could be included? This way the phylogenetic hypothesis could be tested with more of the known individual variation.

Validity of the findings

The taxonomic discussion does not engage with the taxonomic section of Hedrick and Dodson, 2013. This article raises questions about the conclusions in Hedrick and Dodson (2013) based on the morphometrics methods used in part of that paper (as does Napoli et al. 2019). However, Hedrick and Dodson (2013) include a parallel taxonomic study that does not rely on morphometric methods. The taxonomic study of Hedrick and Dodson (2013) included in-person examination of many skulls of Psittacosaurus and Hongshanosaurus.

From the taxonomic methods section of Hedrick and Dodson (2013) “Morphometric techniques are not useful in directly determining taxonomic relationships due to variation from a large number of shape-based factors including sexual dimorphism, intraspecific variation, geographic variation [33], and as we demonstrate in this study, taphonomic variation. Therefore, a reanalysis of the proposed apomorphies of each species (P. lujiatunensis, P. major, and Hongshanosaurus houi) was performed by which each species was shown to be synonymous before morphometric analyses could be performed. Therefore, all known specimens referred to a specific Lujiatun species (IVPP V12617, IVPP V12704, ZMNH M8127, ZMNH M8138, CAGS [Chinese Academy of Geological Sciences, Beijing, China] VD04, CAGS VD05, LHPV1) were analyzed firsthand by B.P.H. (MS in preparation). Seventy-four additional specimens of Psittacosaurus in various degrees of preservation and ontogeny were examined including the holotypes of P. xinjiangensis, P. meileyingensis, P. mongoliensis, P. gobiensis, P. ordosensis, P. sinensis, P. mazhongshanensis, and P. neimongoliensis [1,35–40]. The majority of the examined skulls were also from the Yixian Formation (n = 64), the rest of which comprised of holotype or paratype specimens from other localities. Based on the large sample size of specimens examined, it was possible to determine the wide range of individual variation present in all species level apomorphies that have been proposed to separate Lujiatun psittacosaurids.”

Regrettably many of the details of that work are promised in a manuscript in preparation that has not been completed to my knowledge. But the taxonomic section of Hedrick and Dodson (2013) still reports distinct observations directly relevant to the current question, and draws from a much larger set of direct observations of original specimens than the manuscript under review. Table 2 in Hedrick and Dodson (2013) includes a summarized accounting of characters used to diagnose species of Psittacosaurus from the Lujiatun unit and Hedrick and Dodson’s assessment of each one on the basis of surveying 25 individual skulls of Psittacosaurus from the Lujiatun Unit. This is a sizable body of relevant work that has to be addressed more directly and more extensively in the current manuscript.

I will provide two examples of observations from Hedrick and Dodson (2013) that need to be addressed by the current authors: 1) They report that Psittacosaurs skulls from the Lujiatun Unit have varying prefrontal widths on a continuum that includes the ratios seen in the holotypes of P. lujiatunensis and P. major. 2) They report that the ventral ramus of the squamosal and the dorsal ramus of the quadratojugal are very often (indeed almost always) broken in Lujiatun psittacosaur specimens. Notably, they report that those processes are broken in the holotype of P. major (LHPV1) and in the adult paratype of Hongshanosaurus (IVPP V12617). If the relevant processes are broken, the presence or absence of contact between the squamosal and quadratojugal cannot be evaluated in those specimens or used as a feature uniting IVPP V12617 and ZMNH M12414.

There are many further examples like this. The authors are encouraged to meaningfully engage with them.

The last part of the Taxonomic-based Results in Hedrick and Dodson (2013) directly pertain to Hongshanosaurus houi. They conclude that all the features diagnosing Hongshanosaurus can be explained by dorsoventral crushing.

The fact remains that it must be established in this paper that the morphology seen as distinctive to Hongshanosaurus is not a result of crushing. Having examined many basal ceratopsian skulls myself, including some from the Lujiatun Unit, this type of dorsoventral crushing and shearing is a common failure mode for vertebrate fossil remains, and ceratopsians specifically. It is a plausible possibility. It is the duty of the authors of the current paper to construct a more robust argument as to why they don’t see the breakage and displacement that is present in IVPP V12617 as being an issue. They also must do more to demonstrate that similar breakage and crushing is not present in ZMNH M12414.

The current section focusing on arguing that ZMNH M12414 has no deformation is a little thin. It mostly relies on the vomer of ZMNH M12414 remaining intact. While an intact vomer would indeed indicate that there was little crushing in that particular region of the skull, it is insufficient to demonstrate a lack of deformation elsewhere. Indeed, in Hedrick and Dodson (2013) the description of the crushing in IVPP V12617 suggests that the palate is intact but that the plaster (and the damage) is between the palate and the braincase. Looking at ZMNH M12414 there is some deformation of the skull clearly visible. In Figure 3C the condyles of the left and right quadrate are clearly not at the same level, indicating some torsion and deformation of the skull is present.

There are enough other areas of damage in ZMNH M12414 to suggest the possibility of crushing and deformation of the skull. The left lacrimal is missing, and there is what appears to be an associated band of damage along an antero-ventral line from where the lacrimal should have been along the orbital margin and premaxilla/maxilla contact could be indicative of deformation. The nasals are a mosaic of small pieces, which could allow deformation. There are some sizable cracks and gaps that appear to be present in the left jugal and maxilla. The breaks and missing area of the right postorbital and anterior process of the squamosal could be indicative of crushing.

I want to be clear that these areas of damage do not conclusively prove that the skull has experienced crushing, but I simply wish to point to the authors that the burden is on them to demonstrate that the deformation experienced by the skull is not problematic to the claims they are trying to make. Proving that any deformation is not problematic is an important task of the manuscript because the absence of significant deformation is key to several of the characters the authors wish to advance as diagnostic of P. houi. This needs to be a thorough discussion. One or two sentences are not sufficient.

From my experience the types of characters the authors are advancing (the shape/inclination of skull openings, length ratios of various kinds, etc.) can be easily modified in the same population of individuals by taphonomic modification. The following characters could be affected by dorsoventral crushing or shearing that displaces the dorsal portions of the skull caudally and ventrally relative to the palate:
Preorbital length about a half of the skull length
Anterior margin of the rostral and nasal sloped caudodorsally
Dorsoventrally elongeated laterotemporal fenestra oriented at an angle of about 45 degrees in lateral view
Large surface area of the jugals exposed in dorsal view (due to tilting of the jugal).

In my experience with similarly-sized ceratopsians, there is a particular mode of deformation that is difficult to deal with and often under appreciated. I have seen skulls broken by a large number of small fractures. Millimeter and sub-millimeter deformation is distributed along many fractures such that each individual displacement is not immediately apparent. But the total effect of these small displacements significantly changes the morphology. The photographs of the skull in Figure 1 are not of sufficient resolution to evaluate the condition of the exposed portions of the skull. Higher resolution images for the first three parts of Figure 1 would resolve the issue.

Relating to engagement with Hedrick and Dodson (2013), the manuscript under review is using measurements and descriptions from the literature or figures in the literature for comparative purposes in their taxonomic evaluation. It does not seem that many of the original specimens were visited in person. While no one can be blamed for not traveling to see specimens in these last three years of disruptions and uncertainty due to the pandemic, it does affect those interpretations. I have studied several specimens of Psittacosaurus in-person myself, including IVPP V12617. My photographs and notes diverge in places from the published descriptions and I do not agree with one or two of the statements in this article. Most of these are points in the anatomical description and I have included them with the smaller comments below.

There are also minor points of interpretation, or identification of suture boundaries on the segmented CT scan figures where I would suggest changes. I have included those in a list below.

Table 1 and Table 2 do not appear to be cited in the text.

The work done in Table 1 is very useful and important. As we accumulate new specimens, it is important to revisit anatomical features that have been considered diagnostic or autapomorphic in the past. With increasing knowledge of the range of anatomy of individuals in a taxon or taxa in a group, we often find fewer and fewer characters truly unique to a species. I think, perhaps, the discussion should be expanded to include a specific section re-evaluating these characters that may not be diagnostic anymore. I know from experience that it takes a good bit of work to go an evaluate these characters, so I would encourage highlighting it in the body of the manuscript.

Additional comments

The authors conclude the abstract with the phrase. “The detailed evaluation of ontogenetic, intra-specific, and inter-specific variations are crucial to understand the true taxonomy and diversity of Psittacosaurus.” I heartily agree. But such a study would truly be an immense undertaking. The sheer abundance of specimens of Psittacosaurus available in institutions across the People’s Republic of China, and indeed across the world, from New York and Warsaw to UlaanBaatar and Tomsk, would take time and money to visit and study in detail. The convoluted taxonomic grouping that is Psittacosaurus is a second source of difficulty. While Sereno (2010) is an able review, new taxa have been found since, and many specimens, especially from the Lujiatun Unit, remain undescribed anywhere in the formal literature. The study that truly takes on the taxonomic diversity of Psittacosaurus as a whole has not yet been done. I am one of those researchers who would welcome it.
I want to emphasize to the authors that I do not think the current study needs to be the comprehensive study that addresses this taxonomic problem in full in order to be something worth publishing. I trust that the current study will be a valuable contribution, but I feel it needs some improvement before it gets there. The extensive use of computed tomography has given us a new, reasonably complete image of a Psittacosaurus skull from the Lujiatun Unit. But the current manuscript and analysis does not include a large enough number of specimens to truly settle the taxonomy of Psittacosaurus from the Lujiatun Unit, let alone broader Psittacosaurus taxonomy.

I think the manuscript would greatly improve if the authors meaningfully address my comments. In summary of everything in the previous sections, this involves:

1) expanding the sections of the manuscript defending ZMNH M1214 as not having deformation relevant to the taxonomic characters advanced. Improving figures of the exposed parts of the fossil would assist with this.

2) engaging with the non-morphometric taxonomic statements and anatomical observations of Hedrick and Dodson, 2013 in much more detail than the current manuscript.

3) coding more valid species of Psittacosaurus to the matrix of Han et al. 2018 (as modified in Napoli et al. 2019). Minimally I would suggest P. sinensis, and P. sibircus should be added. Though ideally all valid species of Psittacosaurus would be included. The authors should also seek to include at least one more specimen of P. lujiatunensis to the matrix to match the number of individuals from P. lujiatunensis to that of P. major and P. houi. The more specimens of Psittacosaurus from the Lujiatun beds that could be added, the better.

Of course there were many more minor comments, but these are the major issues that I strongly feel need to be addressed to significantly improve the manuscript.

The authors are welcome to reach out to me for further discussion or comment. I also have photographs of several relevant specimens that I would be willing to share with the authors. I would be glad to help in any way that I can.

Smaller comments.
Line 232-233 I am not sure how the Left frontal-postorbital suture appears to be L-shaped. It looks relatively straight with a slight bend in the middle in Figure 3B.

Line 244-246 Authors note this specimen does not have an incision in the caudal midline of the parietal. In this it is similar with P. meylingensis and differs from that of other Psittacosaurus specimens.
Examining photographs of P. meylingensis, the parietals appear broken to the left and the right side of the midline. I would not be confident that it lacks the subtle embayment seen in other Psittacosaurus.

Line 269-272 - Having examined IVPP V12617 the anterior ramus of the squamosal is not significantly shorter than P. lujiatunensis holotype description. It appears to be shorter on the left side as the left anterior ramus is broken. In my photographs, the articulated right anterior ramus of the squamosal extends past the midway point of the bar between the supratemporal and infratemporal fenestrae in an amount comparable to that seen in the P. lujiatunensis holotype.

Line 298-299 - Figure 3 does not show a quadratojugal of ZMNH M12414 divided in two by the jugal/quadrate contact. The jugal never contracts the quadrate.

The ventral portion of the quadratojugal in ZMNH M12414 in Fig. 3A and B appears to cover the quadrate as extensively as illustrated for P. lujiatunensis in Zhou et al., 2006. I am not seeing how they are different.

Line 311-313 - The quadrates look extensively broken in P. lujiatunensis hologype ZMNH M8137. Hedrick and Dodson (2013) mention the strongly asymmetric curvatures of the quadrates. (note: I have not visited this specimen in person.) Could the stronger curvature be an issue of crushing?

Curvature of the quadrates of IVPP V12617 doesn’t look all that far off from the curvature of P. sinensis IVPP V738.

Lines 322-336 - The description of the palate of Psittacosaurus doesn’t cite the preliminary discussion of basal ceratopsian palates in Dodson et al., 2010, which is relevant to the discussion.

Line 327 - The pterygoid is labeled simply pterygoid in Figure 4, when the text of the manuscript clearly indicates that it is a complex of the pterygoid, vomer, and palatine. I am interpreting this part of the manuscript as saying that it was impossible to segment out the individual bones, which is perfectly understandable. However, the figure caption should reflect that.


Line 322 - There is no palatine indicated in Figure 4. But based on other Psittacosaurus palates (Dodson et al., 2010) I would expect that part of the bone labeled the maxilla in Figure 4B is in fact the palatine.

Line 339-343 - The supraoccipital of IVPP V12617 does not have clear sutures and, despite the description of You and Xu, 2005, I would not hazard to delineate it. It’s not very clear on the actual fossil.

References Cited
Dodson, P., H.-L. You, and K. Tanoue. 2010. Comments on the basicranium and palate of basal neoceratopsians. In: Ryan MJ, Chinnery-Allgeier BJ, and Eberth DA, eds. New perspectives on horned dinosaurs: the Royal Tyrrell Museum ceratopsian symposium. Bloomington: Indiana University Press, 221-233.
Han FL, Forster CA, Xu X, Clark JM. 2018. Postcranial anatomy of Yinlong downsi (Dinosauria: Ceratopsia) from the Upper Jurassic Shishugou Formation of China and the phylogeny of basal ornithischians. Journal of Systematic Palaeontology 16(14): 1159-1187 DOI: 10.1080/14772019.2017.1369185
Hedrick BP, Dodson P. 2013. Lujiatun psittacosaurids: understanding individual and taphonomic variation using 3D geometric morphometrics. PLOS ONE 8(8): e69265 DOI: 10.1371/journal.pone.0069265
Napoli JG, Hunt T, Erickson GM, Norell, MA. 2019. Psittacosaurus amitabha, a new species of ceratopsian dinosaur from the Ondai Sayr Locality, Central Mongolia. American Museum Novitates 2019(3932): 1-36 DOI: 10.1206/3932.1
Sereno PC. 2010. Taxonomy, cranial morphology, and relationships of parrot-beaked dinosaurs (Ceratopsia: Psittacosaurus). In: Ryan MJ, Chinnery-Allgeier BJ, and Eberth DA, eds. New perspectives on horned dinosaurs: the Royal Tyrrell Museum ceratopsian symposium. Bloomington: Indiana University Press, 21-58.
Zhou CF, Gao KQ, Fox RC, Chen SH. 2006. A new species of Psittacosaurus (Dinosauria: Ceratopsia) from the Early Cretaceous Yixian Formation, Liaoning, China. Palaeoworld 15: 100n-114 DOI: 10.1016/j.palwor.2005.11.001
You HL, Xu X. 2005. An adult specimen of Hongshanosaurus houi (Dinosauria: Psittacosauridae) from the Lower Cretaceous of western Liaoning Province, China. Acta Geologica Sinica 79: 168-173 DOI: 10.1111/j.1755-6724.2005.tb00879.x

---

## Round 0.2 · Minor Revisions

Overall, the reviewers and I agree that the new version of the manuscript has been greatly improved through incorporation of feedback on the last version. The thorough description of ZMNH M12414 is a significant contribution on its own, in addition to the commentary on Psittacosaurus species taxonomy.

Following review of the updated manuscript, the reviewers identify two major themes to address in the next round of revision:

1) The impact of taphonomic distortion remains a common note from the reviewers. How did you assess the degree of distortion and its impact on morphology? Reviewer 3 notes that the Hedrick and Dodson study also included some notes on non-morphometric features of the specimens, which should be included more specifically beyond the original revision (e.g., lines 84-87 mentioned by reviewer 3).

2) The systematic paleontology section requires some updates, particularly concerning specimen referral and diagnosis. As noted by reviewer 2, the diagnosis should be reworked to identify autapomorphies specifically (supraoccipital morphology--note comments from reviewer 2 about this element), and for shared features indicate which other taxa share them. Although you make an argument for the referral of ZMNH M12414 and IVPP V12617 to P. houi, you must also address why they are the same taxon as the holotype specimen IVPP V12704 (see comments from Reviewer 3). I do think it is OK to use referred specimens to show diagnostic characters, *but* it is critical to demonstrate that they are the same taxon as the holotype.

Additionally, I note that the holotype for P. houi is inconsistently given as either IVPP V12704 (line 212, 394) or IVPP V12507 (line 75, 698); please correct this. V12704 should be the correct number for the holotype.

·

Basic reporting

The authors have substantially improved the English of the last draft and the new parts included in this draft use professional scientific English that is easy to follow. There is a little ambiguity in places (see suggestions below for how to fix some issues), but I can easily follow the train of thought meant by the authors. The manuscript itself is self-contained and does not pursue unrelated ideas or unsupported hypotheses. There are some changes suggested to some subsection titles for clarity, but these are mostly stylistic rather than an attempt to rein in ideas.

Since the last draft, the authors have expanded the background literature work which sets up a better historical framework for the rest of the manuscript. Saying this, the results and discussion sections could use more citations to help support the authors comparisons or claims. Considering that the main points of the manuscript focus on intraspecific and interspecific comparative anatomy and morphology of the most speciose non-avian dinosaur thus far described, citations should abound throughout the comparative anatomy sections and discussions surrounding the authors' findings. This is not the case in the current draft. I do not disagree with the manuscript as a whole and do not think the authors are making untrue claims. The claims that are made just need more support from existing literature.

Experimental design

The research question and its implications, though not strictly novel, are critical in understanding the phylogeny of psittacosaurs - a group of incredibly speciose non-avian dinosaurs. Herein, the authors find support for the synonymy of a previously proposed genus of basal ceratopsian, Hongshanosaurus, with the well-established genus Psittacosaurus. Even further, the authors support the synonymy and erect a new species within Psittacosaurus, P. houi. These findings give further support to other literature that has already suggested that Hongshanosaurus may actually be Psittacosaurus. In doing so, the research question proposed by the authors retests previous questions and work in dinosaur paleontology - a surprisingly rare event among non-avian dinosaur taxa.

The methods the authors used do not seek to reinvent the wheel but are tried and tested. Their dataset is large - as it should be for such a speciose group of dinosaurs. Ideally, the data collected would have been digital and three-dimensional for all taxa discussed in the manuscript. However, this would have been a massive undertaking. The comparisons made between different specimens are, however, more than sufficient to reach the sensible conclusions the authors reached.

Validity of the findings

The only comment I have concerning the conclusions is that I think that the conclusions - espescially what the authors' findings mean in a broader sense for disentangling psittacosaur phylogeny - could be expanded upon more in the discussion. However, I completely understand that truly expanding upon this would be a large undertaking in its own right.

The underlying data is robust and covers much of the psittacosaur taxa that are currently understood to be valid. A phylogenetically diverse and species rich dataset creates a robust comparative framework, which the authors' appreciably use. The conclusions the authors' reach are all supported by the work of the manuscript and are well-stated throughout the discussion.

Additional comments

I have the following broad and specific comments to make concerning your manuscript. Please use these when editing your next draft.

Broad points:

A comparably minor point, but a little more clarity in some of your statements would really help the reader understand why you chose to interpret some of your statements the way you did. These instances are varied and I typically get the broad meaning of what you meant, but a little more detail would greatly help. I have noted the major instances in my line-by-line points below.

Many of the comparisons you make need to be better cited. This is especially critical since the authors are trying to show support for a new species of psittacosaur and give better morphological/anatomical clarity among psittacosaur taxa. In lieu of comprehensive comparative figures, citations are the only way you can draw comparisons between specimens and taxa. In particular, extra citations are needed in your results (especially your element descriptions and comparisons!) and discussion sections.

Considering that taphonomic distortion is a concern that the authors bring up frequently within the manuscript, an explicit declaration of what is deemed to be too deformed to be deemed a diagnostic character might be wise to include. I was often confused by what metric the authors were using to quantify or even qualify (e.g. ‘large amounts of surface area’) what was and was not usable in certain parts of the manuscript. I do think the authors have done well to address or admit the presence of deformation in this latest draft, but a little more clarification in the definition of deformation would really elevate the draft and make the importance of deformation crystal clear to the reader.

Specific points:

Line 26: Replace ‘with an aid’ with ‘with the aid’

Line 32: Change ‘Large proportional’ to ‘The proportionally large’

Line 33: Delete ‘,’

Line 34: Change ‘Psittacosaurus’ to ‘P.’

Line 37-39: What features make – or at least, how many unique features – make the specimen you are discussing better fit as a distinct species? Please specify.

Line 60: Add ‘the’ between ‘In’ and ‘past’

Line 63: Add a ‘,’ after ‘genus and species’

Line 65: Add ‘between the two’ after ‘overall similarity’

Line 66: Replace ‘coined’ with ‘erected’ and delete ‘the series of’

Line 73: Reword to say ‘nine valid species: P.’

Line 97: Add ‘the’ before ‘diagnosis’

Line 116: Correct ‘within Psitacosauridae’ to ‘within Psittacosauridae’

Line 119-122: Keep generic abbreviations consistent. Abbreviate ‘Psittacosaurus lujiatunensis’ to ‘P. lujiatunensis'. Please doublecheck abbreviations throughout the manuscript

Line 134-135: Rephrase ‘a nearly complete skull including a mandible in articulation’ with ‘a mostly complete and articulated skull’

Line 138: Replace ‘cropping out near’ with ‘that outcrops near’ and change ‘Western Liaoning’ to ‘western Liaoning’

Line 140: Rephrase ‘without detailed information about the locality and horizon’ to say ‘without detailed locality and horizon data’

Line 148: Add ‘of the unit’ after ‘from the bottom to the top’

Line 151: Replace ‘dinosaur’ with ‘terrestrial’

Line 165: Add ‘and’ before ‘an interslice’

Line 212: I understand you had to modify the character matrix to score for 14 dorsal vertebrae, but can you explain your reasoning behind 14 being more basal than 15 vertebrae? I agree that one fewer vertebra is likely a more basal trait, but this could be interpreted by readers as a loss of a vertebra among derived taxa, too. Just explain your reasoning for scoring 14 for being more basal than 15 and this section will be okay.

Line 218-219: Reword ‘except for the following: maximum number of trees in memory equal to 99,999’ to say ‘except that the maximum number of trees was set to 99,999’

Line 254: Is there a way to quantify how much ‘a large amount of surface area’

Line 264: Change ‘damages’ to ‘damage’ and add ‘the’ before ‘premaxillae’

Line 269-275: I agree with the ontogenetic assessment, but you should include that Zhao et al (2013) demonstrated that psittacosaur cranial growth is isometric. You should include a sentence here that explains this to help support your statements that the skulls come from somatically mature individuals

Line 299-300: Cite ‘Unlike other Psittacosaurus species’

Line 306: Cite ‘as in all other Psittacosaurus species’

Line 312: The specimen number for P. meileyingensis should also be included if you are including the same for P. lujiatunensis. If the trait you are describing is noted in all specimens, this should be cited in lieu of a number.

Line 322: ‘The position of the external naris is located above the anterior part of the maxilla’. This position describes the location of the external naris amongst many archosaurs – if not most vertebrates. Can you clarify why this position among psittacosaurs is noteworthy?

Line 409: Reword ‘skull roof bar’ to say ‘bar on the skull roof’ for better clarity

Line 464: Replace ‘partly missing’ with ‘partially damaged’

Line 472: A note about PKUP V1054. That specimen is noted to not be a somatically mature adult. Is it possible that the anatomy you are discussing here is impacted by ontogeny? If yes, address this. If not, no change is needed.

Line 480: Add ‘species’ after ‘Psittacosaurus’

Line 512: Add a ‘-’ between ‘well’ and ‘preserved’

Line 518: Replace ‘fused and obscured’ with ‘obliterated’

Line 520: Do you mean that the vomer is fused with itself? If so, this would make it a paired bone, correct?

Line 528: ‘you’ should be ‘You’

Line 537: Replace ‘On the medial side’ with ‘Medially’

Line 548: Replace ‘and in contact with’ with ‘and contacts the’

Line 551: Delete ‘very’

Line 564: Replace ‘missing’ with ‘damaged’

Line 566: The foramen magnum is frequently measured by its height and width. Length, in a cranial context, typically refers to anterocaudal length. Diameter here implies width, is this correct?

Line 579: The fusion of the exoccipital and opisthotic is often regarded as the otoccipital in many archosaurs wherein the paroccipital processes are features on the opisthotic; also, ‘parooccipital’ should be ‘paroccipital’

Line 587: See earlier comment about length and diameter

Line 595: Hyphenate ‘well preserved’

Line 675-703: Consider adding bullet points or numbers to these for easier reading. The same could be done for the lists that follow after this.

Line 769: Add ‘The’ before ‘Remaining’

Line 796: I would recommend the subsection be entitled ‘Can ZMNH M12414 and IVPP V12617 be attributed to the same species?’ for better clarity and easier reading

Line 835: Delete ‘the’ before ‘dorsoventral compressive’

Line 852: Again, please clarify ‘large amount of surface area’. Through extension, taphonomic distortion can cause large amounts of unnatural surface area. You need to clarify what you mean here and make a case for it being anatomical rather than taphonomic.

Line 928: The subsection title implies bias in the result you are discussing. I know the authors make many good points on the basis of making P. houi a separate species and defend their hypothesis well. However, I would recommend rephrasing the subsection to prevent it from being interpreted as a foregone conclusion before you can explain your reasoning. I would suggest a minor edit such as ‘Should P. houi be considered a separate species within Psittacosaurus?’ or something similar

Line 956: Replace ‘they do not’ with ‘the proposed species does not’

Line 958: Add ‘for the authors’ argument’ after ‘as the basis’

Line 979-988: I would really like to see a small paragraph that expands on these conclusions in your discussion – though I appreciate this might be just beyond the scope of this manuscript. I can see the authors point-of-view (and reproducibility) of their current work, but I completely agree with their conclusions. Expanding this last paragraph and discussing further in the discussion would really drive home why work like this is critical to understanding psittacosaurs. However, this topic could easily be an overview paper in itself, so I do leave this as option for the authors.

·

Basic reporting

I have read the revised manuscript and the replies to my annotations and all reviews. I also went through all the supplementary files. I really appreciate the authors' thorough reply and more detailed descriptions and figures. The language is very clear with minimal typos. A nice overview of the anatomy of Psittacosaurus specimens based on the literature is provided throughout the revised description, although sometimes the anatomical comparisons do not seem to mention all Psittacosaurus species sharing the feature being discussed. Details on the ontogenetic assessment (size compared to individuals of known histological age) and more descriptions relating to deformation have been added. The Diagnosis has been revised. Of the 8 diagnostic features, some are suggested autapomorphies while others are shared with some but not all Psittacosaurus species based on specimens listed in Table S1. The distribution (sharing) of these features and whether they are autapomorphies should be noted in the Diagnosis. Marking the skull length and basal skull length used for this manuscript in Fig. 1 helps, as it was not always clear how the lengths were measured in some publications (“maximum skull length”, “total skull length”, etc). The added figures in the main text and the supplementary files are great and will be widely useful for anatomical comparative studies. Currently I can only see the captions for the supplementary figures and the link to the 3D model in the PeerJ Review page and not in the downloaded files themselves. A file with the abbreviations used and the captions for each supplementary figure would be useful. In addition, a link to the MorphoSource in the manuscript would be helpful – the model looks great in 3D!
An important contribution from the revised manuscript is setting the homology of the lacrimal canal. Sereno (2010) noted the presence of “lacrimal foramen” along the anterior margin of the orbit and “lacrimal canal fenestra” at the premaxilla-lacrimal boundary in Psittacosaurus and suggested that the “lacrimal canal fenestra” are unique to the genus Psittacosaurus. The presence of examples with and without an opening on the premaxilla-lacrimal boundary (“lacrimal canal fenestra”) was noted in P. sinensis, P. mongoliensis, and P. lujiatunensis by Zhou (2007) and Napoli et al. (2019) noted that in the region the bone is very thin and damaged in P. amitabha, leading both authors to conclude that the presence of “lacrimal canal fenestra” in Psittacosaurus species is a product of damage on the bone surface. Based on the CT scan of the new specimen ZMNH M12414, the authors were able to show the lacrimal canal itself. This is significant in that it means that the “lacrimal foramen” is the opening for the lacrimal canal (instead of the “lacrimal canal fenestra” in the premaxilla-lacrimal, contra Sereno, 2010) and supports that the “lacrimal canal fenestra” is just damage on the surface bone. I would just add that when referring to damage in ZMNH M12414 and the apparent lack of a foramen in IVPP V12617 (although I think it is present), I wouldn’t call them ‘deformation’ (Line 816) to avoid confusion with plastic deformation acting on the overall skull.
A phylogenetic reconstruction with all current nominal species using holotypes and referred adult specimens is presented in the revised manuscript - a much-needed update since Sereno (2010).
ZMNH M12414, the focus of the manuscript and here assigned to P. houi, was recently referred to P. lujiatunensis by Sakagami et al. (2024) with coauthors overlapping with the current manuscript. This citation should be added to Line 124 and “but no formal descriptions and species assignments were made prior to the present study” should be corrected accordingly. If the taxonomic distinctiveness hold true, as argued in this manuscript, the implications for the paleobiological interpretations of previous publications sampling ZMNH M12414 and IVPP V12617 could be discussed.

While I acknowledge the contributions to understanding Psittacosaurus anatomy and phylogeny from the current manuscript, the suggested diagnostic features for P. houi still seem to be invalid for taxonomic utility, as I discuss below. The Diagnosis for P. houi is based on shearing and breakage, while the differentiating features from P. lujiatunensis and P. major in Table 2 are either from deformation or hard to tell, based on my observations of relevant specimens (Son et al., 2022b; Son et al., 2024a and b). In the Rebuttal Letter, the argument is that “We believe that these bases are the best we can present at this time and that they are sufficient as explanations. To provide further evidence, we need to observe larger samples of Lujiatun psittacosaurids, so this should be left for future research.” I understand that taxonomic decisions (especially based on fossil morphology) are subjective and more specimens are needed to address variations within and among species. That being said, if this manuscript is to be published while keeping the taxonomic implications, with the diagnosis to be revisited in the future, I will respect the decision.

Experimental design

Some fractures interpreted as bone boundaries should be corrected. I unfortunately missed commenting on these bone boundaries in my first review:
- lacrimal: Both sides of the lacrimal are still not fully separated from the jugals. I get that the suture is hard to see in the CT images, but in that case a tentative suture could be drawn and marked ‘lacrimal or jugal’ to address uncertainty.
- supraoccipital: I only noted the reconstruction for IVPP V12617 by Bullar et al. (2019) in the first review, but based on first-hand observations, pictures, and comparisons, the supraoccipital margin by Bullar et al. (2019) and You & Xu (2005) is following the fractures while the actual bone margin on both left and right sides extend further laterally. This is relevant to the revised diagnosis of P. houi and should be addressed.
- basioccipital: The occipital condyle is rendered as a separate element from the basioccipital (‘basal tubera’ in the previous manuscript) but should be combined as one bone element, the basioccipital (so that the basal tubera and occipital condyle are parts of the bone element basioccipital).
- predentary: It seems the anterior parts of the dentary along breaks (left and right) are treated here as being part of the predentary, while the suture can be seen in lateral and anterior views (Fig5A,B,E). IVPP V12617 by Landi et al. (2021) (although they got the predentary of IVPP V15451 wrong) or other Psittacosaurus with good sutures (e.g., CAGS-IG-VD-004 by You et al., 2008 and KOKM 22985/2 by Podlesnov et al., 2023) show a much more limited predentary of Psittacosaurus. The predentary would extend only anterior to the splenial (Fig5D).

Validity of the findings

The revised manuscript provides reasoning for the diagnostic features not being interpreted as the results of deformation. However, as someone invested in Psittacosaurus anatomy, I would appreciate it if the following comments could be considered and perhaps the response added in the next revision.

The authors comment that the degree of deformation in ZMNH M12414 is hard to know and likely not so significant based on the absence of major displacement among elements, which is the prerequisite to the validity of P. houi. If P. houi is valid, chances are there are specimens from the literature that were misidentified. If the suggested diagnosis holds, we would expect to see P. houi preserved in a position so that it is deformed differently from what is suggested for the referred specimen IVPP V12617 (Sereno, 2010; Hedrick & Dodson, 2013). We would also expect to see Lujiatun Psittacosaurus specimens that are not P. houi preserved in the same direction suggested for IVPP V12617.
One common type of preservation orientation in the articulated skeletons of Lujiatun psittacosaurs is the one posed so that the snout points anterodorsally with the angular subhorizontal. The examples can be seen in ELDM V1038, IVPP V14341, IVPP V18343, and IVPP V18344 (Zhao et al., 2013 figs. 1 and S2; Zhao et al., 2019 fig. 5; MacLennan et al., 2024 fig. 3). In this position, the skulls could have gone through posteroventral shearing relative to the maxillary tooth row. The features I (and others) pointed out as being due to taphonomic effects in the posteroventrally sheared skulls of Hongshanosaurus houi are expected and seen in these specimens as well (e.g., longer preorbital length and dorsally wide jugal horn). This means that the suggested diagnostic features are correlated with the preserved orientation of the skull. This is why I argue against using susceptible to deformation features as being diagnostic.

For the 8 diagnostic features suggested, my comments are as follows:
“(1) long preorbital region reaching about one half of the skull length”: As commented in the first review, this is ontogenetically and taphonomically (by deformation) variable. It is also not diagnostic for the species because this is also seen in IGM 100/1132 (holotype of P. amitabha), according to the main text that follows.
“(2) posterodorsally sloped anterior margin formed by the rostrals and nasals”: This is the case for all ceratopsians (governed by the anteriorly acute-angled rostral), unless heavily anteroventrally sheared as in PKUP V1053 (a paratype of P. lujiatunensis; Zhou et al., 2006) and IVPP V22647 (Bullar et al., 2019).
“(3) narrow prefrontal-premaxilla contact”: It would help if the authors could provide an example of a specimen with a wide contact. The facial sutures are hard to discern in many specimens but at least in LH PV1 (large holotype of P. major), the extent of the contact seems similar to the P. houi specimens.
“(4) posterodorsally-elongated laterotemporal fenestra oriented at an angle of about 45 degrees in lateral view”: Again, deformation by posteroventral shearing could lead to this, especially when compared to PKUP V1053 and IVPP V22647 with anteroventral shearing.
“(5) higher ventral margin of the premaxilla raised above the maxillary tooth row”: This is not very obvious in IVPP V12617 and seems to depend on the preserved slope of the maxillary tooth row lateral surface and angle of view. The authors did not mention the condition in P. lujiatunensis. Variation among specimens of the same species should be noted.
“(6) large amount of surface area of the jugal exposed in dorsal view”: Again, deformation by posteroventral shearing could lead to this, especially when compared to PKUP V1053 and IVPP V22647 with anteroventral shearing, as can be seen in IVPP V18343 and IVPP V18344 (Zhao et al., 2019 fig. 5 and MacLennan et al., 2024 fig. 3).
“(7) subtriangular supraoccipital widest at its ventral margin”: The supraoccipital margins by You & Xu (2005) and Bullar et al. (2019) seem to be following fractures instead of sutures in IVPP V12617. It’s not very clear with cracks on the surface but even in IVPP V12617 the supraoccipital can instead be traced as ‘diamond-shaped’ – this is also phylogenetically bracketed by IVPP V18684 (Yinlong), LH PV1 (holotype of P. major), KOKM 22985/2 (P. sibiricus), and IVPP V12738 (holotype of Liaoceratops). To explain how ZMNH M12414 and IVPP V12617 share the same position and shape of fracture, it is plausible to have the same direction of force applied to the same structure (with the same weak parts) and create the same site of fracture, as noted previously in ceratopsians and not to be mistaken as apomorphies (e.g., Son et al., 2022a). In this case, the fractures are around the nuchal crest acting as a reinforcement structure with the weaker sides breaking to form a triangular shape. Similar cracks seem to be present in CAGS−IG−VD−004 and LH PV1 (P. major), while they both show the conventional "diamond-shaped" supraoccipital (You et al., 2008; Sereno, 2010).
“(8) the posterior margin of the parietal nearly linear perpendicular to the sagittal crest with no indentation on the midline.”: This seems to be present in IVPP V18343 and IVPP V18344 (MacLennan et al., 2024 fig. 3) as well. This is ontogenetically variable in P. lujiatunensis, although more common in smaller skulls than IVPP V12617 and the same size range also has individuals with posterolaterally diverging parietal posterior margins in dorsal view (pers. obs. by M. Son; Son et al., 2024a).

Additional comments

There are citations with first names used instead of last names (especially for Chinese authors in the main text, References, and the supplementary file Table S1) and should be corrected (e.g., Sereno and Chao, 1988 and Sereno, Zhao and Tan, 2010).
Please refer to the annotated pdf attached for additional minor errors (typos, grammar, clarity, etc) and suggested alternatives.

References Cited
Bullar, C. M., Zhao, Q., Benton, M. J., & Ryan, M. J. (2019). Ontogenetic braincase development in Psittacosaurus lujiatunensis (Dinosauria: Ceratopsia) using micro-computed tomography. PeerJ, 7, e7217.
Hedrick, B. P., & Dodson, P. (2013). Lujiatun psittacosaurids: understanding individual and taphonomic variation using 3D geometric morphometrics. PLoS One, 8(8), e69265.
Landi, D., King, L., Zhao, Q., Rayfield, E. J., & Benton, M. J. (2021). Testing for a dietary shift in the Early Cretaceous ceratopsian dinosaur Psittacosaurus lujiatunensis. Palaeontology, 64(3), 371-384.
MacLennan, S. A., Sha, J., Olsen, P. E., Kinney, S. T., Chang, C., Fang, Y., ... & Schoene, B. (2024). Extremely rapid, yet noncatastrophic, preservation of the flattened-feathered and 3D dinosaurs of the Early Cretaceous of China. Proceedings of the National Academy of Sciences, 121(47), e2322875121.
Napoli, J. G., Hunt, T., Erickson, G. M., & Norell, M. A. (2019). Psittacosaurus amitabha, a new species of ceratopsian dinosaur from the Ondai Sayr locality, Central Mongolia. American Museum Novitates, 2019(3932), 1-36.
Podlesnov, A. V., Averianov, A. O., Burukhin, A. A., Feofanova, O. A., & Vladimirova, O. N. (2023). New data on skull morphology of Psittacosaurus sibiricus (Dinosauria: Ceratopsia) using micro-computed tomography. Paleontological Journal, 57(10), 1128-1187.
Sakagami, R., Kawabe, S., Hattori, S., Zheng, W., & Jin, X. (2024). Endocranial anatomy of the ceratopsian dinosaur Psittacosaurus lujiatunensis and its bearing on sensory and locomotor abilities. Memoir of the Fukui Prefectural Dinosaur Museum, 22, 1-12.
Sereno, P. C. (2010). Taxonomy, cranial morphology, and relationships of parrot-beaked dinosaurs (Ceratopsia: Psittacosaurus). In New perspectives on horned dinosaurs: The Royal Tyrrell Museum ceratopsian symposium (pp. 21-58). Bloomington: Indiana University Press.
Son, M., Lee, Y. N., Zorigt, B., Kobayashi, Y., Park, J. Y., Lee, S., ... & Lee, K. Y. (2022a). A new juvenile Yamaceratops (Dinosauria, Ceratopsia) from the Javkhlant Formation (Upper Cretaceous) of Mongolia. PeerJ, 10, e13176.
Son, M., Makovicky, P. J., Erickson, G. M. (2022b). Cranial ontogenetic variation in Psittacosaurus with cladistic approach and its congruence with chronological age. Journal of Vertebrate Paleontology, Program and Abstracts, 2022, 328.
Son, M., Erickson, G. M., Zhou, C.-F., Yin, Y.-L., Makovicky, P. J. (2024a). Intra- and inter-specific variation in Psittacosaurus (Dinosauria; Ornithischia) and its implications for sympatry in early-diverging ceratopsians. Journal of Vertebrate Paleontology, Program and Abstracts, 2024, 512-513.
Son, M., Makovicky, P. J., Erickson, G. M., Zhou, C.-F., Yin, Y.-L. (2024b). Cranial ontogeny in the Early Cretaceous ceratopsian dinosaur Psittacosaurus lujiatunensis: intraspecific variation by age and taxonomic implications. 12th North American Paleontological Convention, 39, 398-399.
You, H. L., & Xing, X. (2005). An adult specimen of Hongshanosaurus houi (Dinosauria: Psittacosauridae) from the Lower Cretaceous of Western Liaoning Province, China. Acta Geologica Sinica‐English Edition, 79(2), 168-173.
You, H. L., Tanoue, K., & Dodson, P. (2008). New data on cranial anatomy of the ceratopsian dinosaur Psittacosaurus major. Acta Palaeontologica Polonica, 53(2), 183-196.
Zhao, Q., Benton, M. J., Sullivan, C., Sander, P. M., & Xu, X. (2013). Histology and postural change during the growth of the ceratopsian dinosaur Psittacosaurus lujiatunensis. Nature communications, 4(1), 2079.
Zhao, Q., Benton, M. J., Hayashi, S., & Xu, X. (2019). Ontogenetic stages of ceratopsian dinosaur Psittacosaurus in bone histology. Acta Palaeontologica Polonica, 64(2).
Zhou, C. F., Gao, K. Q., Fox, R. C., & Chen, S. H. (2006). A new species of Psittacosaurus (Dinosauria: Ceratopsia) from the early cretaceous Yixian Formation, Liaoning, China. Palaeoworld, 15(1), 100-114.
Zhou, C.-F. (2007). Psittacosaurus (Dinosauria, Ceratopsia) from the Early Cretaceous of western Liaoning, China. Doctoral dissertation, Peking University (China).

·

Basic reporting

peerJ-94217-Second Review

Initial comments: Having read the revised manuscript, I wanted to first state that the authors have put in a tremendous amount of work and vastly improved many sections of the manuscript. The phylogenetic treatment is much improved. The language is much clearer and the entire argument of the paper is easier to follow. The supplementary files are now very extensive. It’s good to see so much of the anatomy documented.

In this second review I will be focusing my comments on the main arguments of the paper and the author’s responses to my previous comments. I have read through the extensive comments of my peers, and the authors responses there.


PeerJ Review submission 94217

Psittacosaurus houi, a longer snouted psittacosaurid from the Lower Cretaceous Lujiatun Unit of Yixian Formation, China, with the synonymy of the unresolved genus Hongshanosaurus revisited

Ishikawa et al.

Reviewer: Eric Morschhauser

Basic reporting

The authors describe the anatomy of a specimen of Psittacosaurus from the Lujiatun Unit of the Yixian Formation, ZMNH M12414, using segmented CT scan data. The specimen shows some of the characters that had previously used to erect the distinct genus and species ‘Hongshanosaurus houi’. The species Psittacosaurus meylingensis, P. luijiatunensis and P. major have also been named from the Lujiatun Unit. The authors propose that, while the differences seen in this specimen and others previously referred to ‘Hongshanosaurus’ do not rise to the level of a new genus, this new specimen and the referred adult specimen of ‘Hongshanosaurus’ (IVPP V12617) represent a diagnosable species of Psittacosaurus. They propose the new combination P. houi for this taxon.

Experimental design

Detailed comments on experimental design are in my responses to the authors rebuttal, which I have put in the fourth section.

Validity of the findings

In this second read through the manuscript I realized that the taxonomic act the authors are proposing might not be quite correct. The diagnosis of P. houi must be based on IVPP V12704. That is the one and only name-bearing holotype of the “houi” species name at this time. That specimen is a juvenile and generally considered unsuitable as a basis for the name because it is strongly dorsoventrally crushed. Because the holotype is currently thought not to bear any diagnostic characteristics “Hongshanosaurus houi” is a nomen dubium. The larger skull, IVPP V12617 was referred to “Hongshanosaurus” houi but it bears no formal designation, either as a neotype or paratype. The current diagnosis, however, is based on ZMNH M12414, according to the author’s own rebuttal to my comments later on. If this specimen is diagnostic and distinct from other valid species of Psittacosaurus, then the authors might have two options, though I would appreciate the editor’s help on this. 1)They can name a new species with ZMNH M12414 as the holotype. I believe they could also propose that ZMNH M12414 be named the neotype of P. houi, though I am not certain that this process applies in the current situation. A neotype is typically used in situations where the original holotype is lost, not where the original holotype is insufficient. Neotypes are also typically named for the purpose of preserving a widely-used name, where the loss of the name would cause widespread confusion. P. houi is a new combination that has yet to be published and even “Hongshanosaurus houi” is a relatively recent name and not very widely used. In any nomenclatural acts, I would still strongly encourage the authors to avoid using any characters that could be affected by distortion because, even if your specimen is not distorted, that does not necessarily disprove that the other specimens you are comparing to your specimen are not distorted. Any taxonomic work looking at establishing the validity of Hongshanosaurus houi or Psittacosaurus houi would have to examine the juvenile holotype in detail. Though I suppose someone could propose IVPP V12617 as a neotype, but I think it is heavily crushed and not an ideal specimen for that.

Please note, I am unsure whether to classify these changes to the taxonomic as a minor revision or a major revision. I think, since the authors have the option to choose to name a new species, then it would have to be a major revision to ensure that any new name is properly vetted before publication. But I am open to the editor overruling my assessment here, if they do not feel it is supported by the circumstances.

More detailed comments relating to this issue are in my response to the authors' rebuttal, which I have put in the next section.

Additional comments

Below I will respond to the rebuttal to my comments under each one. The authors have overall done a very thorough job of addressing any concerns or stating their case.

Overall, the article is clearly written. There are a few areas where the language usage is non-standard. For example in lines 116-118 the phrase, “the descriptions of original literatures or from the figures within literatures,” is used. Typically, one would say “the descriptions in the scientific literature or from the figures within the articles, if no written description was included.” Today, the language “descriptions from the original peer-reviewed articles” might be more appropriate. But the scientific literature is typically treated as a singular.
➔The corresponding text has been modified (Lines 153-155).

Morschhauser: Changes are satisfactory.

In the section “Are ZMNH M12414 and IVPP V12617 assignable to the same species?” there is a passage that is difficult to follow. Lines 411-416 seems to be a list of features shared by these two specimens and not seen in other species of Psittacosaurus. But in some sentences, they are only described as being shared between these two specimens. For example, line 414-415 “Furthermore the absence of rugosity in the quadratojugal and the dorsoventrally deep mandibles are shared between specimens.” Then there is a sentence fragment in line 413-414, which does not even state what the relationship is between these characters. (“In addition, the parietal shape is parallel to the frill in dorsal view, and the ventral ramus of the squamosal is not in contact with the dorsal process of the quadratojugal.”) This passage would be clarified if we knew what specimens/species had these characters. The arguments in this passage would be stronger if the authors included explicit statements for characters that were 1) shared by these two specimens and not seen in any other Psittacosaurus, and 2) shared by these two specimens and also seen in some other species of Psittacosaurus. If characters were shared by the two specimens and other Psittacosaurus, the other species should also be clearly specified for each character.
➔This section has been modified following your suggestion (Lines 828-838).

Morschhauser: This section is vastly easier to read. But there are some other Psittacosaurus species that are omitted. Character (4) in line 831-832 is shared with P. sinensis, some specimens of P. lujiatunensis (PKUP V1054 in Figure 1 of Zhao et al., 2007), some specimens of P. major (You et al., 2008) and the holotype of P. meylingensis (contra the published illustration in Sereno et al., 1988). Character (7) slope of anterior margin of the rostral and nasal in P. amitabha is similar to that of at least some specimens of P. major (CAGS IG-VD-004 Fig., 1in You et al., 2008).


The review of the taxonomic situation of the unusually speciose genus Psittacosaurus is generally comprehensive and able. I think the introduction does not adequately summarize the taxonomic statements of Hedrick and Dodson, 2013. There is an entire section “Taxonomic-Based Results” as well as Table 2 of Hedrick and Dodson, 2013 that directly address the same question as the authors, with important direct anatomical and taphonomic observation of specimens from the Lujiatun Unit. The reaction to Hedrick and Dodson (2013) here seems similar to that in Napoli et al, (2019). Napoli et al (2019) critiqued the morphometric methods of Hedrick and Dodson (2013) but completely ignored the independent taxonomic observations of the same paper. I will bring this up again as Hedrick and Dodson (2013) has direct bearing on the taxonomy of Lujiatun psittacosaurs.
➔We have incorporated more detail in the introduction and in discussion about intra-specific and taphonomic variations as described in Hedrick and Dodson (2013).

Morschhauser: Lines 84-87 currently do not make it clear that Hedrick and Dodson conducted a taxonomic assessment of the diagnosis of these taxa independent of the morphometric analysis. Sorry to emphasize this, but Napoli (2019) simply did not evaluate the taxonomic statements at all, so it is important to note the distinction.

Experimental design
The research question of the paper is well stated. The authors use CT scanning and manual segmentation to recover the anatomy of ZMNH M12414 that is still buried in the matrix of the original specimen. The anatomy of these elements is described in a standard and able fashion. I like the large number of figures in multiple views. The figures explaining the character of preorbital length (Figure 2) and the changes in preorbital length through ontogeny (Figure 14) were very well done. The figures of the CT segmentation, the phylogenies, and the illustrations are of sufficient resolution.
➔We were unable to observe at first hand all the P. lujiatunensis specimens referred in the present study, but some data were taken from the literature. We have removed the figure 14 because it appears like we compare ontogeny of P. lujiatunensis, which we do not intend for the present study.

Morschhauser: Completely understandable. We all can only wish we had the time and money to see every specimen we would like to.

Given that the paper describes the cranial sutures as nearly obliterated, I would have liked a little bit more description of the segmentation methods and standards used to determine where the edges of the individual elements are, if possible.
➔We noted in the manuscript that suture lines are “nearly obliterated in some parts,” but they are visible enough to distinguish a bone from another. So, we have separated the skull into 32 elements by the sutures that can be seen on the CT images.

Morschhauser: Authors address this concern.

The phylogenetic methods represent accepted methods and settings. There are a few items of note with the matrix and the methods described. Looking at the TNT matrix I see that several of the characters are treated as additive (ordered). That is fine, but the methods should explicitly specify how the characters (referenced by character number) are treated (ordered, unordered, etc.) in the analyses used to produce the trees. This ensures that future workers can double-check to faithfully duplicate the same settings in their own analyses. It also ensures that all readers are aware of the assumptions made in coding and analyzing the matrix.
➔For additive characters treated in this study, please refer to “Materials & Methods” (Lines 187-190).

Morschhauser: The authors addressed this concern.

The phylogenetic matrix from Naopoli et al., 2019 was used. This is a recent, generally appropriate matrix with extensive outgroup sampling.The phylogenetic matrix comes from Napoli et al. 2019, but that paper is not the original paper coding this matrix. While Napoli et al. 2019 did change some of the codings, the bulk of the work coding the terminals and defining the characters was done by Han et al., 2018. (Han, F., C.A. Forster, X. Xu, and J.M. Clark. 2018. Postcranial anatomy of Yinlong downsi (Dinosauria: Ceratopsia) from the Upper Jurassic Shishugou Formation of China and the phylogeny of basal ornithischians. Journal of Systematic Palaeontology 16 (14): 1159–1187.) The third supplementary file of that paper has the extensive character descriptions and figures of examples of the different character states. Now that many systematics workers are making detailed descriptions of characters and character states, I think it’s important to cite them every time one uses an iteration of their matrix. It keeps the codings from changing due to different researchers using different definitions of the same characters and character states in future without explicitly discussing any changes to character state definitions in the main body of their articles. It also helps people using the matrix in the future remember where it originally came from. After two or three citation cycles (people citing papers that cited papers that cited the original paper for the matrix), these trails of “matrix ancestry” can be pretty hard to tease out after the fact. Referring to the matrix as that of Han et al. 2018 (incorporating the modifications of Napoli et al. 2019) minimizes that confusion.
➔We have revised the sentence as suggested (Lines 162-197).

Morschhauser: Authors have addressed this concern. Though I personally would have advised against changing the Psittacosaurus mongoliensis composite scoring. The original specimens could have had a perfectly stable referral. Recoding P. lujiatunensis from a composite to just the holotype was probably a good idea, given the taxonomic questions around Yixian Fm. psittacosaurs. Though I would wager that a quick check of Han et al., 2018 would give you a list of the specimens used for coding that terminal.


Additionally, the taxon sampling in the matrix from Napoli et al. 2019 is rather light to address the question of relationships within Psittacosaurus. Napoli et al. 2019 did increase the sampling of species of Psittacosaurus over that found in Han et al. (2018). But Napoli et al 2019 carefully tailored their taxon sampling to only include taxa that could be considered conspecific with P. amitabha on other grounds. It oddly excludes well-studied and well-represented species of Psittacosaurus like P. sinensis, and P. sibiricus. Given the current question, I would expand the coding of species of Psittacosaurus to at least include individuals from those two taxa. Ideally, the holotypes of all valid species of Psittacosaurus would be included. Also, since the current matrix is seeking to determine the relationships between Psittacosaurus specimens from the Lujiatun Unit, would it be possible to code more specimens from that unit into the matrix? P. lujiatunensis is only represented by a single specimen, but multiple were referred in the original description, including others from the ZMNH. Perhaps one or more of the referred specimens could be included? This way the phylogenetic hypothesis could be tested with more of the known individual variation.
➔We have added P. sibiricus and P. sinensis to our phylogenetic analysis.
We are aware of the importance of examining individual variation, but we were not able to add other Lujiatun psittacosaurs without first-hand observation. We will leave this for future research.

Morschhauser: Authors have addressed this concern.

Validity of the findings
The taxonomic discussion does not engage with the taxonomic section of Hedrick and Dodson, 2013. This article raises questions about the conclusions in Hedrick and Dodson (2013) based on the morphometrics methods used in part of that paper (as does Napoli et al. 2019). However, Hedrick and Dodson (2013) include a parallel taxonomic study that does not rely on morphometric methods. The taxonomic study of Hedrick and Dodson (2013) included in-person examination of many skulls of Psittacosaurus and Hongshanosaurus. From the taxonomic methods section of Hedrick and Dodson (2013) “Morphometric techniques are not useful in directly determining taxonomic relationships due to variation from a large number of shape-based factors including sexual dimorphism, intraspecific variation, geographic variation [33], and as we demonstrate in this study, taphonomic variation. Therefore, a reanalysis of the proposed apomorphies of each species (P. lujiatunensis, P. major, and Hongshanosaurus houi) was performed by which each species was shown to be synonymous before morphometric analyses could be performed. Therefore, all known specimens referred to a specific Lujiatun species (IVPP V12617, IVPP V12704, ZMNH M8127, ZMNH M8138, CAGS [Chinese Academy of Geological Sciences, Beijing, China] VD04, CAGS VD05, LHPV1) were analyzed firsthand by B.P.H. (MS in preparation). Seventy-four additional specimens of Psittacosaurus in various degrees of preservation and ontogeny were examined including the holotypes of P. xinjiangensis, P. meileyingensis, P. mongoliensis, P. gobiensis, P. ordosensis, P. sinensis, P. mazhongshanensis, and P. neimongoliensis [1,35–40]. The majority of the examined skulls were also from the Yixian Formation (n = 64), the rest of which comprised of holotype or paratype specimens from other localities. Based on the large sample size of specimens examined, it was possible to determine the wide range of individual variation present in all species level apomorphies that have been proposed to separate Lujiatun psittacosaurids.” Regrettably many of the details of that work are promised in a manuscript in preparation that has not been completed to my knowledge. But the taxonomic section of Hedrick and Dodson (2013) still reports distinct observations directly relevant to the current question, and draws from a much larger set of direct observations of original specimens than the manuscript under review. Table 2 in Hedrick and Dodson (2013) includes a summarized accounting of characters used to diagnose species of Psittacosaurus from the Lujiatun unit and Hedrick and Dodson’s assessment of each one on the basis of surveying 25 individual skulls of Psittacosaurus from the Lujiatun Unit. This is a sizable body of relevant work that has to be addressed more directly and more extensively in the current manuscript.
I will provide two examples of observations from Hedrick and Dodson (2013) that need to be addressed by the current authors: 1) They report that Psittacosaurs skulls from the Lujiatun Unit have varying prefrontal widths on a continuum that includes the ratios seen in the holotypes of P. lujiatunensis and P. major. 2) They report that the ventral ramus of the squamosal and the dorsal ramus of the quadratojugal are very often (indeed almost always) broken in Lujiatun psittacosaur specimens. Notably, they report that those processes are broken in the holotype of P. major (LHPV1) and in the adult paratype of Hongshanosaurus (IVPP V12617). If the relevant processes are broken, the presence or absence of contact between the squamosal and quadratojugal cannot be evaluated in those specimens or used as a feature uniting IVPP V12617 and ZMNH M12414.
There are many further examples like this. The authors are encouraged to meaningfully engage with them.
➔Thank you for reminding us of the significance of Hedrick and Dodson (2013).
In our revised manuscript, we have addressed to Hedrick and Dodson (2013) more comprehensively (Lines 81-93, 844-890, 893-899). In addition, we have excluded some characters, which was invalidated by Hedrick and Dodson (2013), from the diagnosis of P. houi.

Morschhauser: The authors have added substantial engagement with the claims of Hedrick and Dodson (2013) to the discussion and conclusions that is now more than appropriate with its role in the history of “Hongshanosaurus houi.” I do not think they have effectively countered the claims of Hedrick and Dodson (2013), however.

The last part of the Taxonomic-based Results in Hedrick and Dodson (2013) directly pertain to Hongshanosaurus houi. They conclude that all the features diagnosing Hongshanosaurus can be explained by dorsoventral crushing.
➔While Hedrick and Dodson (2013) mentioned the separation between the braincase and the palate in IVPP V12617 as evidence of dorsoventral crushing, the connection between them in ZMNH M12414 is completely preserved and no misalignment in the bones comprising the preorbital region and the skull length portion, suggesting that it did not undergo the significant dorsoventral deformation. Therefore, the common features observed in ZMNH M12414 and IVPP V12617, even though some states in the latter and thus once invalidated as diagnosis, can be treated as autapomorphies of P. houi. For more details, please refer to “Note on deformations in ZMNH M12414” (Lines 706-783, 803-841).

Morschhauser: Hedrick and Dodson (2013) also mention having seen characters used to define “Hongshanosaurus” in a continuum of variation during their survey of Psittacosaurus specimens from the Lujiatun Beds. They state that, “The long preorbital region, elliptical, caudodorsally oriented orbit and lateral temporal fenestra, and the jugal-quadratojugal process located ventral to the maxillary tooth row all would occur if the entire skull were taphonomically distorted such that the caudal aspect of the skull is dorsoventrally compressed and rotated about the undistorted rostral aspect of the skull. Varying degrees of these features are seen in Lujiatun Psittacosaurus sp. specimens based on their degree of dorsoventral compression. “ The concerns noted by Hedrick and Dodson are not restricted to IVPP V12617, but they are reporting a continuum of distortion between IVPP V12617 and other morphologies. The presence of these features in ZMNH M12414 do not change the fact that IVPP V12617 is distorted, but merely put ZMNH M12414 in that same continuum of distortion reported by Hedrick and Dodson (2013). In order to refute their observations, someone would either have to revisit those same skulls and find discreet morphologies rather than a continuum, or not be able to locate the intermediates that they have seen. Failure to test those observations leaves the judgement of Hedrick and Dodson (2013) valid. The implication of that judgement is that features such as “(1) a long preorbital length, which is about one half of the skull length, and (2) the dorsoventrally elongated laterotemporal fenestra, which is oriented at an angle of about 45 degrees in lateral view …(5) the large surface area of the jugals exposed in dorsal view … (7) the anterior margin of the rostral and nasal sloped posterodorsally (as in Psittacosaurus amitabha)” from this manuscript are of very uncertain validity. They do not seem a valid basis for diagnosis to me.

The fact remains that it must be established in this paper that the morphology seen as distinctive to Hongshanosaurus is not a result of crushing. Having examined many basal ceratopsian skulls myself, including some from the Lujiatun Unit, this type of dorsoventral crushing and shearing is a common failure mode for vertebrate fossil remains, and ceratopsians specifically. It is a plausible possibility. It is the duty of the authors of the current paper to construct a more robust argument as to why they don’t see the breakage and displacement that is present in IVPP V12617 as being an issue. They also must do more to demonstrate that similar breakage and crushing is not present in ZMNH M12414.
➔We think that the points you made are very critical.
Of course, it is never easy to evaluate the extent to which fossils are affected by deformation, as it can be biased by the subjectivity of the researcher.
The possibility that IVPP V12617 underwent dorsoventral compressive deformation was mentioned in Sereno (2010) and Hedrick and Dodson (2013), but only “the significant amount of plaster connecting the palate with the braincase” (misinterpreted as the crushed vomer in the previous version of manuscript) was mentioned as specific evidence for deformation.
The original manuscript relied heavily on a complete connection between the braincase and the palate as evidence for minimal deformation in ZMNH M12414. In the present study, we could not clarify how much a depositional deformation affected ZMNH M12414 after all. However, at least, we demonstrate that these post-depositional deformations are minimal to address taxonomic questions based on the diagnostic characters of P. houi raised in this study (Lines 706-783).

Morschhauser: I have quoted Hedrick and Dodson (2013) above that, while the braincase fracture is a major issue, they were positing a particular failure mode of the skull under compressive forces. This would be indicated by the caudal tilting of structures throughout the skull, especially the occiput. The relative length of the preorbital region would be due to the orbits being tilted caudally with the rest of the dorsal aspect of the skull. The occipital condyle would also be tilted caudally, increasing the overall basal skull length, but because the occipital condyle is more ventrally located than the orbital margin (note that in the holotype of P. lujiatunensis, the most anterior part of the orbital margin is dorsally located), the preorbital length would increase faster than the basal skull length as the tilt of the skull increases.

This type of distortion resulted from multiple areas of fracture and bending in the skull. I do not know why they were not detailed in that paper, but they are visible on the specimen.

Additionally, they were not basing this conclusion on a single skull, but noting this morphology was one of a range of morphologies seen in the skulls of Psittacosaurus. Regrettably, Hedrick and Dodson (2013) do not detail the range of distortion they observed in detail, except numerically in their morphometric analysis. And while the criticisms of Napoli et al., 2019 might suggest that you wouldn’t want to trust any statistical failure to distinguish the skulls in question, the wireframes from the homologous landmarks are representative of the types of qualitative data we use in morphological taxonomy all the time. (Though I would also add that the criticisms of Napoli are rather cursory and have shortcomings of their own.) Therefore the observation that needs to be refuted is that IVPP V12617 is not unique, but rather one step in a continuum of anatomy seen in Psittacosaurus specimens from the formation. The data in the current paper do not include enough individuals to assess the existence or non-existence of this continuum of individuals.



The current section focusing on arguing that ZMNH M12414 has no deformation is a little thin. It mostly relies on the vomer of ZMNH M12414 remaining intact. While an intact vomer would indeed indicate that there was little crushing in that particular region of the skull, it is insufficient to demonstrate a lack of deformation elsewhere. Indeed, in Hedrick and Dodson (2013) the description of the crushing in IVPP V12617 suggests that the palate is intact but that the plaster (and the damage) is between the palate and the braincase. Looking at ZMNH M12414 there is some deformation of the skull clearly visible. In Figure 3C the condyles of the left and right quadrate are clearly not at the same level, indicating some torsion and deformation of the skull is present.
➔ For the interpretation of the deformations in ZMNH M12414, please refer to “Note on deformations in ZMNH M12414” (Lines 706-783). Fig. 3C shows that the left and right quadrate condyles are not at the same level, but we do not think this is a critical point for the skull to have undergone significant deformation.

Morschhauser: This section has been expanded relative to the first manuscript. In the referred section the authors focus on describing why the distortion of the skull of XMNH M12414 does not affect the taxonomic characters they are using to refer ZMNH M12414 and IVPP V12617 to the same taxon.

There are enough other areas of damage in ZMNH M12414 to suggest the possibility of crushing and deformation of the skull. The left lacrimal is missing, and there is what appears to be an associated band of damage along an antero-ventral line from where the lacrimal should have been along the orbital margin and premaxilla/maxilla contact could be indicative of deformation. The nasals are a mosaic of small pieces, which could allow deformation. There are some sizable cracks and gaps that appear to be present in the left jugal and maxilla. The breaks and missing area of the right postorbital and anterior process of the squamosal could be indicative of crushing.
➔There is the band of damage along the missing lacrimal, which could have been accidentally peeled off, to the premaxilla/maxilla contact in the left side, but there is no significant difference in the proportions of the skull compared to the right side, so the deformation is likely small. Similarly, the cracks and gaps seen in the left jugal and maxilla do not appear to be the result of a major deformation when compared with another side. Although the nasals appear fragmented, they retain their symmetry and do not appear to have undergone significant deformation (Fig. 3C, E). The right supratemporal fenestra is considered slightly deformed due to the broken anterior process of the right squamosal and postorbital (Lines 353-356).

Morschhauser: I’m not sure what sedimentological process could have peeled off the lacrimal and adjacent bones, though I suppose they could have been damaged during collecting, however the fractures in the maxilla and jugal, while somewhat symmetrical, would suggest to me similar deformation rather than lack of deformation. The jugals appear rotated both laterally, with the anterior margin of the jugals projecting outwards at perhaps a steeper angle than in life and the dorsal border of the jugal rotated caudally and medially such that the lateral surface of the jugal is oriented more vertically than would have been in life. The authors also do not seem to account for the possibility of large deformation occurring along multiple sets of fractures, such that each individual displacement is minimal and may not be resolvable in the CT data used, but they add up to larger displacements. The large depression on the nasals between the left and right prefrontals is a case in point. I remain unconvinced that there is no deformation given how strongly the occiput is rotated relative to the palate. I will not comment further below on what is basically the same issue.

I want to be clear that these areas of damage do not conclusively prove that the skull has experienced crushing, but I simply wish to point to the authors that the burden is on them to demonstrate that the deformation experienced by the skull is not problematic to the claims they are trying to make. Proving that any deformation is not problematic is an important task of the manuscript because the absence of significant deformation is key to several of the characters the authors wish to advance as diagnostic of P. houi. This needs to be a thorough discussion. One or two sentences are not sufficient.
➔ ZMNH M12414 shows some breakages (Lines 234-236), and has undergone slight deformations in the right supratemporal fenestra and the mandibular condyle (Lines 236-238). However, we demonstrate that these post-depositional deformations are minimal to address taxonomic questions based on the diagnostic characters of P. houi raised in this study (See the section, Lines 706-783). We believe that these bases are the best we can present at this time and that they are sufficient as explanations. To provide further evidence, we need to observe larger samples of Lujiatun psittacosaurids, so this should be left for future research.



From my experience the types of characters the authors are advancing (the shape/inclination of skull openings, length ratios of various kinds, etc.) can be easily modified in the same population of individuals by taphonomic modification. The following characters could be affected by dorsoventral crushing or shearing that displaces the dorsal portions of the skull caudally and ventrally relative to the palate:
Preorbital length about a half of the skull length
Anterior margin of the rostral and nasal sloped caudodorsally
Dorsoventrally elongeated laterotemporal fenestra oriented at an angle of about 45 degrees in lateral view
Large surface area of the jugals exposed in dorsal view (due to tilting of the jugal).
➔Diagnosis of P. houi is based on the observation of ZMNH M12414, which is less affected by significant deformation, so our perception of its validity remains unchanged (See our explanations, Lines 706-783).

Morschhauser: (Note this repeats some of what I had said at the start.) The authors are attempting a type of taxonomic argument that is not permitted. The diagnosis of P. houi must be based on IVPP V12704. That is the one and only name-bearing holotype of the “houi” species name. That specimen is a juvenile and generally considered unsuitable as a basis for the name. IVPP V12617 was referred to “Hongshanosaurus” houi but it bears no formal designation, either as a neotype or paratype. And, having observed that specimen myself, it is quite crushed, even more extensively than has been formally defined in the literature. I have offered the authors of the current paper photographs which they may reference and draw their own conclusions. I would not base anything on that particular specimen. The current diagnosis based on ZMNH M12414 would have to be the holotype of a new species. Or the authors could argue that ZMNH M12414 should be the neotype of P. houi, though this is typically done to prevent confusion to the taxonomic community at large and I do not feel like “Hongshanosaurus houi” being lost would be a particularly confusing problem in the very small community that it would impact. It has not been in use for very long. But I would still strongly encourage the authors to avoid using any characters that in future diagnoses that could be affected by distortion in many specimens because, even if your specimen is not distorted, that does not necessarily disprove that the other specimens you are trying to compare with are not distorted.



In my experience with similarly-sized ceratopsians, there is a particular mode of deformation that is difficult to deal with and often under appreciated. I have seen skulls broken by a large number of small fractures. Millimeter and sub-millimeter deformation is distributed along many fractures such that each individual displacement is not immediately apparent. But the total effect of these small displacements significantly changes the morphology. The photographs of the skull in Figure 1 are not of sufficient resolution to evaluate the condition of the exposed portions of the skull. Higher resolution images for the first three parts of Figure 1 would resolve the issue.
➔The resolution of the photographs in Fig. 1 has been increased. Regarding the exposed portion of the skull, we have CT images to evaluate it and we do not think excessively high-quality photographs of the specimen is necessary for our purpose. We do not find concerning number of small-scale fractures in our specimen. Even if there are smaller fractures that are not visible, it does not affect our interpretation that the effect of deformation on the diagnosis of P. houi was minimal.

Morschhauser: Perhaps the image quality issues must be with the pdf send out for review. Because I can’t even tell the difference between matrix and bone in places it is so blurry.

Relating to engagement with Hedrick and Dodson (2013), the manuscript under review is using measurements and descriptions from the literature or figures in the literature for comparative purposes in their taxonomic evaluation. It does not seem that many of the original specimens were visited in person. While no one can be blamed for not traveling to see specimens in these last three years of disruptions and uncertainty due to the pandemic, it does affect those interpretations. I have studied several specimens of Psittacosaurus in-person myself, including IVPP V12617. My photographs and notes diverge in places from the published descriptions and I do not agree with one or two of the statements in this article. Most of these are points in the anatomical description and I have included them with the smaller comments below.
➔We have not been able to observe comparative specimens in person in recent years, so we had to rely on the literature for our observations of specimens. We have addressed your comments as best we could.

Morschhauser: While it is completely understandable that the authors have been unable to directly visit comparative specimens given the state of the world, I think the authors need to then change the scope of the conclusions of the paper to be something less ambitious. The current data set of a single specimen without the experience of directly observing the relevant holotypes of Psittacosaurus species from the Yixian Formation, or even consulting a decent number of high quality photographs of those specimens, is really insufficient to address the question. The current data are not expansive enough to support the conclusions in the discussion.

There are also minor points of interpretation, or identification of suture boundaries on the segmented CT scan figures where I would suggest changes. I have included those in a list below.
➔We have newly added a figure that indicates the boundaries of the suture lines in the supplemental information. Please see Fig. S3.

Morschhauser: The figure must have moved in the revision process. Fig. S3 shows the maxilla.

Table 1 and Table 2 do not appear to be cited in the text.
➔We have cited both tables in the revised manuscript.

Morschhauser: The authors have addressed this concern.

The work done in Table 1 is very useful and important. As we accumulate new specimens, it is important to revisit anatomical features that have been considered diagnostic or autapomorphic in the past. With increasing knowledge of the range of anatomy of individuals in a taxon or taxa in a group, we often find fewer and fewer characters truly unique to a species. I think, perhaps, the discussion should be expanded to include a specific section re-evaluating these characters that may not be diagnostic anymore. I know from experience that it takes a good bit of work to go an evaluate these characters, so I would encourage highlighting it in the body of the manuscript.
➔Table 1 (now Table 3) is presented to highlight the diagnostic characters of P. houi against those of other Psittacosaurus species. We agree that it is desirable to evaluate the anatomical features previously raised to define a series of Psittacosaurus species and test their validities. On the other hand, we believe such a large work should be presented on its in a separate volume, with all the species and associated features evaluated equally, with internal anatomy and post-depositional deformation addressed carefully. We hope our own work here will facilitate to initiate such a project.

Morschhauser: The authors choose not to emphasize this topic beyond the table. Fair enough.

Additional comments
The authors conclude the abstract with the phrase. “The detailed evaluation of ontogenetic, intra-specific, and inter-specific variations are crucial to understand the true taxonomy and diversity of Psittacosaurus.” I heartily agree. But such a study would truly be an immense undertaking. The sheer abundance of specimens of Psittacosaurus available in institutions across the People’s Republic of China, and indeed across the world, from New York and Warsaw to UlaanBaatar and Tomsk, would take time and money to visit and study in detail. The convoluted taxonomic grouping that is Psittacosaurus is a second source of difficulty. While Sereno (2010) is an able review, new taxa have been found since, and many specimens, especially from the Lujiatun Unit, remain undescribed anywhere in the formal literature. The study that truly takes on the taxonomic diversity of Psittacosaurus as a whole has not yet been done. I am one of those researchers who would welcome it. I want to emphasize to the authors that I do not think the current study needs to be the comprehensive study that addresses this taxonomic problem in full in order to be something worth publishing. I trust that the current study will be a valuable contribution, but I feel it needs some improvement before it gets there. The extensive use of computed tomography has given us a new, reasonably complete image of a Psittacosaurus skull from the Lujiatun Unit. But the current manuscript and analysis does not include a large enough number of specimens to truly settle the taxonomy of Psittacosaurus from the Lujiatun Unit, let alone broader Psittacosaurus taxonomy.
➔What we have mentioned in the end of abstract is very important but cannot be encompassed by the results of this study. Although our present study does not completely resolve the issues, we think it can be an important step to understanding the true diversity of Psittacosaurus. In the revised manuscript, P. sibiricus and P. sinensis were newly added to discuss the phylogenetic position of P. houi, which is the limitation of this study. The more comprehensive taxonomy of Psittacosaurus with more specimens is pending for the future work.

Morschhauser: The authors have addressed the issue.


I think the manuscript would greatly improve if the authors meaningfully address my comments. In summary of everything in the previous sections, this involves:
1) expanding the sections of the manuscript defending ZMNH M1214 as not having deformation relevant to the taxonomic characters advanced. Improving figures of the exposed parts of the fossil would assist with this.
➔In the revised manuscript, we demonstrate that the post-depositional deformations are minimal to address taxonomic questions based on the diagnostic characters of Psittacosaurus houi raised in this study (Lines 706-783). Additional views of individual skull elements from six angles have been added to the supplemental file (Fig. S1-32).

Morschhauser: The additional supplemental figures are very nice, though I notice that while they have very useful indicators of orientation, they appear to all lack scale bars. Or many of them lack scale bars. They should all have scale bars.

2) engaging with the non-morphometric taxonomic statements and anatomical observations of Hedrick and Dodson, 2013 in much more detail than the current manuscript.
➔We have referred to the work of Hedrick and Dodson (2013) in the revised “Discussion” in more detail (Line 706-783, 844-890, 893-899). As you pointed out, establishing the taxonomic validity of P. houi requires considering the taxonomic statements and anatomical observations in Hedrick and Dodson (2013). So, to engage with the previous study, we have listed numerous features that are shared or not shared between P. houi and, P. major and P. lujiatunensis, except for the invalidated characters for the morphological comparison in Table 2. For more details please refer to the revised manuscript (Lines 893-899).

Morschhauser: The revised manuscript has a much improved engagement with Hedrick and Dodson (2013). I have already commented on that engagement.

3) coding more valid species of Psittacosaurus to the matrix of Han et al. 2018 (as modified in Napoli et al. 2019). Minimally I would suggest P. sinensis, and P. sibircus should be added. Though ideally all valid species of Psittacosaurus would be included. The authors should also seek to include at least one more specimen of P. lujiatunensis to the matrix to match the number of individuals from P. lujiatunensis to that of P. major and P. houi. The more specimens of Psittacosaurus from the Lujiatun beds that could be added, the better.
➔We have added P. sinensis and P. sibiricus to our phylogenetic analysis as suggested. Currently, other Psittacosaurus species are unavailable for us to observe and add to the matrix beyond the one in Napoli et al. (2019).

Morschhauser: Understandable limitation.

Of course there were many more minor comments, but these are the major issues that I strongly feel need to be addressed to significantly improve the manuscript.
The authors are welcome to reach out to me for further discussion or comment. I also have photographs of several relevant specimens that I would be willing to share with the authors. I would be glad to help in any way that I can.
➔Thank you for your helpful remarks.

Morschhauser: Please do still reach out if you have any questions or would like to see the photographs.

Smaller comments.
Line 232-233 I am not sure how the Left frontal-postorbital suture appears to be L-shaped. It looks relatively straight with a slight bend in the middle in Figure 3B.
➔We have added a reference to Figure S9G, H which shows the L-shaped suture more clearly.

Morschhauser: Excellent. It is very clear now.

Line 244-246 Authors note this specimen does not have an incision in the caudal midline of the parietal. In this it is similar with P. meylingensis and differs from that of other Psittacosaurus specimens.
Examining photographs of P. meylingensis, the parietals appear broken to the left and the right side of the midline. I would not be confident that it lacks the subtle embayment seen in other Psittacosaurus.
➔We appreciate your information, but we currently have no evidence of damage (such as photographs you have) to contradict the absence of the midline incision depicted in Fig. 3C in Sereno et al. (1988).

Morschhauser: Are you sure you don’t want to see slightly more recent photographs of the specimen? Then you would have either evidence of damage, or photographic evidence that I’m mistaken. Reliance on Sereno’s nearly forty-year old illustrations is not ideal. Not that Sereno isn’t an excellent illustrator. But the specimens have changed in that time. Not everything is still preserved that was there. Some of these changes are 20 years old now.

Line 269-272 - Having examined IVPP V12617 the anterior ramus of the squamosal is not significantly shorter than P. lujiatunensis holotype description. It appears to be shorter on the left side as the left anterior ramus is broken. In my photographs, the articulated right anterior ramus of the squamosal extends past the midway point of the bar between the supratemporal and infratemporal fenestrae in an amount comparable to that seen in the P. lujiatunensis holotype.
➔We have removed this character from the diagnosis of P. houi.

Morschhauser: Authors have addressed the issue.

Line 298-299 - Figure 3 does not show a quadratojugal of ZMNH M12414 divided in two by the jugal/quadrate contact. The jugal never contracts the quadrate.
➔As pointed out, the jugal never contacts the quadrate. We have revised the sentences (Lines 402-403).

Morschhauser: Authors have addressed the issue.

The ventral portion of the quadratojugal in ZMNH M12414 in Fig. 3A and B appears to cover the quadrate as extensively as illustrated for P. lujiatunensis in Zhou et al., 2006. I am not seeing how they are different.
➔We have revised the sentence as “The posterior margin of the quadrate extensively overlaps the quadrate, while the quadrate condyle is exposed in lateral view as in P. lujiatunensis and P. gobiensis” (Lines 414-416).

Morschhauser: Authors have addressed the issue.

Line 311-313 - The quadrates look extensively broken in P. lujiatunensis hologype ZMNH M8137. Hedrick and Dodson (2013) mention the strongly asymmetric curvatures of the quadrates. (note: I have not visited this specimen in person.) Could the stronger curvature be an issue of crushing?
➔Whether the quadrate curvature is due to deformation is not well-discussed in Hedrick and Dodson (2013) and cannot be determined in the present study.

Morschhauser: Fair enough, but I was under the impression that one of the authors works at the institution housing ZMNH M8137. Are they not able to take time assess the specimen?

Curvature of the quadrates of IVPP V12617 doesn’t look all that far off from the curvature of P. sinensis IVPP V738.
➔The curvature of the quadrate in IVPP V12617 is comparable to that of P. sinensis (IVPP V738), but another specimen of P. sinensis (BNHM BPV149) has a strongly arched curvature of the quadrate. We have revised the sentences to mention this (Lines 429-432).

Morschhauser: Authors have addressed this issue.

Lines 322-336 - The description of the palate of Psittacosaurus doesn’t cite the preliminary discussion of basal ceratopsian palates in Dodson et al., 2010, which is relevant to the discussion.
➔We have separated the pterygoid complex into four bones (pterygoid, ectopterygoid, palatine and vomer) and described each one, citing Dodson et al. 2010 in the “Palate” section (Lines 447-449).

Morschhauser: Authors have addressed this issue.

Line 327 - The pterygoid is labeled simply pterygoid in Figure 4, when the text of the manuscript clearly indicates that it is a complex of the pterygoid, vomer, and palatine. I am interpreting this part of the manuscript as saying that it was impossible to segment out the individual bones, which is perfectly understandable. However, the figure caption should reflect that.
➔We have separated “the pterygoid complex” into four individual bones (palatine, the vomer, pterygoid and ectopterygoid) in the revised manuscript (Lines 454-495).

Morschhauser: Authors have addressed this issue.

Line 322 - There is no palatine indicated in Figure 4. But based on other Psittacosaurus palates (Dodson et al., 2010) I would expect that part of the bone labeled the maxilla in Figure 4B is in fact the palatine.
➔The palatine was separated from the maxilla. Please see Fig. S15 in the supplemental file.

Morschhauser: Authors have addressed this issue.

Line 339-343 - The supraoccipital of IVPP V12617 does not have clear sutures and, despite the description of You and Xu, 2005, I would not hazard to delineate it. It’s not very clear on the actual fossil.
➔Bullar et al. (2019) isolated the supraoccipital of IVPP V12617 based on the suture lines. Although they noted that the suture boundaries are unclear, we think their reconstruction of the supraoccipital based on CT data is reliable.
Morschhauser: Bullar et al. (2019) are very tentative in their description of the supraoccipital. They did not strike me as very certain of their conclusions. I would not base a taxonomically important character on their segmentation based on their description. The parietal of that specimen is pretty badly crushed and the supraoccipital is likely not in good shape either.

Miscellaneous Comments and Corrections Relating to the Revised Manuscript.

Line 183 – The description of the new state (14 dorsal vertebrae) added to Character 238 seems to suggest that this character state is possessed only by P. sibiricus. If that is true then this new character state is uninformative and can only add instability to the analysis.

Line 183 – The description of the new state (14 dorsal vertebrae) added to Character 238 seems to suggest that this character state is possessed only by P. sibiricus. If that is true, then this new character state is uninformative and can only add instability to the analysis. All characters and character states in a phylogenetic analysis must be present in a minimum of 2 taxa (with the exception of basal states present only in the outgroup taxon). The character state of 14 dorsal verteberae should either be added to the 12-13 dorsal vertebrae character state or to the 15 dorsal vertebrae character state, based on the authors’ judgement.

Line 241-243 “Because the basal skull length of ZMNH M12414 is 139.1 mm long and almost equivalent to that of IVPP V12617 (143.7 mm long; Bullar et al., 2019), the former can also be regarded as mature.” – This type of reasoning is not valid. We have good data that size is only a very coarse approximation of maturity in dinosaurs (Barta et al., 2022). While it is the best attempt available for the current specimen, it’s an extremely tentative conclusion at best. Some would argue that the correlation is so weak that it’s not a valid conclusion. It needs to be presented as much more tentative in the paper.

Line 512 “As in other Psittacosaurus, the basioccipital presents as paired processes projecting ventrally to the occipital condyle…” This statement should have an attached citation for the braincase anatomy of other Psittacosaurus specimens.

Line 514. A different statement, but a similar need for a reference.

Line 570 etc. – The description of the mandibular elements seems very telegraphic. Just a single sentence per element in many cases. Seems oddly short compared to the other descriptions. While these may not be very character-dense elements (though the angular, surangular, dentary and articular surely are), it seems odd.

Line 579/580 – Characters are derived, not taxa. P. houi and P. major come out as sister taxa in this analysis. Many people refer to taxa as earlier or later diverging.

Line 713/714 – P. houi appears to be diagnosed by the following unique combination of characters. I would argue that the term “diagnostic character” be reserved for autapomorphies. It is the combination that is here diagnostic, not each individual character, as any one character could unite the specimen with other species of Psittacosaurus, or other more inclusive clades.

Line 722 – There is no left lacrimal shown on Figure 3 (either in 3A or 3C). There also seems to be a sizable crack where the prefrontal, premaxilla and Jugal meet in Figure 3A. The Nasals appear to have multiple fractures and gaps in them. This is not indicative of a lack of distortion.

Problem with the structure of the reasoning to establish P. houi. The authors have to establish that all the skulls being compared are not distorted in a relevant fashion. Both the two skulls of P. houi and the comparative skulls of P. lujiatunensis need to be undistorted in the relevant feature for that character to be valid. The referenced skulls of P. lujiatunensis area also distorted.

Line 727 – Not sure why only figure S14 (the vomer) was referenced here? The break in IVPP V12617 was between the basipterygoid and the rest of the braincase. Vomer is included, but the entire braincase reconstruction in Fig 4 would also support this argument.

Line 799-800 This sentence should be rephrased for clarity. The shared characters are a concrete way of demonstrating the morphological similarity between these two terminal taxa. Maybe, “This result could be attributed to the six shared characters between P. houi and P. amitabha.”

Line 804/805 I don’t think this needs to be a separate paragraph. In listing the characters here, it isn’t clear which relates to which taxon. It looks like you clarify that below.

Line 866 – “without significant postdepositional distortion of the skull” The dorsoventral compression of the P. Amitabha skull is obvious from the published photographs. The quadrate wing of the pterygoid/pterygoid wing of the quadrate are bowed heavily anteriorly, indeed they are creased and broken in R lateral view (Figure 4 B Napoli et al. 2019). Why Napoli didn’t discuss the taphonomic condition of that specimen, I don’t know, but it’s there. It’s not an undeformed skull.


REFERENCES (in just my new comments)

Barta, DE, Griffin, CT, and MA Norell. 2022. Osteohistology of a Triassic dinosaur population reveals highly variable growth trajectories typified early dinosaur ontogeny. Scientific Reports. 12:17321. https://doi.org/10.1038/s41598-022-22216-x

Cullar, CM, Q. Zhao, MJ Benton, and MJ Ryan. 2019. Ontogenetic braincase development in Psittacosaurus lujiatunensis (Dinosauria: Ceratopsia) using micro-computed tomography. PeerJ 7:e7217 DOI 10.7717/peerj.7217

Hedrick BP, Dodson P (2013) Lujiatun Psittacosaurids: Understanding Individual and Taphonomic Variation Using 3D Geometric Morphometrics. PLoS ONE 8(8): e69265. doi:10.1371/journal.pone.0069265

Napoli, JG, T Hunt, GM Erickson, and MA Norell. 2019. Psittacosaurus Amitabha, a new species of ceratopsian dinosaur from the Ondai Sayr locality, Central Mongolia. American Museum Novitates 3932:1-36.

Sereno, PC, Chao S, Cheng, Z, and Rao C. 1988. Psittacosaurus meileyingensis (Ornithischia: Ceratopsia), a new psittacosaur from the Lower Cretaceous of northeastern China. Journal of Vertebrate Paleontology 8:366-377.
You, H.−L., Tanoue, K., and Dodson, P. 2008. New data on cranial anatomy of the ceratopsian dinosaur Psittacosaurus major. Acta Palaeontologica Polonica 53 (2): 183–196.
Zhou, C, K.Q. Gao, R.C. Fox, and X.-K. Du. 2007. Endocranial morphology of Psittacosaurus (Dinosauria: Ceartopsia) based on CT scans of new fossils from the Lower Cretaceous of China. Paleoworld 16:285-293.

---

## Round 0.3 · Minor Revisions

The manuscript is nearly ready to go -- I have marked some minor changes in the attached PDF. Once those are incorporated, I should be able to accept the manuscript.

---

## Round 0.4 · Minor Revisions

Thank you for your close attention to the final round of comments - your manuscript is now ready for publication, but please address the following comment from the Section Editor regarding data availability:

> The CT-data should be made available upon publication and linked to the publication. I could see the model but not the raw data (e.g., image stack), which would be customary to make available upon publication. I feel an explicit mention in the manuscript or on the publication website would be needed, as it is currently hard to understand the context.

---

## Round 0.5 · accepted · Accept

Thank you for your attention to the request from the section editor. The manuscript is ready to proceed to publication.